# Elevated genetic risk for multiple sclerosis emerged in steppe pastoralist populations

William Barrie[1,20,21], Yaoling Yang[2,3,21], Evan K. Irving-Pease[4,21], Kathrine E. Attfield[5,21], Gabriele Scorrano[4,21], Lise Torp Jensen[5,6,21], Angelos P. Armen[5], Evangelos Antonios Dimopoulos[7], Aaron Stern[8], Alba Refoyo-Martinez[4], Alice Pearson[9], Abigail Ramsøe[4], Charleen Gaunitz[4], Fabrice Demeter[4,10], Marie Louise S. Jørkov[11], Stig Bermann Møller[12], Bente Springborg[12], Lutz Klassen[13], Inger Marie Hyldgård[13], Niels Wickmann[14], Lasse Vinner[4], Thorfinn Sand Korneliussen[4], Morten E. Allentoft[4,15], Martin Sikora[4], Kristian Kristiansen[4,16], Santiago Rodriguez[3], Rasmus Nielsen[4,8], Astrid K. N. Iversen[5,17,22✉], Daniel J. Lawson[2,3,22✉], Lars Fugger[5,6,18,22✉] & Eske Willerslev[1,4,19,22✉]

Multiple sclerosis (MS) is a neuro-inflammatory and neurodegenerative disease that is most prevalent in Northern Europe. Although it is known that inherited risk for MS is located within or in close proximity to immune-related genes, it is unknown when, where and how this genetic risk originated[1]. Here, by using a large ancient genome dataset from the Mesolithic period to the Bronze Age[2], along with new Medieval and post-Medieval genomes, we show that the genetic risk for MS rose among pastoralists from the Pontic steppe and was brought into Europe by the Yamnaya-related migration approximately 5,000 years ago. We further show that these MS-associated immunogenetic variants underwent positive selection both within the steppe population and later in Europe, probably driven by pathogenic challenges coinciding with changes in diet, lifestyle and population density. This study highlights the critical importance of the Neolithic period and Bronze Age as determinants of modern immune responses and their subsequent effect on the risk of developing MS in a changing environment.

MS is an autoimmune disease of the brain and spinal cord that currently affects more than 2.5 million people worldwide[1]. Its prevalence varies markedly with ethnicity and geographical location, with the highest prevalence observed in Europe (142.81 cases per 100,000 people); Northern Europeans are particularly susceptible to developing the disease[3]. The origins of and reasons for this geographical variation are poorly understood, yet such biases may hold important clues as to why the prevalence of autoimmune diseases, including MS, has continued to rise during the past 50 years.

Although still elusive, MS aetiology is thought to involve gene–gene and gene–environment interactions. Accumulating evidence suggests that exogenous triggers initiate a cascade of events involving a multitude of cells and immune pathways in genetically vulnerable individuals, which may ultimately lead to MS neuropathology[1].

Genome-wide association studies (GWAS) have identified 233 commonly occurring genetic variants that are associated with MS; 32 variants are located in the human leukocyte antigen (HLA) region and 201 are located outside the HLA region[4]. The strongest MS associations are found in the HLA region, with the most prominent of these, HLA-DRB1*15:01, conferring an approximately threefold increase in the risk of MS in individuals carrying at least one copy of this allele. Collectively, genetic factors are estimated to explain approximately 30% of the overall disease risk, while environmental and lifestyle factors are considered the major contributors to MS. For instance, although infection with Epstein–Barr virus (EBV) frequently occurs in childhood and usually is symptomless, delayed infection into early adulthood, as typically observed in countries with high standards of hygiene, is associated with a 32-fold-increased risk of MS[5,6]. Lifestyle factors associated with increased MS risk, such as smoking, obesity during adolescence

[1]Department of Zoology, University of Cambridge, Cambridge, UK. [2]Department of Statistical Sciences, School of Mathematics, University of Bristol, Bristol, UK. [3]MRC Integrative Epidemiology Unit, Population Health Sciences, University of Bristol, Bristol, UK. [4]Lundbeck Foundation GeoGenetics Centre, Globe Institute, University of Copenhagen, Copenhagen, Denmark. [5]Oxford Centre for Neuroinflammation, Nuffield Department of Clinical Neurosciences, John Radcliffe Hospital, University of Oxford, Oxford, UK. [6]Department of Clinical Medicine, Aarhus University Hospital, Aarhus, Denmark. [7]Pathogen Genomics and Evolution Group, Department of Veterinary Medicine, University of Cambridge, Cambridge, UK. [8]Departments of Integrative Biology and Statistics, University of California, Berkeley, Berkeley, CA, USA. [9]Department of Genetics, University of Cambridge, Cambridge, UK. [10]Eco-anthropologie (EA), Muséum National d'Histoire Naturelle, CNRS, Université de Paris, Musée de l'Homme, Paris, France. [11]Laboratory of Biological Anthropology, Department of Forensic Medicine, University of Copenhagen, Copenhagen, Denmark. [12]Ålborg Historiske Museum, Nordjyske Museer, Vestbjerg, Denmark. [13]Museum Østdanmark–Djursland og Randers, Randers, Denmark. [14]Museum Vestsjælland, Holbæk, Denmark. [15]Trace and Environmental DNA (TrEnD) Laboratory, School of Molecular and Life Sciences, Curtin University, Perth, Western Australia, Australia. [16]Department of Historical Studies, University of Gothenburg, Gothenburg, Sweden. [17]Nuffield Department of Clinical Neurosciences, John Radcliffe Hospital, University of Oxford, Oxford, UK. [18]MRC Human Immunology Unit, John Radcliffe Hospital, University of Oxford, Oxford, UK. [19]MARUM Center for Marine Environmental Sciences and Faculty of Geosciences, University of Bremen, Bremen, Germany. [20]Present address: Department of Genetics, University of Cambridge, Cambridge, UK. [21]These authors contributed equally: William Barrie, Yaoling Yang, Evan K. Irving-Pease, Kathrine E. Attfield, Gabriele Scorrano, Lise Torp Jensen. [22]These authors jointly supervised this work: Astrid K. N. Iversen, Daniel J. Lawson, Lars Fugger, Eske Willerslev. ✉e-mail: astrid.iversen@ndcn.ox.ac.uk; dan.lawson@bristol.ac.uk; lars.fugger@ndcn.ox.ac.uk; ew482@cam.ac.uk

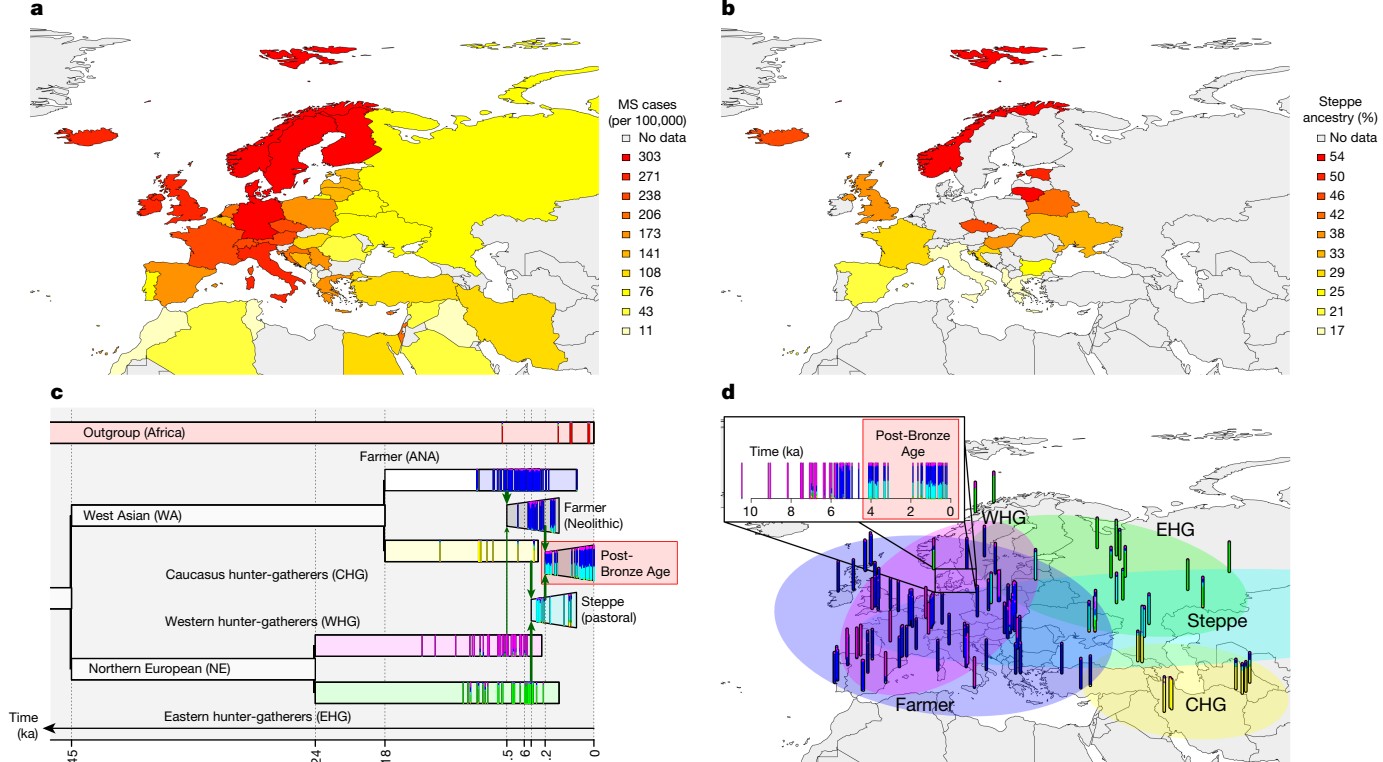

**Fig. 1 | The population history of Europe is associated with the modern-day distribution of MS. a**, The modern-day geographical distribution of MS in Europe. Prevalence data for MS (cases per 100,000) were obtained from ref. 3. **b**, Steppe ancestry in modern samples as estimated by ref. 26. **c,d**, A model of European prehistory[21] onto which our reference samples were projected using non-negative least squares (NNLS) for population painting (see Methods) (**c**) and the same data represented spatially (**d**). Samples are shown as vertical bars representing their 'admixture estimate' obtained by NNLS (see Methods) from six ancestries: EHG (green), WHG (pink), CHG (yellow), farmer (ANA + Neolithic; blue), steppe (cyan) or an outgroup (represented by ancient Africans; red). Important population expansions are shown as growing bars, and 'recent'

(post-Bronze Age) non-reference admixed populations are shown for the Denmark time transect (see Extended Data Fig. 2 for details). Chronologically, WHG and EHG were largely replaced by farmers amid demographic changes during the 'Neolithic transition' around 9,000 years ago. Later migrations during the Bronze Age about 5,000 years ago brought a roughly equal steppe ancestry component from the Pontic-Caspian steppe to Europe, an ancestry descended from the EHG from the middle Don River region and the CHG[2]. Steppe ancestry has been associated with the Yamnaya culture and then with the expansion westwards of the Corded Ware culture and Bell Beaker culture, with eastward expansion in the form of the Afanasievo culture[26,27]. ka, thousand years ago.

and nutrition or gut health, also vary geographically[7]. Autoimmunity could also result from altered pressure from other pathogens, creating a shift in the delicate balance of pro- and anti-inflammatory pathways[8].

European genetic ancestry (henceforth 'ancestry') has been postulated to explain part of the global difference in MS prevalence in admixed populations[9]. Specifically, African American individuals with MS exhibit increased European ancestry in the HLA region compared with control individuals, with European haplotypes conferring more MS risk for most HLA alleles, including HLA-DRB1*15:01. Conversely, Asian American individuals with MS have decreased European ancestry in the HLA region compared with control individuals. Although ancient European ancestry and MS risk in Europe are known to be geographically structured (Fig. 1a,b), the effect of ancestry variation within Europe on MS prevalence is unknown.

Present-day ancestral variation can be modelled as a mixture of genetic ancestries derived from ancient populations, who can be distinguished by their subsistence lifestyle: western hunter-gatherers (WHG), eastern hunter-gatherers (EHG), Caucasus hunter-gatherers (CHG), farmers (Anatolian (ANA) + Neolithic) and steppe pastoralists (Fig. 1c,d). By using a large ancient genome dataset from the Mesolithic to the Bronze Age, presented in an accompanying study[2], coupled with new Medieval and post-Medieval genomes, we quantified present-day European genetic ancestry with respect to these ancestral populations to identify signals of lifestyle-specific evolution. We then determined whether variants associated with an increased risk of MS have

undergone positive selection. We asked when selection occurred and whether the targets of selection were specific to lifestyle. Finally, we examined the environmental conditions that may have caused selection for risk variants, including human subsistence practices and exposure to pathogens. An overview of the evidence provided by all methods used can be found in Extended Data Fig. 1.

To examine the ancestry patterns within modern genomes, we estimated ancestry at specific loci ('local ancestry') for ~410,000 self-identified 'white British' individuals in the UK Biobank[10], using a reference panel of 318 ancient DNA samples (Fig. 1 and Extended Data Fig. 2; ref. 11) from the Mesolithic and Neolithic, including steppe pastoralists (Methods). Comparing the ancestry at each labelled single-nucleotide polymorphism (SNP; $n$ = 549,323) to genome-wide ancestry in the UK Biobank provided an 'anomaly score'. Two regions stood out as having the most extreme ancestry compositions (Fig. 2a): the *LCT/MCM6* region on chromosome 2, which is well established as regulating lactase persistence[11,12], and the HLA region on chromosome 6.

The HLA region is strongly associated with autoimmune diseases[13], of which we examined MS and rheumatoid arthritis (RA), a common systemic inflammatory disease that characteristically affects the joints. Our dataset (comprising a large ancient genome dataset from the Mesolithic to the Bronze Age[2] and 86 new Medieval and post-Medieval genomes from Denmark; Extended Data Fig. 2, Supplementary Note 1 and Supplementary Table 1) includes a total of 1,750 imputed diploid

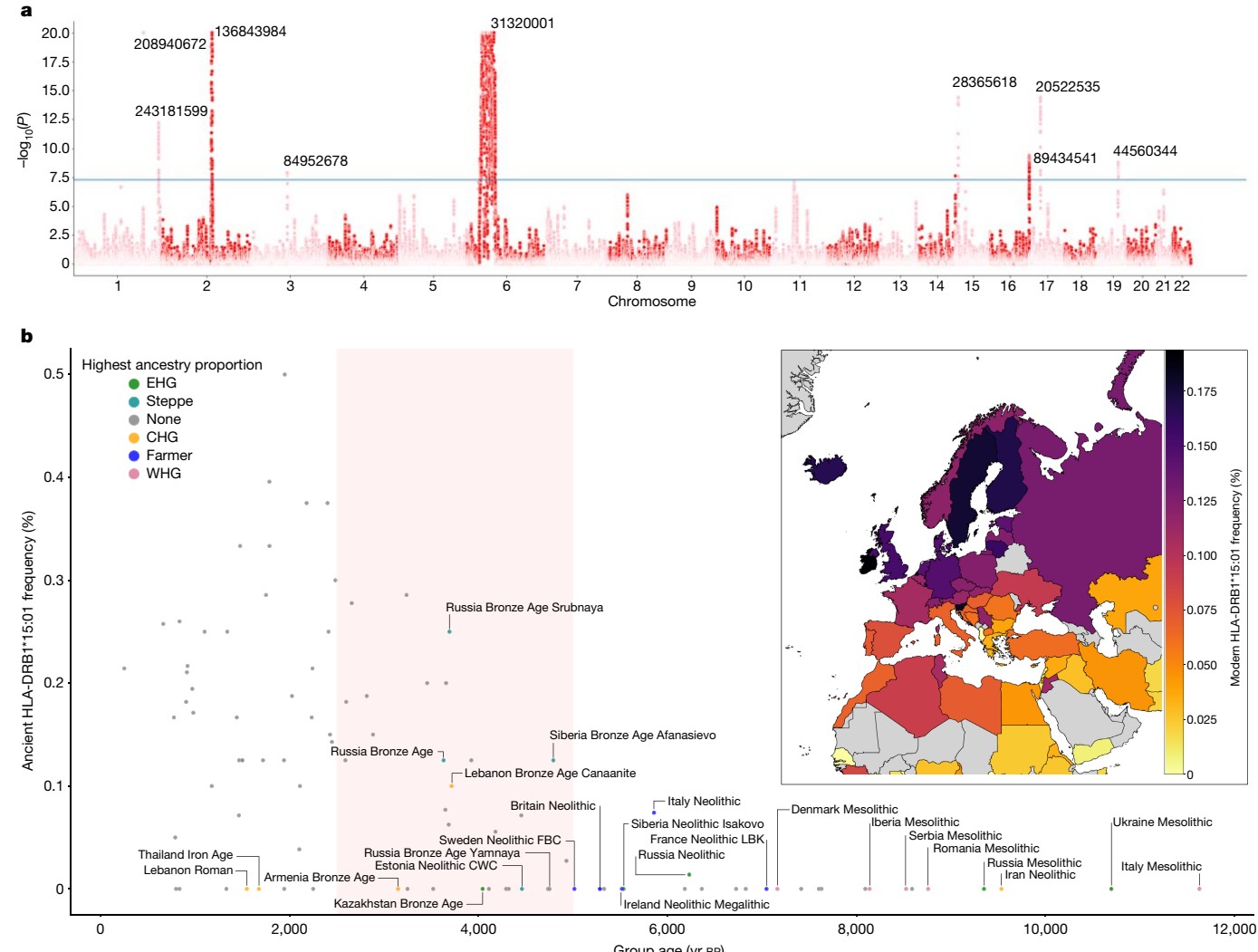

**Fig. 2 | Areas of unusual local ancestry in the genome and ancient and modern frequencies of HLA-DRB1\*15:01. a**, Local ancestry anomaly score measuring the difference between the local ancestry and the genome-wide average (capped at $-\log_{10}(P) = 20$; Methods). Significant peaks (reaching genome-wide significance $P < 5 \times 10^{-8}$, two-tailed $t$ test before adjustment for multiple testing, as shown by the blue horizontal line) are labelled with chromosome position (build GRCh37/hg19). **b**, HLA-DRB1\*15:01 frequency ($y$ axis) in ancient populations over time ($x$ axis; yr BP, years before the present); this is the highest effect variant for MS risk (calculated using the rs3135388

tag SNP). For each ancestry (CHG, EHG, WHG, farmer, steppe), the five populations with the highest amount of that ancestry are labelled; other populations are shown as grey points. HLA-DRB1\*15:01 was present in one sample before the steppe expansion but rose to high frequency during the Yamnaya formation (approximate time period shaded red). The geographical distribution of HLA-DRB1\*15:01 frequency in modern populations from the UK Biobank[11] is also shown (inset; grey represents no data). FBC, funnel beaker culture; LBK, linear pottery culture (Linearbandkeramik); CWC, corded ware culture.

shotgun-sequenced ancient genomes (Supplementary Table 13), of which 1,509 are from Eurasia; together with modern data[10], we achieved an almost complete transect from approximately 10,000 years ago to the present.

The frequencies of the alleles conferring the highest risk for MS (odds ratio (OR) > 1.5), all of which are within the HLA class II region, showed striking patterns in our ancient groups. In particular, the tag SNP (rs3135388[T]) for HLA-DRB1\*15:01, which carries the highest risk for MS (OR = 2.9), was first observed in an Italian Neolithic individual (sample R3 from Grotta Continenza, dated with carbon-14 to between 5836 and 5723 BCE (before common era), 4.05× coverage) and rapidly increased in frequency around the time of the emergence of the Yamnaya culture around 5,300 years ago in steppe and steppe-derived populations (Fig. 2). From risk allele frequencies of individuals in the UK Biobank born in, and having a 'typical ancestral background' for, a specific country[11], we found that the frequency of HLA-DRB1\*15:01 was highest in modern populations from Finland, Sweden and Iceland

and in ancient populations with a high proportion of steppe ancestry (Fig. 2b, inset).

To investigate the risk for a particular genetic ancestry, we used the local ancestry dataset to calculate the risk ratio (Methods; weighted average prevalence, WAP) for each ancestry at all MS-associated fine-mapped loci present in the UK Biobank imputed dataset ($n = 205/233$; ref. 4 and Methods). For MS, steppe ancestry had the highest risk ratio at nearly all HLA SNPs, whereas farmer and outgroup ancestries were often the most protective (Fig. 3a), indicating that a steppe-derived haplotype at these positions confers MS risk.

Having shown that some ancestries carry higher risk at particular SNPs, we wanted to calculate an aggregate risk score for each ancestry. We used a statistic, the ancestral risk score (ARS; introduced in ref. 11), which is equivalent to a polygenic risk score (PRS) for a modern individual consisting entirely of one ancestry. ARS offers an improvement over calculating a PRS using ancient genotype calls directly, as it mitigates the effects of low ancient DNA sample numbers and bias[14] while

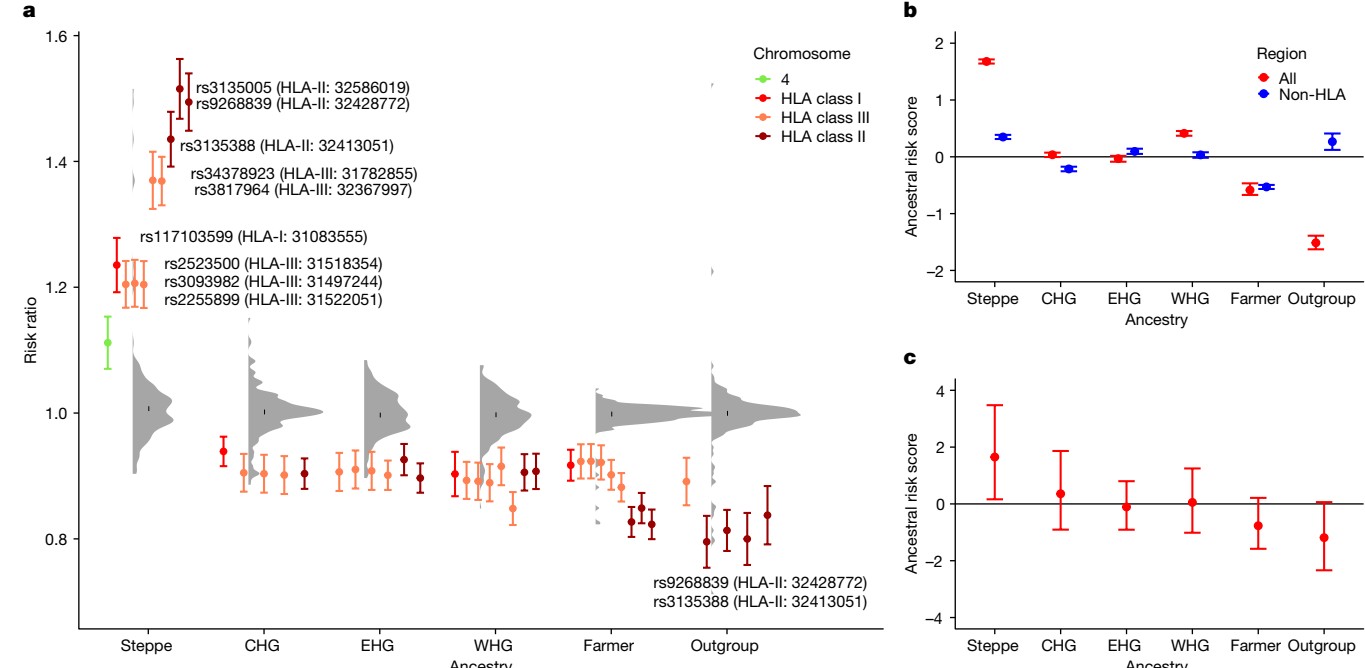

**Fig. 3 | Associations between local ancestry at fine-mapped MS-associated SNPs and MS in a modern population. a**, Risk ratio of SNPs for MS based on WAP (see Methods) when decomposed by inferred ancestry. The mean and s.d. were calculated for each ancestry on the basis of bootstrap resampling for each chromosome ($n = 408,884$ individuals). The distribution of risk ratios for each ancestry is shown as a raincloud plot. SNPs significant at the 1% level are shown individually, coloured by chromosome or HLA region, and those with a risk ratio of >1.2 or <0.8 are annotated with their rsID, HLA region and position (build GRCh37/hg19). **b,c**, ARS (see Methods) for MS. The mean and confidence intervals were estimated by either bootstrapping over individuals (**b**; which can be interpreted as testing the power to reject a null hypothesis of no association between MS and ancestry; $n = 1,000$ bootstrap resamples with replacement over 24,000 individuals) or bootstrapping over SNPs (**c**; which can be interpreted as testing whether ancestry is associated with MS across the genome; $n = 1,000$ bootstrap resamples with replacement over 204 SNPs). We show the results for all associated SNPs (red) and non-HLA SNPs only (blue) when bootstrapping over individuals.

being robust to intervening drift and selection. We used effect size estimates from previous association studies, under an additive model, with confidence intervals obtained via an accelerated bootstrap[15] (Supplementary Note 4). In the ARS for MS (Fig. 3b), steppe ancestry had the largest risk, followed by WHG, CHG and EHG ancestry; the farmer and outgroup ancestries had the lowest ARS. Therefore, steppe ancestry contributes the most risk for MS across all associated SNPs. We tested for a genome-wide association by resampling loci and found that steppe risk still clearly exceeded that for farmers (Fig. 3c). Although most of the signal was driven by SNPs in the HLA region, this pattern persisted even when we excluded these SNPs (Fig. 3b).

The fact that steppe ancestry confers risk at all but two MS-associated HLA SNPs (Fig. 3a) implies that these alleles have a common evolutionary history. We therefore investigated whether ancestry could be used for phenotype prediction. We conducted three types of association analysis in the UK Biobank for disease-associated SNPs, controlling for age, sex and the first 18 principal components. The first was a regular SNP-based association analysis, as in a genome-wide association study. The second tested for association with local ancestry probabilities instead of genotype values (Supplementary Note 3). The third was based on haplotype trend regression (HTR), which is used to detect interactions between SNPs[16] by treating haplotypes as a set of features from which to predict a trait, instead of using SNPs as in a regular genome-wide association study. We developed a new method called HTR with extra flexibility (HTRX; Supplementary Note 5 and more details in ref. 17) that searches for haplotype patterns that include single SNPs and non-contiguous haplotypes. To evaluate the performance of our models and prevent overfitting, we assessed its ability to predict out-of-sample data, which measures how well the model can generalize to new data. We showed by simulation (Supplementary Fig. 11) that

HTRX explains the same amount of variance as a regular genome-wide association study when interactions are absent and more variance as interaction strength increases.

Although our cohort of self-identified white British individuals is relatively underpowered with respect to MS (1,949 cases and 398,049 controls; prevalence of 0.487%), MS was associated with steppe and farmer ancestry ($P < 1 \times 10^{-10}$) in the HLA region (Supplementary Fig. 6). In three of four main linkage disequilibrium (LD) blocks within the HLA region (class I, two subregions of class II determined by LD blocks at 32.41–32.68 Mb and 33.04–33.08 Mb, and class III), local ancestry explained significantly more variation than genotypes (Fig. 4; measured by average out-of-sample McFadden's $R^2$ for logistic regression; Methods). While the increased performance of local ancestry in some regions compared with regular GWAS can be explained by tagging of SNPs outside the region, the increased performance of HTRX over GWAS quantifies the total effect of a haplotype, including rare SNPs and epistasis. Across the entire HLA region, haplotypes explained more out-of-sample variation than regular GWAS (at least 2.90%, compared to 2.48%). Interaction signals were also observed within the HLA class I region, within the HLA class II region, and between the HLA class I and class III regions.

We further tested whether co-occurring ancestries at each locus were associated with MS (see Methods and Supplementary Fig. 7) but found no evidence that risk was associated with any ancestry other than steppe ancestry.

Having established that steppe ancestry contributes most of the HLA-associated risk for MS, we investigated whether MS risk evolved under selection. We tested for evidence of directional selection across all associated SNPs, decomposed by ancestry, over time. This test used a 'pathway-based chromosome painting' technique (see Methods)

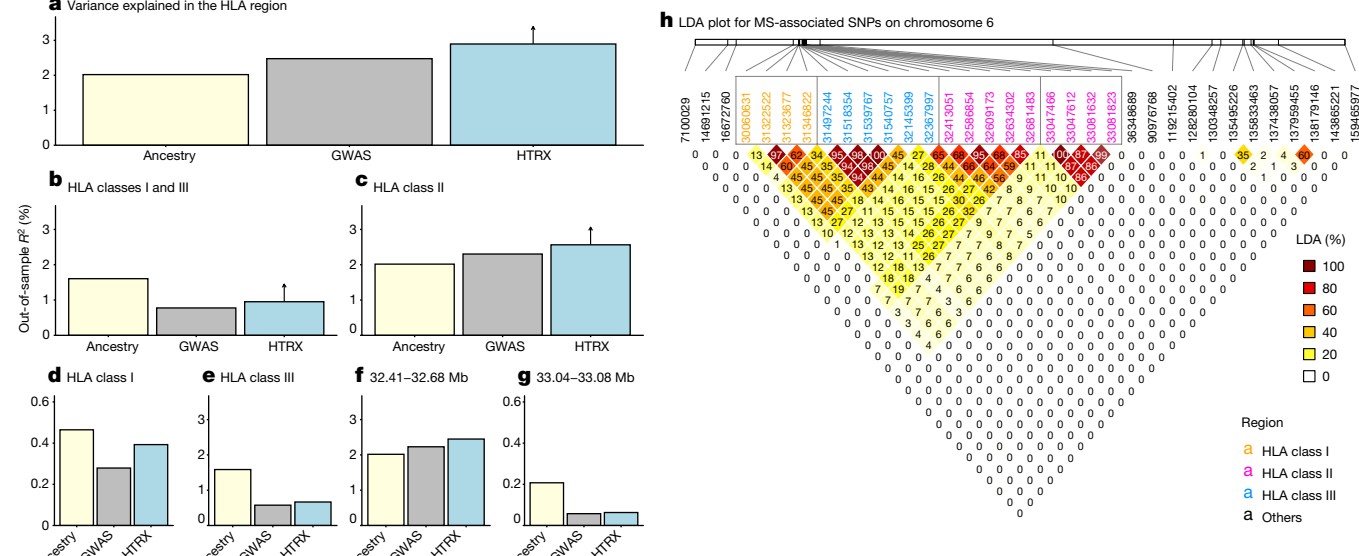

**Fig. 4 | MS association in the HLA region. a–g,** Comparison of variance explained in MS within the UK Biobank for all fine-mapped HLA SNPs with an independent contribution[4]. The plots compare GWAS (treating SNPs as having independent effects), local ancestry at the SNPs and HTRX (haplotypes), after accounting for covariates (Methods), for fine-mapped MS-associated SNPs in the HLA region (**a**), the HLA class I and class III regions (**b**), the HLA class II region (**c**), the HLA class I region (**d**), the HLA class III region (**e**) and subregions of the HLA class II region chosen from LD (**f,g**). Upward-pointing arrows for HTRX indicate where the values are lower bounds (Methods). **h,** Genetic correlations in the HLA region at our time depth from ancestry-based LD (LDA; Methods; see Supplementary Fig. 50 for LD).

based on inference of a sample's nearest neighbours in the marginal trees of an ancestral recombination graph (ARG) that contains labelled individuals[11]. The resulting ancestral path labels, for haplotypes in both ancient and modern individuals, allowed us to infer allele frequency trajectories for risk-associated variants while controlling for changes in admixture proportions over time. The paths extend backwards from the present day to approximately 15,000 years ago and are labelled with the unique population through which a path travels (ANA, CHG, EHG or WHG). Because it uses distinct pathways, the approach does not use the labels of the relatively recent steppe admixture or outgroup populations, and the path labels are not representative of a continuous population but rather represent a path backwards in time that encompasses the corresponding population. For example, the CHG path originates in the CHG population, before merging with EHG to form the steppe population, and then merges with other ancestries in later European populations (Fig. 1).

In our ancestry path analysis, a substantial fraction of the fine-mapped MS-associated variants were not imputed in our ancient dataset, owing to quality-control filtering and the difficulty of accurately inferring HLA alleles in ancient samples[18]. To address this, we LD pruned genome-wide-significant summary statistics from the same study[4], for which we could reliably assign ancestry path labels (*n* = 62; see Methods). This allowed us to test for polygenic selection across disease-associated variants using CLUES[19] and PALM[20].

For MS, we found evidence that disease risk was selectively increased, when considering all ancestries collectively ($P = 1.02 \times 10^{-5}$, polygenic selection gradient ($\omega$) = 0.017), between 5,000 and 2,000 years ago (Fig. 5). Conditioning on each of the four long-term ancestral paths (CHG, EHG, WHG and ANA), we found a statistically significant signal of selection in the WHG ($P = 7.22 \times 10^{-5}$, $\omega$ = 0.021), EHG ($P = 2.60 \times 10^{-3}$, $\omega$ = 0.016) and CHG ($P = 3.06 \times 10^{-2}$, $\omega$ = 0.009) paths but not in the ANA path ($P = 0.64$, $\omega$ = 0.004). Again, it is likely that selection occurred in the pastoralist population of the steppe, as that population consisted of approximately equal proportions of EHG and CHG ancestry[21] (Fig. 1). The SNP driving the largest change in genetic risk over time in the pan-ancestry analysis was rs3129934 ($P = 1.31 \times 10^{-11}$, selection

coefficient (*s*) = 0.018), which tags the HLA-DRB1*15:01 haplotype[22]. We also tested three other SNPs that tag the HLA-DRB1*15:01 haplotype (rs3129889, rs3135388 and rs3135391) for evidence of selection and found that the ancestry-stratified signal was consistently strongest in CHG (Fig. 5b).

To further examine the nature of selection, we developed a new summary statistic: linkage disequilibrium of ancestry (LDA). LDA is the correlation between local ancestries at two SNPs, measuring whether recombination events between ancestries have occurred at a high frequency compared with recombination events within ancestries. We subsequently defined the 'LDA score' of a SNP as the total LDA of the SNP with the rest of the genome. A high LDA score indicates that the haplotype inherited from the reference population is longer than expected, whereas a low score indicates that the haplotype is shorter than expected (that is, underwent more recombination). For example, the *LCT/MCM6* region exhibited a high LDA score (Extended Data Fig. 3), as expected from a relatively recent selective sweep[23].The HLA region had significantly lower LDA scores than the rest of chromosome 6 (Extended Data Fig. 3). Through simulations, we showed that this signal must have been driven by selection favouring haplotypes of mixed ancestry over single-ancestry haplotypes (Supplementary Figs. 46–48 and Methods). Extending multi-SNP selection models[24], our explanation is that at least two separate loci arose selectively in separate populations that later admixed and remained selected in the HLA region, justifying a new term, 'recombinant-favouring selection'. This means that there was selection for diverse ancestry in the HLA region, driven by recombination. Unlike other measures of balancing selection such as $F_{ST}$, LDA describes excess ancestry LD from specific, dated populations and therefore is an independent signal. For the HLA class II region, the selection measures all lined up (LDA score, $F_{ST}$ and $\pi$; Extended Data Fig. 4), but for the HLA class I region the LDA score had an additional non-diverse minimum at 30.8 Mb, implying that here the genome is ancestrally diverse but genetically strongly constrained. The LDA score is thus informative about the type of selection being detected and whether it has been subject to change.

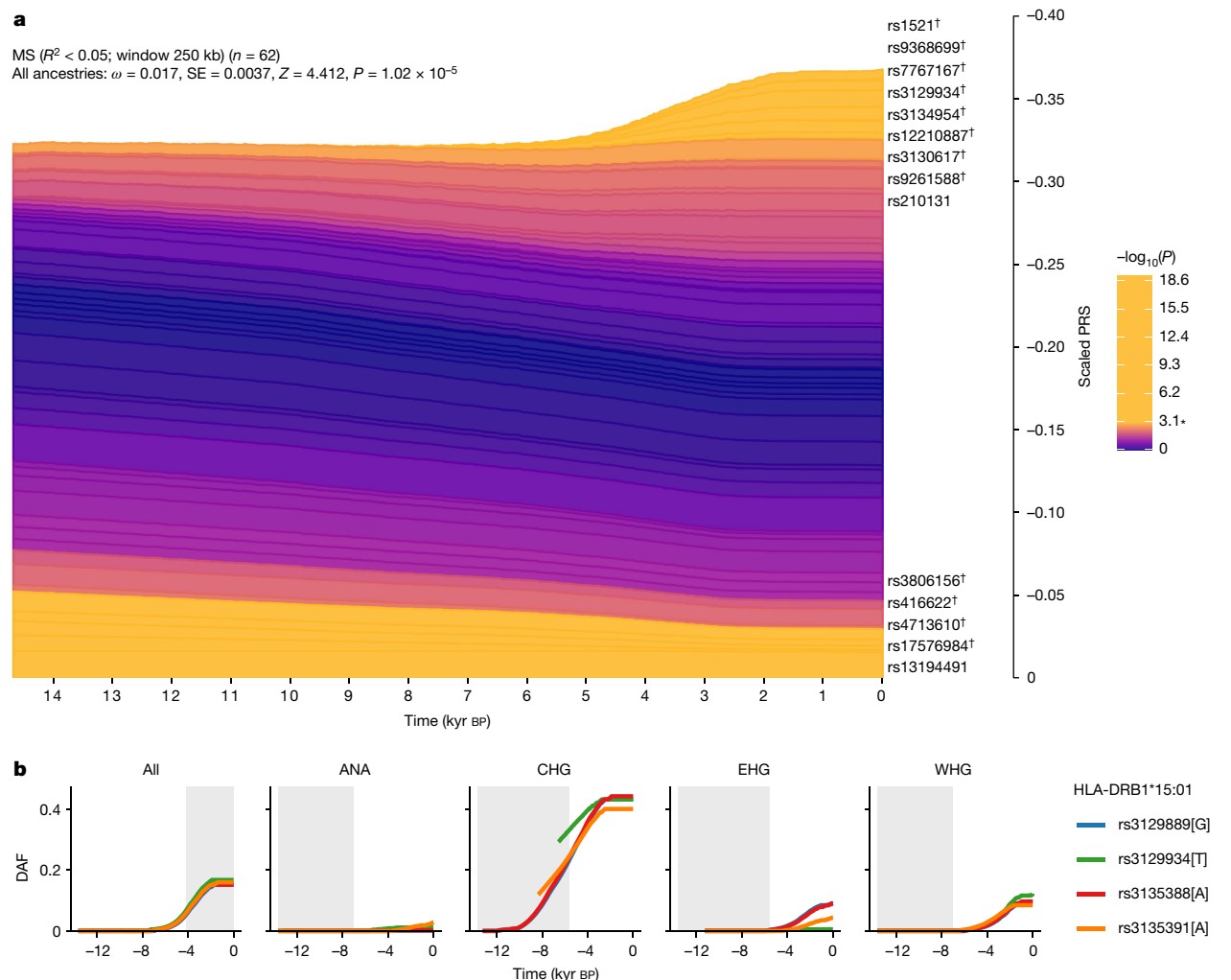

**Fig. 5 | Evidence for selection on MS-associated SNPs. a**, Stacked line plot of the pan-ancestry PALM analysis for MS, showing the contribution of SNPs to disease risk over time. SNPs are shown as stacked lines, with the width of each line proportional to the population frequency of the positive risk allele, weighted by its effect size. When a line widens over time, the positive risk allele has increased in frequency, and vice versa. SNPs are sorted by the magnitude and direction of selection, with positively selected SNPs at the top, negatively selected SNPs at the bottom and neutral SNPs in the middle. SNPs are coloured by their corresponding $P$ value in a single-locus selection test. The asterisk marks the Bonferroni-corrected significance threshold, and nominally significant SNPs are shown in yellow and labelled by their rsID. SNPs marked with the dagger symbol are located in the HLA locus. The $y$ axis shows the scaled average PRS in the population, ranging from 0 to 1, with 1 corresponding to the maximum possible average PRS (that is, when all individuals in the population are homozygous for all positive risk alleles), and the $x$ axis shows time in units of thousands of years before the present. SE, standard error. **b**, Maximum-likelihood trajectories for four SNPs tagging HLA-DRB1*15:01, for all ancestry paths combined (All) and for each path separately (Extended Data Fig. 1 and Methods). Portions of the trajectories with high uncertainty (that is, posterior density of <0.08) have been masked. The background is shaded for the approximate time period in which the ancestry existed as an actual population. The $y$ axis shows the derived allele frequency (DAF), and the $x$ axis shows time in units of thousands of years before the present.

Because MS would not have conferred a fitness advantage on ancient individuals, it is likely that this selection was driven by traits with shared genetic architecture, of which increased risk for MS in the present is a pleiotropic by-product. We therefore looked at LD-pruned MS-associated SNPs that showed statistically significant evidence for selection using CLUES ($n = 32$) in one or more ancestries and which also had a genome-wide-significant trait association ($P < 5 \times 10^{-8}$) for any of the 4,359 traits from the UK Biobank (ref. 10; UK Biobank Neale laboratory, round 2; http://www.nealelab.is/uk-biobank/) and any of the 2,202 traits in the FinnGen study[25]. We observed that all selected SNPs were also associated with multiple other traits (Supplementary Figs. 19–27). To determine whether the observed signal of polygenic selection favouring MS risk could be better explained by selection acting on a genetically correlated trait, we performed a systematic analysis of traits in UK Biobank and FinnGen with at least 20% overlap among the MS-associated selected SNPs ($n = 115$ traits). Using a joint test in PALM specifically designed for disentangling polygenic selection on correlated traits, we found no UK Biobank or FinnGen traits for which the selection signal favouring MS risk was significantly attenuated by selection acting on a genetically correlated trait, when accounting for the number of tests (Supplementary Note 6). This demonstrates that the selection signal for MS could not be explained by selection acting on any genetically correlated trait that we tested.

Because both the UK Biobank and FinnGen are underpowered with respect to many traits and diseases, we also undertook a manual literature search (Methods) for all LD-pruned MS-associated SNPs that showed statistically significant evidence for selection using CLUES ($n = 32$, of which 25 (78%) are in the HLA region). We found that most of the alleles under positive selection were associated with protective effects against specific pathogens and/or infectious diseases (disease or pathogen

associated/total selected in ancestry path: pan-ancestry, 11/14; ANA, 8/9; CHG, 6/9; EHG, 6/7; WHG, 17/18; Supplementary Note 8, Supplementary Table 11 and Extended Data Fig. 5), although we note that GWAS data are not available for many infectious diseases. We observed that the selected alleles had protective associations with several chronic viruses (EBV, varicella-zoster virus, herpes simplex virus and cytomegalovirus) and with viruses or diseases not associated with transmission in small hunter-gatherer groups (for example, mumps and influenza). Moreover, many selected alleles conferred a reduction of risk for parasites, for skin and subcutaneous tissue, gastrointestinal, respiratory, urinary tract and sexually transmitted infections, or for pathogens associated with these or other infections (for example, *Clostridioides difficile*, *Streptococcus pyogenes*, *Mycobacterium tuberculosis* and coronavirus) (Supplementary Note 8, Supplementary Table 11 and Extended Data Fig. 5). We emphasize that, although this evidence is strongly suggestive, many of these putative associations may not be statistically robust owing to underpowered GWAS and the bias in candidate gene studies.

We compared these findings for MS with results for RA, which in contrast to MS is a systemic inflammatory disease, although it is mostly known for its characteristic joint lesions[13]. Our findings for RA show a strikingly different ancestry risk profile. HLA-DRB1*04:01 is the largest genetic risk factor for RA; in CLUES analysis, the tag SNP for this allele (rs660895) showed evidence of continuous negative selection until approximately 3,000 years ago ($P = 7.95 \times 10^{-7}$; Extended Data Fig. 6). We found that WHG and EHG ancestries often conferred the most risk at SNPs associated with RA (relative risk ratio of RA-associated SNPs based on WAP; see Methods), and these ancestries contributed the greatest risk for RA in aggregate, as reflected by a higher ARS for these ancestries (Supplementary Note 4), while the steppe and outgroup ancestries had the lowest scores (Extended Data Fig. 7). These results were recapitulated in a local ancestry GWAS (Supplementary Note 3).

We found that RA-associated SNPs have undergone negative polygenic selection ($P = 3.26 \times 10^{-3}$; Extended Data Fig. 6) over the last approximately 15,000 years. When decomposing by ancestry path, we found that all paths exhibited a negative selection gradient; none achieved nominal significance, although the CHG path came close ($P = 6.33 \times 10^{-2}$, $\omega = -0.014$).

These results demonstrate that genetic risk for RA was higher in the distant past, in contrast to MS, with RA-associated risk variants present at higher frequencies in European hunter-gatherer populations before the arrival of agriculture. To understand what might underlie the higher genetic risk in hunter-gatherer populations and subsequent negative selection, we again undertook a manual literature search for pleiotropic effects of LD-pruned SNPs that showed statistically significant evidence of selection ($n = 55$, of which 36 (65%) were in the HLA region). We found that the majority of selected SNPs were associated with protection against distinct pathogens and/or infectious diseases across all paths (disease or pathogen associated/total selected in ancestry path: pan-ancestry, 16/20; ANA, 12/16; CHG, 8/13; EHG, 14/20; WHG, 16/21). We found that selected RA risk alleles were typically linked to the same pathogens or diseases as in the MS analysis, although some SNPs were protective against pathogens or diseases not observed in the MS risk analysis (for example, *Entamoeba histolytica*, measles, viral hepatitis, arthropod-borne viral fevers and viral haemorrhagic fevers, and pneumococcal pneumonia; Supplementary Note 8, Supplementary Table 12 and Extended Data Fig. 5).

## Discussion

The last 10,000 years have seen some of the most extreme global changes in lifestyle, with the emergence of farming in some regions and pastoralism in others. While 5,000 years ago farmer ancestry predominated across Europe, a relatively diverged genetic ancestry arrived with the steppe migrations around this time[26,27]. We have shown that this genetic ancestry contributes the most genetic risk for MS today and

that these variants were the result of positive selection coinciding with the emergence of a pastoralist lifestyle on the Pontic-Caspian steppe and continued selection in the subsequent admixed populations in Europe. These results address the long-standing debate around the north–south gradient in MS prevalence in Europe and indicate that the steppe ancestry gradient in modern populations—specifically in the HLA region—across the continent may cause this phenomenon, in combination with environmental factors. Furthermore, although epistasis between MS-associated variants in the HLA region has been demonstrated before[28–31], we have shown that accounting for this explains more variance than independent SNP effects alone. Many of the haplotypes carrying these risk alleles have ancestry-specific origins, which could be exploited for individual risk prediction and may offer a pathway from genetic ancestry associations to a mechanistic understanding of MS risk. We have compared these findings with results for RA, another HLA class II-associated chronic inflammatory disease, and found that the genetic risk for RA exhibits a contrasting pattern; for RA, genetic risk was highest in Mesolithic hunter–gatherer ancestry and has decreased over time.

Our interpretation of this history is that co-evolution between a range of pathogens and their human hosts may have resulted in massive and divergent genetic ancestry-specific selection on immune response genes according to lifestyle and environment followed by recombinant-favouring selection after these populations merged. Similar examples of pathogen-driven evolution have recently been published[32,33]. The late Neolithic and Bronze Age were a time of massively increased prevalence of infectious diseases in human populations, owing to increased population density as well as contact with, and consumption of, domesticated animals and their products. The most recent common ancestor of many disease-associated pathogens existed in this period[34–42]; although these diseases are common today, it is difficult to infer their geographical ranges in the past, which may have been more limited[43]. We have shown that many of the MS- and RA-associated variants under selection confer some resistance to a range of infectious diseases and pathogens (Supplementary Note 8; for example, HLA-DRB1*15:01 is associated with protection against tuberculosis[44] and increased risk for lepromatous leprosy[45]). We were, however, underpowered to detect specific associations beyond this hypothesis owing to poor knowledge of the distribution and diversity of past diseases, poor preservation of endogenous pathogens in the archaeological record and a lack of well-powered GWAS for many infectious diseases, partly owing to widespread vaccination programmes. Together, these findings indicate that population dispersals, changing lifestyles and increased population density may have resulted in high and sustained transmission of both new and old pathogens, driving selection of variants in immune response genes, which are now associated with autoimmune diseases.

A pattern that repeatedly appears is that of lifestyle change driving changes in risk and phenotypic outcomes. Our data indicate that, in the past, environmental changes driven by lifestyle innovation may have inadvertently driven an increase in genetic risk for MS. Today, with increasing prevalence of MS cases observed over the last five decades[46,47], we again observe a striking correlation with changes in our environment, including lifestyle choices and improved hygiene, which no longer favours the previous genetic architecture. Instead, the fine balance of genetically driven cell functions within the immune system, which are needed to combat a broad repertoire of pathogens and parasites without harming self-tissue, has been met with new challenges, including a potential absence of requirement. For example, while a population of immune cells, CD4+ T helper type 1 ($T_H1$) cells, direct strong cellular immune responses against intracellular pathogens, T helper type 2 ($T_H2$) cells mediate humoral immune responses against extracellular bacteria and parasites and aid tissue homeostasis and repair. We have shown that the majority of selected MS-associated SNPs are associated with protection against a wide range of infectious

challenges, in line with selection for strong but balanced $T_H1/T_H2$ immunity in the Bronze Age. The skewed $T_H1/T_H2$ balance observed in MS may partly result from the developed world's increased sanitation, which has led to a substantially reduced burden of parasites, which the immune system had evolved to efficiently combat[48].

Similarly, the new pathogenic challenges associated with agriculture, animal domestication, pastoralism and higher population densities might have substantially increased the risk of triggering a systemic RA-associated inflammatory state in genetically predisposed individuals. This could have led to an increased risk of a serious outcome following subsequent infections[49], years before any potential joint lesions[50], resulting in negative selection and might thus represent a parallel between RA-associated inflammation in the Bronze Age and MS today, in which lifestyle changes have exposed previously favourable genetic variants as risks for autoimmune disease.

More broadly, it is clear that the late Neolithic and Bronze Age were a critical period in human history during which highly genetically and culturally divergent populations evolved and mixed[2]. These separate histories probably dictate the genetic risk and prevalence of several autoimmune diseases today. Unexpectedly, the emergence of the pastoralist steppe lifestyle may have had an impact on immune responses as great as or greater than that of the emergence of farming during the Neolithic transition, which is commonly held to be the greatest lifestyle change in human history.

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

## Methods

### Data generation

**Overview.** To examine variants associated with phenotypes backwards in time, we assembled a large ancient DNA dataset. Here we present new genomic data from 86 ancient individuals from Medieval and post-Medieval periods from Denmark (Extended Data Fig. 2, Supplementary Note 1 and Supplementary Table 1). The samples range in age from around the eleventh to the eighteenth century. We extracted ancient DNA from tooth cementum or petrous bone and shotgun sequenced the 86 genomes to a depth of genomic coverage ranging from 0.02× to 1.6× (mean of 0.39× and median of 0.27×). The genomes of the 86 new individuals were imputed using 1000 Genomes phased data as a reference panel by an imputation method designed for low-coverage genomes (GLIMPSE)[51], and we also imputed 1,664 ancient genomes presented in the accompanying study[2]. Depending on the specific data quality requirements for the downstream analyses, we filtered out samples with poor coverage and variant sites with low minor allele frequency (MAF) and low imputation quality (average genotype probability of <0.98). Our dataset of ancient individuals spans approximately 15,000 years across Eurasia (Extended Data Fig. 2).

Authorizations for excavating the three sites, Kirkegård, Holbæk and Tjærby, were granted, respectively, to the Aalborg Historiske Museum, the Museum Vestsjælland (previously Museet for Holbæk og Omeg) and the Kulturhistorisk Museum Randers. The current study of samples from these three sites is covered by agreements given to GeoGenetics, Globe Institute, University of Copenhagen, by the Aalborg Historiske Museum, the Museum Vestsjælland and the Kulturhistorisk Museum Randers, respectively.

**Ancient DNA extraction and library preparation.** Laboratory work was conducted in the dedicated ancient DNA clean-room facilities at the Lundbeck Foundation GeoGenetics Centre (Globe Institute, University of Copenhagen). A total of 86 Medieval and post-Medieval human samples from Denmark (Supplementary Table 2) were processed using semi-automated procedures. Samples were processed in parallel. For each extract, non-USER-treated and USER-treated (NEB) libraries were built[52]. All libraries were sequenced on the NovaSeq 6000 instrument at the GeoGenetics Sequencing Core, Copenhagen, using S4 200-cycle kits v1.5. A more detailed description of DNA extraction and library preparation can be found in Supplementary Note 1.

**Basic bioinformatics.** The sequencing data were demultiplexed using the Illumina software BCL Convert (https://emea.support.illumina.com/sequencing/sequencing_software/bcl-convert.html). Adaptor sequences were trimmed and overlapping reads were collapsed using AdapterRemoval (v2.2.4)[53]. Single-end collapsed reads of at least 30 bp and paired-end reads were mapped to human reference genome build 37 using BWA (v0.7.17)[54] with seeding disabled to allow for higher sensitivity. Paired- and single-end reads for each library and lane were merged, and duplicates were marked using Picard MarkDuplicates (v2.18.26; http://picard.sourceforge.net) with a pixel distance of 12,000. Read depth and coverage were determined using samtools (v1.10)[55] with all sites used in the calculation (-a). Data were then merged to the sample level and duplicates were marked again.

**DNA authentication.** To determine the authenticity of the ancient reads, post-mortem DNA damage patterns were quantified using mapDamage2.0 (ref. 56). Next, two different methods were used to estimate the levels of contamination. First, we applied ContamMix to quantify the fraction of exogenous reads in the mitochondrial reads by comparing the mitochondrial DNA consensus genome to possible contaminant genomes[57]. The consensus was constructed using an in-house Perl script that used sites with at least 5× coverage, and bases were only called if observed in at least 70% of reads covering the site. Additionally, we applied ANGSD (v0.931)[58] to estimate nuclear contamination by quantifying heterozygosity on the X chromosome in males. Both contamination estimates used only filtered reads with a base quality of ≥20 and mapping quality of ≥30.

**Imputation.** We combined the 86 newly sequenced Medieval and post-Medieval Danish individuals with 1,664 previously published ancient genomes[2]. We then excluded individuals showing contamination (more than 5%), low autosomal coverage (less than 0.1×) or low genome-wide average imputation genotype probability (less than 0.98), and we chose the higher-quality sample in a close relative pair (first- or second-degree relatives). A total of 1,557 individuals passed all filters and were used in downstream analyses. We restricted the analysis to SNPs with an imputation INFO score of ≥0.5 and MAF of ≥0.05.

### Kinship analysis and uniparental haplogroup inference

READ[59] was used to detect the degree of relatedness for pairs of individuals.

The mitochondrial DNA haplogroups of the Medieval and post-Medieval individuals were assigned using HaploGrep2 (ref. 60; Supplementary Fig. 3). Y-chromosome haplogroup assignment was inferred following an already published workflow[61] (Supplementary Fig. 5). More details can be found in Supplementary Note 2.

### Standard population genetics analyses

The main population genetics approach on which we based our inference was population-based painting (detailed below). However, to robustly understand population structure, we applied other standard techniques. First, we used principal-component analysis (PCA) (Extended Data Fig. 2) to investigate the overall population structure of the dataset. We used PLINK[62], excluding SNPs with MAF < 0.05 in the imputed panel. On the basis of 1,210 ancient western Eurasian imputed genomes, the Medieval and post-Medieval samples clustered close to each other, displaying a relatively low genetic variability and situated within the genetic variability observed in the post-Bronze Age western Eurasian populations.

We then used two additional standard methods to estimate ancestry components in our ancient samples. First, we used model-based clustering (ADMIXTURE)[63] (Supplementary Note 1 and Supplementary Fig. 1) on a subset of 826,248 SNPs. Second, we used qpAdm[64] (Supplementary Note 1, Supplementary Fig. 2 and Supplementary Table 15) with a reference panel of three genetic ancestries (WHG, ANA and steppe) on the same 826,248 SNPs. We performed qpAdm applying the option 'allsnps: YES' and a set of seven outgroups was used as 'right populations': Siberia_UpperPaleolithic_UstIshim, Siberia_UpperPaleolithic_Yana, Russia_UpperPaleolithic_Sunghir, Switzerland_Mesolithic, Iran_Neolithic, Siberia_Neolithic and USA_Beringia. We set a minimum threshold of 100,000 SNPs, and only results with $P < 0.05$ were considered.

### Population painting

Our main analysis used chromosome painting[65] with a panel of six ancient ancestries. This allows fine-scale estimation of ancestry as a function of these populations. We ran chromosome painting on all ancient individuals not in the reference panel, using a reference panel of ancient donors grouped into populations to represent specific ancestries: WHG, EHG, CHG, farmer (ANA + Neolithic), steppe and African (method described in ref. 11). Painting followed the pipeline of ref. 66 based on GLOBETROTTER[67], with admixture proportions estimated using NNLS. NNLS explains the genome-wide haplotype matches of an individual as a mixture of genome-wide haplotype matches of the reference populations. This set-up allows both the reference panel and any additional samples to be described using these six ancestries (Fig. 1).

We then painted individuals born in Denmark of a typical ancestry (typical on the basis of density-based clustering of the first 18 principal components[11]). The reference panel used for chromosome painting was

designed to capture the various components of European ancestry only, and so we urge caution in interpreting these results for non-European samples.

This dataset provides the opportunity to study the population history of Denmark from the Mesolithic to the post-Medieval period, covering around 10,000 years, which can be considered a typical Northern European population. Our results clearly demonstrate the impact of previously described demographic events, including the influx of Neolithic farmer ancestry ~9,000 years ago and steppe ancestry ~5,000 years ago[26,27]. We highlight genetic continuity from the Bronze Age to the post-Medieval period (Supplementary Note 1 and Supplementary Fig. 1), although qpAdm detected a small increase in the farmer component during the Viking Age (Supplementary Note 1, Supplementary Fig. 2 and Supplementary Table 15), while the Medieval period marked a time of increased genetic diversity, probably reflecting increased mobility across Europe. This genetic continuity is further confirmed by the haplogroups identified in the uniparental genetic markers (Supplementary Note 2). Together, these results indicate that after the steppe migration in the Bronze Age there may have been no other major gene flow into Denmark from populations with significantly different Neolithic and Bronze Age ancestry compositions and therefore no changes in these ancestry components in the Danish population.

## Local ancestry from population painting

Chromosome painting provides an estimate of the probability that an individual from each reference population is the closest match to the target individual at every position in the genome. This provided our first estimate of local ancestry from ref. 2: the population of the first reference individual to coalesce with the target individual, as estimated by Chromopainter[65]. This was estimated for all white British individuals in the UK Biobank, using the population painting reference panel described above. We refer to this as 'local ancestry', although we note that the closest relative in the sample may not represent ancestry in the conventional sense.

## Pathway painting

An alternative approach is to identify to which of the four major ancestry pathways (ANA farmer, CHG, EHG and WHG) each position in the genome best matches. This has the advantage of not forcing haplotypes to choose between 'steppe' ancestry and its ancestors but the disadvantage of being more complex to interpret. To do this, we modelled ancestry path labels in the GBR, FIN and TSI 1000 Genomes populations[68] and 1,015 ancient genomes generated using a neural network to assign ancestry paths on the basis of a sample's nearest neighbours at the first five informative nodes of a marginal tree sequence, with an informative node defined as a node that had at least one leaf from the reference set of ancient samples described above (ref. 11; Supplementary Note 1c). We refer to these as 'ancestry path labels'.

## SNP associations

We aimed to generate SNP associations from previous studies for each phenotype in a consistent approach. To generate a list of SNPs associated with MS and RA, we used two approaches: in the first, we downloaded fine-mapped SNPs from previous association studies. For each fine-mapped SNP, if the SNP did not have an ancestry path label, we found the SNP with the highest LD that did, with a minimum threshold of $r^2 \geq 0.7$, in the GBR, FIN and TSI 1000 Genomes populations using LDLinkR[69]. The final SNPs used for each phenotype can be found in Supplementary Table 4 (MS) and Supplementary Table 5 (RA).

For MS, we used data from ref. 4. For non-MHC SNPs, we used the 'discovery' SNPs with $P$(joined) and OR(joined) generated in the replication phase. For MHC variants, we searched the literature for the reported

HLA alleles and amino acid polymorphisms (Supplementary Table 3). In total, we generated 205 SNPs that were either fine-mapped or in high LD with a fine-mapped SNP (15 MHC, 190 non-MHC).

For RA, we downloaded 57 genome-wide-significant non-MHC SNPs for seropositive RA in Europeans[70]. We retrieved MHC associations separately (ref. 71; with associated ORs and $P$ values from ref. 72). In total, we generated 51 SNPs that were either fine-mapped or in high LD with a fine-mapped SNP (3 MHC, 48 non-MHC).

Second, because we could not always find LD proxies for fine-mapped SNPs that were present in our ancestry path label dataset, we found that we were losing significant signal from the HLA region; therefore, we generated a second set of SNP associations. We downloaded full summary statistics for each disease (using ref. 4 for MS and ref. 73 for RA), restricted to sites present in the ancestry path label dataset, and ran PLINK's (v1.90b4.4)[74] clump method (parameters: --clump-p1 5e-8 --clump-r2 0.05 --clump-kb 250; as in ref. 75) using LD in the GBR, FIN and TSI 1000 Genomes populations[68] to extract genome-wide-significant independent SNPs.

In the main text, we report results for the first set of SNPs ('fine-mapped') for analyses involving local ancestry in modern data and the second set of SNPs ('pruned') for analyses involving polygenic measures of selection (CLUES and PALM).

## Anomaly score: regions of unusual ancestry

To assess which regions of ancestry were unusual, we converted the ancestry estimates to $Z$ scores by standardizing to the genome-wide mean and standard deviation. Specifically, let $A(i,j,k)$ denote the probability of the $k$th ancestry ($k = 1, ..., K$) at the $j$th SNP ($j = 1, ..., J$) of a chromosome for the $i$th individual ($i = 1, ..., N$). We first computed the mean painting for each SNP, $A(j,k) = \frac{1}{N} \sum_{i=1}^{N} A(i,j,k)$. From this, we estimated a location parameter $\mu_k$ and a scale parameter $\sigma_k$ using a block-median approach. Specifically, we partitioned the genome into 0.5-Mb regions and, within each, computed the mean and standard deviation of the ancestry. The parameter estimates were the median values over the whole genome. We then computed an anomaly score for each SNP for each ancestry $Z(j,k) = (A(j,k) - \mu_k)/\sigma_k$. This is the normal-distribution approximation to the Poisson binomial score for excess ancestry, for which a detailed simulation study is presented in ref. 76.

To arrive at an anomaly score for each SNP aggregated over all ancestries, we also had to account for correlations in the ancestry paintings. Instead of scaling each ancestry deviation $A^*(j,k) = A(j,k) - \mu_k$ by its standard deviation, we instead 'whitened' them, that is, rotated the data to have an independent signal. Let $C = A^{*T}A^*$ be a $K \times K$ covariance matrix, and let $C^{-1} = UDV^T$ be a singular value decomposition. Then, $W = UD^{\frac{1}{2}}$ is the whitening matrix from which $Z = A^*W$ is normally distributed with covariance matrix diag(1) under the null hypothesis that $A^*$ is normally distributed with mean 0 and unknown covariance $\Sigma$. The ancestry anomaly score test statistic for each SNP is $t(j) = \sum_{k=1}^{K} Z(j,k)^2$, which is chi-squared distributed with $K$ degrees of freedom under the null, and we report $P$ values from this.

To test for gene enrichment, we formed a list of all SNPs reaching genome-wide significance ($P < 5 \times 10^{-8}$) and, using the R package gprofiler2 (ref. 77), converted these to a list of unique genes. We then used $gost$ to perform an enrichment test for each Gene Ontology (GO) term, for which we used default $P$-value correction via the g:Profiler SCS method. This is an empirical correction based on performing random lookups of the same number of genes under the null, to control the error rate and ensure that 95% of reported categories (at $P = 0.05$) are correct.

## Allele frequency over time

To investigate how effect allele frequencies have changed over time, we extracted high-effect alleles for each phenotype from the ancient data. We excluded all non-Eurasian samples, grouped them by 'groupLabel',

excluded any group with fewer than four samples and coloured points by ancestry proportion according to genome-wide NNLS based on chromosome painting (described above).

## Weighted average prevalence

To understand whether risk-conferring haplotypes evolved in the steppe population or in a pre- or post-dating population, we developed a statistic that could account for the origin of risk to be identified with multiple ancestry groups, which do not have to be the same set for each SNP.

We first applied $k$-means clustering to the dosage of each ancestry for each associated SNP and investigated the dosage distribution of clusters with significantly higher MS prevalence. For the target SNPs, the elbow method[78] suggested selecting around 5–7 clusters, and we chose 6 clusters. After performing the $k$-means cluster analysis, we calculated the average probability for each ancestry for case individuals. Furthermore, we calculated the prevalence of MS in each cluster and performed a one-sample $t$ test to investigate whether it differed from the overall MS prevalence (0.487%). This tested whether any particular combinations of ancestry were associated with the phenotype at a SNP. Clusters with high MS risk ratios had a high proportion of steppe components (Supplementary Fig. 7), leading to the conclusion that steppe ancestry alone is driving this signal.

We can then compute the WAP, which summarizes these results into the ancestries. For the $j$th SNP, let $P_{jkm} = n_{jm}\overline{P}_{jkm}$ denote the sum of the $k$th ancestry probabilities of all the individuals in the $m$th cluster ($k,m = 1, ..., 6$), where $n_{jm}$ is the cluster size of the $m$th cluster. Letting $\pi_{jm}$ denote the prevalence of MS in the $m$th cluster, the WAP for the $k$th ancestry is defined as

$$\overline{\pi}_{jk} = \frac{P_{jkm}\pi_{jm}}{\sum_{m=1}^{6} P_{jkm}},$$

where $P_{jkm}$ is defined as the weight for each cluster.

The standard deviation of $\overline{\pi}_{jk}$ is computed as s.d.$(\overline{\pi}_{jk}) = \sqrt{\sum_{m=1}^{6} w_{jkm}^2 \sigma_m^2}$, where $w_{jkm} = \frac{P_{jkm}}{\sum_{m=1}^{6} P_{jkm}}$, $\sigma_m = \frac{s(y_{jm})}{\sqrt{n_{jm}}}$ and $s(y_{jm})$ is the standard deviation of the outcome for the individuals in the $m$th cluster. We also tested the hypothesis $H_0 : \overline{\pi}_{jk} = \overline{\pi}$ against $H_1 : \overline{\pi}_{jk} \neq \overline{\pi}$ and computed the $P$ value as $p_{jk} = 2\left(1 - \phi\left(\frac{|\overline{\pi} - \overline{\pi}_{jk}|}{\text{s.d.}(\overline{\pi}_{jk})}\right)\right)$.

For each ancestry, WAP measures the association of that ancestry with MS risk across all clusters. To make a clear comparison, we calculated the risk ratio (compared to the overall MS prevalence) for each ancestry at each SNP and assigned a mean and confidence interval for the risk ratio of each ancestry on each chromosome (Fig. 3 and Extended Data Fig. 7).

## PCA and UMAP of WAP and average dosage

To sort risk-associated SNPs into ancestry patterns according to that risk, we performed PCA on the average ancestry probability and WAP at each MS-associated SNP (Supplementary Fig. 8). The former showed that all of the HLA SNPs except three from the HLA class II and III regions had much larger outgroup components than the other SNPs. The latter analysis indicated a strong association between steppe ancestry and MS risk. Additionally, outgroup ancestry at rs10914539 on chromosome 1 exceptionally reduced the incidence of MS, whereas outgroup ancestry at rs771767 (chromosome 3) and rs137956 (chromosome 22) significantly boosted MS risk.

## Ancestral risk score

To assign risk to ancient ancestries by computing the equivalent of a polygenic score for each, we followed methods developed in ref. 11. We calculated the effect allele painting frequency for a given ancestry $F_{\{\text{anc},i\}}$ for SNP $i$ using the formula:

$$f_{\{\text{anc},i\}} = \frac{\sum_{j}^{M_{\text{effect}}} \text{paintingcertainty}_{\{j,i,\text{anc}\}}}{\sum_{j}^{M_{\text{alt}}} \text{paintingcertainty}_{\{j,i,\text{anc}\}} + \sum_{j}^{M_{\text{effect}}} \text{paintingcertainty}_{\{j,i,\text{anc}\}}},$$

where there are $M_{\text{effect}}$ individuals homozygous for the effect allele, $M_{\text{alt}}$ individuals homozygous for the other allele and $\sum_{j}^{M_{\text{effect}}}$ paintingcertainty$_{\{j,i,\text{anc}\}}$ is the sum of the painting probabilities for that ancestry in individuals homozygous for the effect allele at SNP $i$. This can be interpreted as an estimate of an ancestral contribution to effect allele frequency in a modern population. The per-SNP painting frequencies can be found in Supplementary Tables 4–6.

To calculate the ARS, we summed over all $I$ pruned SNPs in an additive model:

$$\text{ARS}_{\text{anc}} = \sum_{i}^{I} f_{\{\text{anc},i\}} \times \beta_i.$$

We then ran a transformation step as in ref. 79, centring results around the ancestral mean (that is, all ancestries) and reporting as a $Z$ score. To obtain 95% confidence intervals, we ran an accelerated bootstrap over loci, which accounts for the skew of data to better estimate confidence intervals[80].

## GWAS of ancestry and genotypes

The total variance of a trait explained by genotypes (SNP values), ancestry and haplotypes (described below) is a measure of how well each captures the causal factors driving that trait. We therefore computed the variance explained for each data type in a 'head-to-head' comparison at either specific SNPs or SNP sets. In this section, we describe the model and covariates accounted for.

We used the UK Biobank to fit GWAS models for local ancestry values and genotype values separately, using only SNPs known to be associated with the phenotype (fine-mapped SNPs). We used the following phenotype codes for each phenotype: MS, data field 131043; RA, data field 131849 (seropositive).

Let $Y_i$ denote the phenotype status for the $i$th individual ($i = 1, ..., 399,998$), which takes a value of 1 for a case and 0 for a control, and let $\pi_i = \text{Pr}(Y_i = 1)$ denote the probability that this individual is a case. Let $X_{ijk}$ denote the $k$th ancestry probability ($k = 1, ..., K$) for the $j$th SNP ($j = 1, ..., 205$) of the $i$th individual. $C_{ic}$ is the $c$th predictor ($c = 1, ..., N_c$) for the $i$th individual. We used the following logistic regression model for GWAS, which assumes the effects of alleles are additive:

$$Y_i \sim \text{Binomial}(1, \pi_i) ; \log\left(\frac{\pi_i}{1 - \pi_i}\right) = \sum_{k=1}^{K} \beta_{jk} X_{ijk} + \sum_{c=1}^{N_c} \gamma_c C_{ic}.$$

We used $N_c = 20$ predictors in the GWAS models, including sex, age and the first 18 principal components, which are sufficient to capture most of the population structure in the UK Biobank[81].

First, we built the model with $K = 1$. By using only one ancestry probability in each model, we aimed to find the statistical significance of each SNP under each ancestry. We then built the model with $K = 5$, that is, using all six local ancestry probabilities, which sum to 1. We calculated the variance explained by each SNP by summing the variance explained by $X_{ijk}$ ($k = 1, ..., 5$).

We considered fitting multivariate models by using all the SNPs as covariates. However, the dataset contains only 1,982 cases. Even when only one ancestry is included, the multivariate model has 191 predictors, which could result in overfitting problems. Therefore, the GWAS models were preferred to multivariate models.

We also fitted a logistic regression model for GWAS using the genotype data as follows:

$$Y_i \sim \text{Binomial}(1, \pi_i) ; \log\left(\frac{\pi_i}{1 - \pi_i}\right) = \beta_j X_{ij} + \sum_{c=1}^{N_c} \gamma_c C_{ic},$$

where $X_{ij} \in \{0,1,2\}$ denotes the number of copies of the reference allele of the $j$th SNP ($j = 1, ..., 205$) that the $i$th individual has and $C_{ic}$ ($c = 1, ..., N_c$) denotes the covariates, including age, sex and the first 18 principal components, for the $i$th individual, where $N_c = 20$. Because the UK Biobank is underpowered compared to the case–control study in which these SNPs were found, the only statistically significant ($P < 10^{-5}$) association was for the HLA class II SNP tagging HLA-DRB1*15:01.

## GWAS comparison for trait-associated SNPs

In this section, we describe how we moved from associations between SNPs (either genotype values or ancestry) and a trait to total variance explained.

We compared the variance explained by SNPs from the GWAS model using the painting data (all six local ancestry probabilities; the seventh was a linear combination of the first six) with that from the GWAS model using the genotype data. McFadden's pseudo-$R^2$ measure[82] is widely used for estimating the variance explained by logistic regression models. McFadden's pseudo-$R^2$ is defined as

$$R^2 = 1 - \frac{\ln(L_M)}{lm(L_0)},$$

where $L_M$ and $L_0$ are the likelihoods for the fitted and null model, respectively. Taking overfitting into account, we use the adjusted McFadden's pseudo-$R^2$ value by penalizing the number of predictors:

$$\text{Adjusted } R^2 = 1 - \frac{\frac{\ln(L_M)}{N-k}}{\frac{\ln(L_0)}{N-1}},$$

where $N$ is the sample size and $k$ is the number of predictors.

Specifically, $R^2$(SNPs) is calculated as the extra variance in addition to sex, age and the 18 principal components that can be explained by SNPs:

$$R^2(\text{SNPs}) = R^2(\text{sex} + \text{age} + 18\text{PCs} + \text{SNPs}) - R^2(\text{sex} + \text{age} + 18\text{PCs}).$$

Notably, two SNPs stood out for explaining much more variance than the others when fitting the GWAS model using the genotype data, but overall more SNPs from GWAS painting explained more than 0.1% of the variance, which indicates that the painting data are probably more efficient for estimating the effect sizes of SNPs and detecting significant SNPs. Additionally, some SNPs from GWAS models using painting data explained almost the same amount of variance, suggesting that these SNPs consist of very similar ancestries.

## HTRX

Ancestry is a strong predictor of MS, but we wanted to understand whether it was tagging some causal factor that was not in our genetic data or whether it was tagging either interactions or rare SNPs. To address this, we propose HTRX, which searches for haplotype patterns that include single SNPs and non-contiguous haplotypes. HTRX is an association between a template of $n$ SNPs and a phenotype. The template gives a value for each SNP, with values of 0 or 1 reflecting that the reference allele of the SNP is present or absent, respectively, while an 'X' means that either value is allowed. For example, haplotype 1X0 corresponds to a three-SNP haplotype in which the first SNP is the alternative allele and the third SNP is the reference allele, while the second SNP can be either the reference or alternative allele. Therefore, haplotype 1X0 is essentially only a two-SNP haplotype.

To examine the association between a haplotype and a binary phenotype, we replace the genotype term with a haplotype in the standard GWAS model:

$$Y_i \sim \text{Binomial}(1, \pi_i); \log\left(\frac{\pi_i}{1-\pi_i}\right) = \beta_j H_{ij} + \sum_{c=1}^{N_c} \gamma_c C_{ic},$$

where $H_{ij}$ denotes the $j$th haplotype probability for the $i$th individual:

$$H_{ij} = \begin{cases} 1, & \text{if } i\text{th individual has haplotype } j \text{ in both genomes,} \\ \frac{1}{2}, & \text{if } i\text{th individual has haplotype } j \text{ in one of the two genomes,} \\ 0, & \text{otherwise.} \end{cases}$$

HTRX can identify gene–gene interactions and is superior to HTR not only because it can extract combinations of significant SNPs within a region, leading to improved predictive performance, but also because the haplotypes are more interpretable as multi-SNP haplotypes are only reported when they lead to increased predictive performance.

## HTRX model selection procedure for shorter haplotypes

Fitting HTRX models directly on the whole dataset can lead to significant overfitting, especially as the number of SNPs increases. When overfitting occurs, the models experience poorer predictive accuracy against unseen data. Further, HTRX introduces an enormous model space, which must be searched.

To address these problems, we implemented a two-step procedure:

Step 1: Select candidate models. This step aims to address the model search problem by obtaining a set of models more diverse than those obtained with traditional bootstrap resampling[83].

(1) Randomly sample a subset (50%) of data. Specifically, when the outcome is binary, stratified sampling is used to ensure the subset has approximately the same proportion of cases and controls as the whole dataset.

(2) Start from a model with fixed covariates (18 principal components, sex and age) and perform forward regression on the subset, that is, iteratively choose a feature (in addition to the fixed covariates) to add whose inclusion enables the model to explain the largest variance, and select $s$ models with the lowest Bayesian information criterion (BIC)[84] to enter the candidate model pool.

(3) Repeat (1) and (2) $B$ times and select all the different models in the candidate model pool as the candidate models.

Step 2: Select the best model using tenfold cross-validation.

(1) Randomly split the whole dataset into ten groups with approximately equal size, using stratified sampling when the outcome is binary.

(2) In each of the ten folds, use a different group as the test dataset and take the remaining groups as the training dataset. Then, fit all the candidate models on the training dataset and use these fitted models to compute the additional variance explained by features (out-of-sample $R^2$) in the test dataset. Finally, select the candidate model with the highest average out-of-sample $R^2$ as the best model.

## HTRX model selection procedure for longer haplotypes (cumulative HTRX)

Longer haplotypes are important for discovering interactions. However, there are $3^k - 1$ haplotypes in HTRX if the region contains $k$ SNPs, making this unrealistic for regions with large numbers of SNPs. To address this issue, we propose cumulative HTRX to control the number of haplotypes, which is also a two-step procedure.

Step 1: Extend haplotypes and select candidate models.

(1) Randomly sample a subset (50%) of data, using stratified sampling when the outcome is binary. This subset is used for all the analysis in (2) and (3).

(2) Start with $L$ randomly chosen SNPs from the entire $k$ SNPs and keep the top $M$ haplotypes that are chosen from forward regression. Then, add another SNP to the $M$ haplotypes to create $3M + 2$ haplotypes. There are $3M$ haplotypes obtained by adding 0, 1 or X to the previous $M$ haplotypes, as well as two bases of the added SNP, that is, 'XX...X0' and 'XX...X1' (as X was implicitly used in the previous step).

The top $M$ haplotypes are then selected using forward regression. Repeat this process until $M$ haplotypes are obtained that include $k-1$ SNPs.

(3) Add the last SNP to create $3M+2$ haplotypes. Afterwards, start from a model with fixed covariates (18 principal components, sex and age), perform forward regression on the training set and select $s$ models with the lowest BIC to enter the candidate model pool.

(4) Repeat (1)–(3) $B$ times and select all the different models in the candidate model pool as the candidate models.

Step 2: Select the best model using tenfold cross-validation, as described in 'HTRX model selection procedure for shorter haplotypes'.

We note that, because the search procedure in step 1(2) may miss some highly predictive haplotypes, cumulative HTRX acts as a lower bound on the variance explainable by HTRX.

As a model criticism, only common and highly predictive haplotypes (that is, those with the greatest adjusted $R^2$) are correctly identified, but the increased complexity of the search space of HTRX leads to haplotype subsets that are not significant on their own but are significant when interacting with other haplotype subsets being missed. This issue would be eased if we increased all the parameters $l$, $M$ and $B$ but with higher computational cost or improved the search by optimizing the order of adding SNPs. This leads to decreased certainty that the exact haplotypes proposed are 'correct' but reinforces the inference that interaction is extremely important.

## Simulation study for HTRX
To investigate how the total variance explained by HTRX compares to that from GWAS and HTR, we used a simulation study comparing

(1) linear models (denoted by 'lm') and generalized linear models with a logit link function (denoted by 'glm');
(2) models with or without actual interaction effects;
(3) models with or without rare SNPs (frequency of less than 5%);
(4) removing or retaining rare haplotypes when rare SNPs exist.

We started from creating the genotypes for four different SNPs $G_{ijq}$ (where $i=1, ..., 100,000$ denotes the index of individuals, $j=1$ (1XXX), 2 (X1XX), 3 (XX1X) and 4 (XXX1) represents the index of SNPs and $q=1,2$ for the two genomes as individuals are diploid). If no rare SNPs were included, we sampled the frequency $F_j$ of these four SNPs from 5% to 95%; otherwise, we sampled the frequency of the first two SNPs from 2% to 5% (in practice, we obtained $F_1 = 2.8\%$ and $F_2 = 3.1\%$ under our seed) while the frequency of the last two SNPs was sampled from 5% to 95%. For the $i$th individual, we sampled $G_{ijq} \sim \text{Binomial}(1,F_j)$ for the $q$th genome of the $j$th SNP and took the average value of the two genomes as the genotype for the $j$th SNP of the $i$th individual: $G_{ij} = \frac{G_{ij1} + G_{ij2}}{2}$. On the basis of the genotype data, we obtained the haplotype data for each individual, and we considered removing haplotypes rarer than 0.1% or not when rare SNPs were generated. In addition, we sampled 20 fixed covariates (including sex, age and 18 principal components) $C_{ic}$, where $c=1, ..., 20$ from UK Biobank for 100,000 individuals.

Next, we sampled the effect sizes of SNPs $\beta_{G_j}$ and covariates $\beta_{C_c}$ and normalized them by their standard deviations: $\beta_{G_i} \sim \frac{U(-1,1)}{\text{s.d.}(G_j)}$ and $\beta_{C_c} \sim \frac{U(-1,1)}{\text{s.d.}(C_c)}$ for each fixed $j$ and $c$, respectively. When an interaction existed, we created a fixed effect size for haplotype 11XX as twice the average absolute SNP effects: $\beta_{H_1} = \frac{1}{2} \sum_{j=1}^{4} |\beta_{G_j}|$ where $H_1$ refers to 11XX; otherwise, $H_1 = 0$. Note that $F_{H_1} = 0.09\%$ when rare SNPs were included.

Finally, we sampled the outcome on the basis of the outcome score (for the $i$th individual):

$$O_i = \sum_{c=1}^{20} \beta_c C_{ic} + \gamma \left( \sum_{j=1}^{4} \beta_{G_j} G_{ij} + \beta_{H_1} H_1 \right) + e_i + w,$$

where $\gamma$ is a scale factor for the effect sizes of SNPs and haplotype 11XX, $e_i \sim N(0,0.1)$ is the random error and $w$ is a fixed intercept term.

For linear models, outcome $Y_i = O_i$; for generalized linear models, we sampled the outcome from the binomial distribution $Y_i \sim \text{Binomial}(1,\pi_i)$, where $\pi_i = \frac{e^{O_i}}{1 + e^{O_i}}$ is the probability that the $i$th individual is a case.

As the simulation was intended to compare the variance explained by HTRX, HTR and SNPs (GWAS) in addition to fixed covariates, we tripled the effect sizes of SNPs and haplotype 11XX (if an interaction existed) by setting $\gamma = 3$. In 'glm', to ensure a reasonable case prevalence (for example, below 5%), we set $w = -7$, which was also applied in 'lm'.

We applied the procedure described in 'HTRX model selection procedure for shorter haplotypes' for HTRX, HTR and GWAS and visualized the distribution of the out-of-sample $R^2$ for each of the best models selected by each method in Supplementary Fig. 11. In both 'lm' and 'glm', HTRX had equal predictive performance to the true model. It performed as well as GWAS when interaction effects were absent, explained more variance when an interaction was present and was significantly more explanatory than HTR. When rare SNPs are included, the only effective interaction term is rare. In this case, the difference between GWAS and HTRX became smaller, as expected, and removing the rare haplotypes minimally reduced the performance of HTRX.

In conclusion, we demonstrated through simulation that our HTRX implementation (1) searches the haplotype space effectively and (2) protects against overfitting. This makes it a superior approach compared with HTR and GWAS to integrate SNP effects with gene–gene interactions. Its robustness is also retained when there are rare effective SNPs and haplotypes.

## Quantifying selection using historical allele frequencies from pathway painting
The historical trajectory of SNP frequencies is a strong signal of selection when ancient DNA data are available. This is the main purpose of our pathway painting method and can be used to infer selection at individual loci and combined into a polygenic score by analysing sets of SNPs associated with a trait.

First, we inferred allele frequency trajectories and selection coefficients for a set of LD-pruned genome-wide-significant trait-associated variants using a modified version of CLUES (Coalescent Likelihood Under Effects of Selection)[19]. To account for population structure in our samples, we applied a new chromosome painting technique based on inference of a sample's nearest neighbours in the marginal trees of an ARG that contains labelled individuals[11]. We ran CLUES using a time series of imputed ancient DNA genotype probabilities obtained from 1,015 ancient western Eurasian samples that passed all quality-control filters. We produced four additional models for each trait-associated variant by conditioning the analysis on one of the four ancestral path labels from our chromosome painting model: WHG, EHG, CHG or ANA.

Second, we were able to infer polygenic selection gradients ($\omega$) and $P$ values for each trait, that is, for MS and RA, in all ancestral paths, using PALM (Polygenic Adaptation Likelihood Method)[20]. Full methods and results can be found in Supplementary Note 6.

## LDA and LDA score
In population genetics, LD is defined as the non-random association of alleles at different loci in a given population[85]. Just like the values of the genotype, ancestries can be correlated along the genome, and, further, deviation from the expected length distribution for a particular ancestry is a signal of selection, dated by the affected ancestry. We propose an ancestry linkage disequilibrium (LDA) approach to measure the association of ancestries between SNPs and an LDA score to quantify deviations from the null hypothesis that ancestry is inherited at random across loci.

LDA is defined in terms of local ancestry. Let $A(i,j,k)$ denote the probability of the $k$th ancestry ($k=1, ..., K$) at the $j$th SNP ($j=1, ..., J$) of a chromosome for the $i$th individual ($i=1, ..., N$).

We define the distance between SNPs $l$ and $m$ as the average $L_2$ norm between ancestries at those SNPs. Specifically, we compute the $L_2$ norm for the $i$th genome as

$$D_i(l, m) = \|A(i, l, \cdot) - A(i, m, \cdot)\|_2 = \sqrt{\frac{1}{K}\sum_{k=1}^{K}(A(i, l, k) - A(i, m, k))^2}.$$

We then compute the distance between SNPs $l$ and $m$ by averaging $D_i(l, m)$:

$$D(l, m) = \frac{1}{N}\sum_{i=1}^{N} D_i(l, m).$$

We define $D^*(l, m)$ as the theoretical distance between SNPs $l$ and $m$ if there is no LDA between them. $D^*(l, m)$ is estimated by

$$D^*(l, m) \approx \frac{1}{N}\sum_{i=1}^{N} \|A(i^*, l, \cdot) - A(i, m, \cdot)\|_2,$$

where $i^* \in \{1, ..., N\}$ is resampled without replacement at SNP $l$. Using the empirical distribution of ancestry probabilities accounts for variability in both the average ancestry and its distribution across SNPs. Ancestry assignment can be very precise in regions of the genome where the reference panel matches the data and uncertain in others where only distant relatives of the underlying populations are available.

The LDA between SNPs $l$ and $m$ is a similarity, defined in terms of the negative distance $-D(l, m)$ normalized by the expected value $D^*(l, m)$ under no LD, expressed as

$$\mathrm{LDA}(l, m) = \frac{D^*(l, m) - D(l, m)}{D^*(l, m)}.$$

LDA therefore takes an expected value of 0 when haplotypes are randomly assigned at different SNPs and positive values when the ancestries of the haplotypes are correlated.

LDA is a pairwise quantity. To arrive at a per-SNP property, we define the LDA score of SNP $j$ as the total LDA of this SNP with the rest of the genome, that is, the integral of the LDA for that SNP. Because this quantity decreases to zero as we move away from the target SNP, this is in practice computed within a window of $X$ cM (we use $X = 5$ as LDA is approximately zero outside this region in our data) on both sides of the SNP. Note that we measure this quantity in terms of the genetic distance, and therefore the LDA score is measuring the length of ancestry-specific haplotypes compared to individual-level recombination rates.

As a technical note, when SNPs are present near either end of the chromosome, they no longer have a complete window, which results in a smaller LDA score. This would be appropriate for measuring total ancestry correlations, but to make LDA score useful for detecting anomalous SNPs we use the LDA score of the symmetric side of the SNP to estimate the LDA score within the non-existent window.

$$\mathrm{LDAS}(j; X) = \begin{cases} \int_{\mathrm{gd}(j)-X}^{\mathrm{gd}(j)+X} \mathrm{LDA}(j, l)\, d\mathrm{gd}, & \text{if } X \le \mathrm{gd}(j) \le \mathrm{tg} - X, \\ \int_{0}^{\mathrm{gd}(j)+X} \mathrm{LDA}(j, l)\, d\mathrm{gd} + \int_{2\mathrm{gd}(j)}^{\mathrm{gd}(j)+X} \mathrm{LDA}(j, l)\, d\mathrm{gd}, & \text{if } \mathrm{gd}(j) < X, \\ \int_{\mathrm{gd}(j)-X}^{\mathrm{tg}} \mathrm{LDA}(j, l)\, d\mathrm{gd} + \int_{\mathrm{gd}(j)-X}^{2\mathrm{gd}(j)-\mathrm{tg}} \mathrm{LDA}(j, l)\, d\mathrm{gd}, & \text{if } \mathrm{gd}(j) > \mathrm{tg} - X. \end{cases}$$

where $\mathrm{gd}(l)$ is the genetic distance (that is, position in cM) of SNP $l$ and tg is the total genetic distance of a chromosome. We also assume the LDA on either end of the chromosome equals the LDA of the SNP closest to the end: $\mathrm{LDA}(j, \mathrm{gd} = 0) = \mathrm{LDA}(j, l_{\min(\mathrm{gd})})$ and $\mathrm{LDA}(j, \mathrm{gd} = \mathrm{td}) = \mathrm{LDA}(j, l_{\max(\mathrm{gd})})$, where gd is the genetic distance and $l_{\min(\mathrm{gd})}$ and $l_{\max(\mathrm{gd})}$ are the indexes of the SNP with the smallest and largest genetic distance, respectively.

The integral $\int_{\mathrm{gd}(j)-X}^{\mathrm{gd}(j)+X} \mathrm{LDA}(j, l)\, d\mathrm{gd}$ is computed assuming linear interpolation of the LDA score between adjacent SNPs.

LDA thus quantifies the correlations between the ancestry of two SNPs, measuring the proportion of individuals who have experienced a recombination leading to a change in ancestry, relative to the genome-wide baseline. LDA score is the total amount of genome in LDA with each SNP (measured in recombination map distance).

## Simulation study for LDA and LDA score

For the simulation in Supplementary Fig. 46, an ancient population $P_0$ evolved for 2,200 generations before splitting into two subpopulations, $P_1$ (steppe) and $P_2$ (farmer). After evolution for 400 generations, we added mutations $m_1$ and $m_2$ at different loci in $P_1$ and $P_2$. Both added mutations were then positively selected in the following 300 generations, after which we sampled 20 individuals from each of $P_1$ and $P_2$ as reference samples. At generation 2,900, $P_1$ and $P_2$ admixed to $P_3$, in which both added mutations experienced strong positive selection for 20 generations. Finally, we sampled 1,000 individuals from $P_3$ to compute their ancestry proportions of $P_1$ and $P_2$ using the chromosome painting technique and calculated the LDA score of the simulated chromosome positions.

We investigated balancing selection at two loci as well. The balancing selection in $P_1$ and $P_2$ ensured that the mutant allele reached around 50% frequency, while positive selection made the mutant allele become almost the only allele. In $P_3$, if $m_1$ or $m_2$ was positively selected, its frequency reached greater than 80% regardless of whether the allele experienced balancing or positive selection in $P_1$ or $P_2$, because we set strong positive selection. If $m_1$ or $m_2$ underwent balancing selection in $P_3$, its frequency slightly increased; for example, if $m_1$ underwent balancing selection in $P_1$, it had a frequency of 25% when $P_3$ was created, and the frequency reached around 37.5% after 20 generations of balancing selection in $P_3$.

As shown in Supplementary Fig. 47, positive selection in $P_3$ resulted in low LDA scores around the selected locus if this allele was not uncommon (that is, if it had a frequency of 50% (balancing selection) or 100% (positive selection) in subpopulation $P_1$ or $P_2$). Note that the balancing selection in $P_1$ or $P_2$ worked the same as 'weak positive selection', because $m_1$ and $m_2$ were rare when they first occurred and were positively selected until they reached a frequency of 50%.

We also performed simulations for selection at a single locus (Supplementary Figs. 47 and 48).

Stage 1: An ancient population $P_0$ evolved for 1,600 generations, and then we added a mutation $m_0$, which underwent balancing selection until generation 2,200, when $P_0$ split into $P_1$ and $P_2$, where the frequency of $m_0$ was around 50%.

Stage 2: We then explored different combinations of positive, balancing and negative selection of $m_0$ in $P_1$ and $P_2$. The frequency of $m_0$ reached 80%, 50% and 20% when it was positively selected, underwent balancing selection or was negatively selected, respectively, until generation 2,899, when we sampled 20 individuals each in $P_1$ and $P_2$ as the reference samples.

Stage 3: $P_1$ and $P_2$ then merged into $P_3$ in generation 2,900. In $P_3$, for each combination of selection in stage 2, we simulated positive, balancing and negative selection for $m_0$. The selection lasted for 20 generations, and we then sampled 4,000 individuals from $P_3$ as the modern population.

When $m_0$ was positively selected in at least one of $P_1$ and $P_2$ and it experienced negative selection in $P_3$, the LDA scores around the loci of $m_0$ were low. Otherwise, no abnormal LDA scores were found surrounding $m_0$.

## Reporting summary

Further information on research design is available in the Nature Portfolio Reporting Summary linked to this article.

## Data availability

All collapsed and paired-end sequence data for new samples sequenced in this study are publicly available on the European Nucleotide Archive (accession code PRJEB65098), together with trimmed sequence alignment map files, aligned using human genome build GRCh37. Previously published ancient genomic data used in this study are detailed in Supplementary Table 13 and are all already publicly available.

## Code availability

The modified version of CLUES used in this study is available from https://github.com/standard-aaron/clues (CLUES: https://doi.org/10.5281/zenodo.8228252; PALM: https://doi.org/10.5281/zenodo.8228262). The pipeline and conda environment necessary to replicate the analysis of allele frequency trajectories and polygenic selection in Supplementary Note 6 are available on GitHub at https://github.com/ekirving/ms_paper (https://doi.org/10.5281/zenodo.8228192). The code to create ancestry anomaly scores based on chromosome painting is on GitHub at https://github.com/danjlawson/ms_paper (https://doi.org/10.5281/zenodo.8232688). The code to compute LDA and LDA score is available on GitHub at https://github.com/YaolingYang/LDAandLDAscore (https://doi.org/10.5281/zenodo.8228298). The code for HTRX is on GitHub at https://github.com/YaolingYang/HTRX (https://doi.org/10.5281/zenodo.8228295). The code for ARS calculation is on GitHub at https://github.com/will-camb/ms_paper (https://doi.org/10.5281/zenodo.8228406).

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

**Acknowledgements** We extend our thanks to all the former and current staff at the Lundbeck Foundation GeoGenetics Centre and the GeoGenetics Sequencing Core and to colleagues across the many institutions detailed below. We are particularly grateful to M. Madrona, L. Hansen and J. Bitz-Thorsen for laboratory assistance; to J. Hansen, S. Mularczyk, K. Thorø Michler and E. Neerup Nielsen for their help with sampling; and to L. Olsen as project manager for the Lundbeck Foundation GeoGenetics Centre project. The Lundbeck Foundation GeoGenetics Centre is supported by grants from the Lundbeck Foundation (R302-2018-2155, R155-2013-16338), the Novo Nordisk Foundation (NNF18SA0035006), the Wellcome Trust (214300), Carlsberg Foundation (CF18-0024), the Danish National Research Foundation (DNRF94, DNRF174), the University of Copenhagen (KU2016 programme), the Rise II project 'Towards a New European Prehistory' (M16-0455) and Ferring Pharmaceuticals A/S (to E.W.). We thank UK Biobank for access to the UK Biobank genomic resource. We also thank and acknowledge the participants and investigators of the FinnGen study. We are thankful to Illumina for collaboration. E.W. thanks St John's College, Cambridge, for providing a stimulating environment of discussion and learning and the Lundbeck Foundation, the Novo Nordisk Foundation, the Wellcome Trust, the Carlsberg Foundation and the Danish National Research Foundation for financial support. R.N. acknowledges US National Institutes of Health grant R01GM138634. K.E.A., A.P.A., A.K.N.I. and L.F. thank the OAK Foundation.

**Author contributions** W.B., Y.Y., E.K.I.-P., K.E.A., G.S. and L.T.J. contributed equally to this work. A.K.N.I., D.J.L., L.F. and E.W. led the study. W.B., A.R.-M., M.E.A., L.F., R.N. and E.W. conceptualized the study. R.N., K.K., L.F. and E.W. acquired funding for research. A.R., C.G., F.D., M.L.S.J., S.B.M., B.S., L.K., I.M.H., N.W., L.V. and T.S.K. were involved in sample collection and processing. W.B., Y.Y., E.K.I.-P., A.S., A.P., S.R. and D.J.L. were involved in developing and applying methodology. W.B., Y.Y., E.K.I.-P., G.S., A.P.A., A.R., E.A.D., M.S., S.R., A.K.N.I. and D.J.L. undertook formal analyses of data. W.B., Y.Y., E.K.I.-P., K.E.A., L.T.J., A.K.N.I., L.F. and E.W. drafted the main text (W.B. led this). W.B., Y.Y., E.K.I.-P., G.S., L.T.J., E.A.D., A.S., F.D., M.L.S.J., S.B.M., B.S., L.K., I.M.H., N.W., L.V., A.K.N.I. and D.J.L. drafted the supplementary notes and materials. W.B., Y.Y., E.K.I.-P., K.E.A., L.T.J., A.P.A., K.K., R.N., A.K.N.I., D.J.L., L.F. and E.W. were involved in reviewing drafts and editing. All co-authors read, commented on and agreed the submitted manuscript.

**Competing interests** The authors declare no competing interests.

**Additional information**
**Correspondence and requests for materials** should be addressed to Astrid K. N. Iversen, Daniel J. Lawson, Lars Fugger or Eske Willerslev.

**Pathway Painting** *Methods: Local ancestry*
*Irving-Pease et al. 2022*
Comparing each modern sample to ancient samples using the Ancestral Recombination Graph, to attribute genetic inheritance to a specific ancestry pathway.

**NNLS** (Non-Negative Least Squares) Fig. 1c-d
*Hellenthal et al. 2014; Margaryan et al. 2020*
A fast method for decomposing genome-wide ancestry into a mixture for each individual.

**Anomaly Score** Fig. 2a
*Methods: Anomaly Score; Nelson et al. 2017*
Identifies genome regions in modern individuals with excess/lack of an ancestry than expected.

**WAP** (weighted average prevalence) Fig. 3a
*Methods: WAP*
Separates disease risk into ancestry groups on a per-SNP basis.

**CLUES** (Coalescent Likelihood Under Effects of Selection) Fig. S5.1
*Stern et al. 2019*
Pathway painting estimates of SNP selection from of frequency trajectories.

**PALM** (Polygenic Adaptation Likelihood Method) Fig. 5 *Stern et al. 2021*
Pathway painting estimates of polygenic (trait) selection from CLUES frequency trajectories.

**LDAS** (Linkage Disequilibrium of Ancestry Score) Fig. 6c, S6.1-4
*Methods: LDAS (Inc Simulation study)*
Ancestry-based selection statistic which provides further information on the nature of selection.

We use two genetic ancestry representations: Population Painting using all 7 ancient reference populations, and Pathway Painting using the 4 major ancestry pathways. The ancestry composition of the data is summarised using NNLS as a mixture of the 7 populations. An Ancestry Anomaly Score for for every position in the genome identifies the HLA as the joint-biggest hit and extreme cases can be visualized as a Frequency over time

To attribute MS risk per SNP to ancient populations we compute the WAP confirming Steppe Ancestry at the HLA locus as the primary source. Statistical significance was confirmed using ARS which further demonstrates a small non-HLA Steppe ancestry risk. HTRX found that genetic risk contains a considerable non-additive component.

CLUES provides frequency trajectories for each SNP along each Pathway Painting and demonstrates widespread selection for MS SNPs. Each pathway's total risk is calculated as a polygenic risk score trajectory using PALM which identifies the Steppe as a source of MS risk. Selection is balancing as shown from Pi and Fst statistics, but multiple selection types create similar patterns.

LDAS allows us to establish that selection was multi-locus and differed by population, motivating an investigation into immune response. Evidence from Literature-reported SNP/selection associations shows a striking overlap between selected MS-associated SNPs and SNPs associated with protection against infectious diseases, supporting the hypothesis that subsistence lifestyle contributed to disease risk.

Legend:
Pathway Painting
Population Painting
Raw data

**Population Painting** *Methods: Local ancestry*
*Lawson et al. 2012, Margaryan et al. 2018*
Comparing haplotypes of modern and ancient populations to all 7 ancient populations identified by clustering individuals by haplotype. At every genomic position we obtain a local painting, i.e. an estimate of the most similar ancient individual.

**Frequency over time** Fig. 2b
Empirical allele frequency relying on aDNA and archeological dating.

**ARS** (Ancestral Risk Score) Fig. 3b-c
*Methods: ARS*
Composite ancestry disease risk either genome-wide or excluding HLA, using bootstrap uncertainty estimates.

**HTRX** (Haplotype Trend Regression with eXtra Flexibility) Fig. 4, S4
*Methods: HTRX (Inc Simulation study); Yang & Lawson 2022*
Variance explained in a trait for haplotypes, summing gene-gene interaction, epistasis and effects from rare SNPs.

**Pi** (Nucleotide Diversity), **Fst** (Population Diversity) Fig. 6c-d
Traditional summary statistics to quantify positive or negative selection.

**Literature-reported SNP/selection associations** Fig. S8.1
Identifies trait selection using overlap between functional classes and the CLUES selection scan.

**Extended Data Fig. 1 | Methods map detailing datasets used, methods, and statistics.** A narrative of the evidence used is provided in the centre, with boxes on each side detailing the methods used. Boxes are coloured by the dataset used.

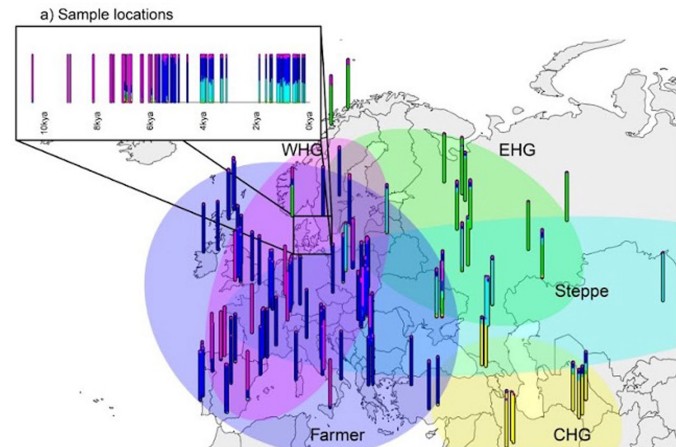

a) Sample locations

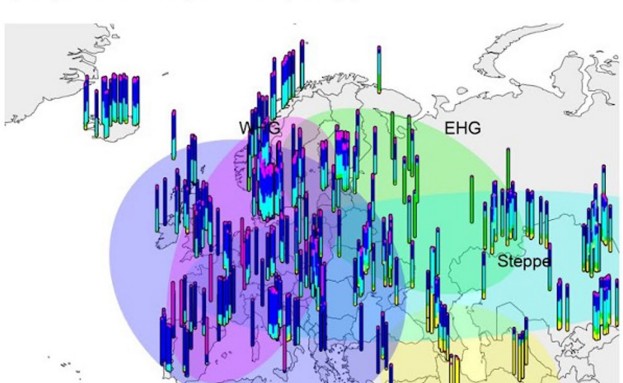

b) Sample locations for Target Individuals (<=4.2kya)

**Extended Data Fig. 2 | Ancient sample PCA, map, ancestry proportions through time for samples in Denmark.** (1) PC1 vs PC2 of the filtered Western Eurasian ancient samples included in this study. Black circled points are Danish Medieval and post-Medieval samples published here for the first time. Major component ancestry locations are labelled. (2) Map of ancient filtered Eurasian and African ancient samples included in this study. (3a) Map of reference data and time transect of Denmark as in Fig. 1. (3b) More recent ancient data (samples <4,200 years ago) not used as reference, showing the clines of the main ancestry components from (3a).

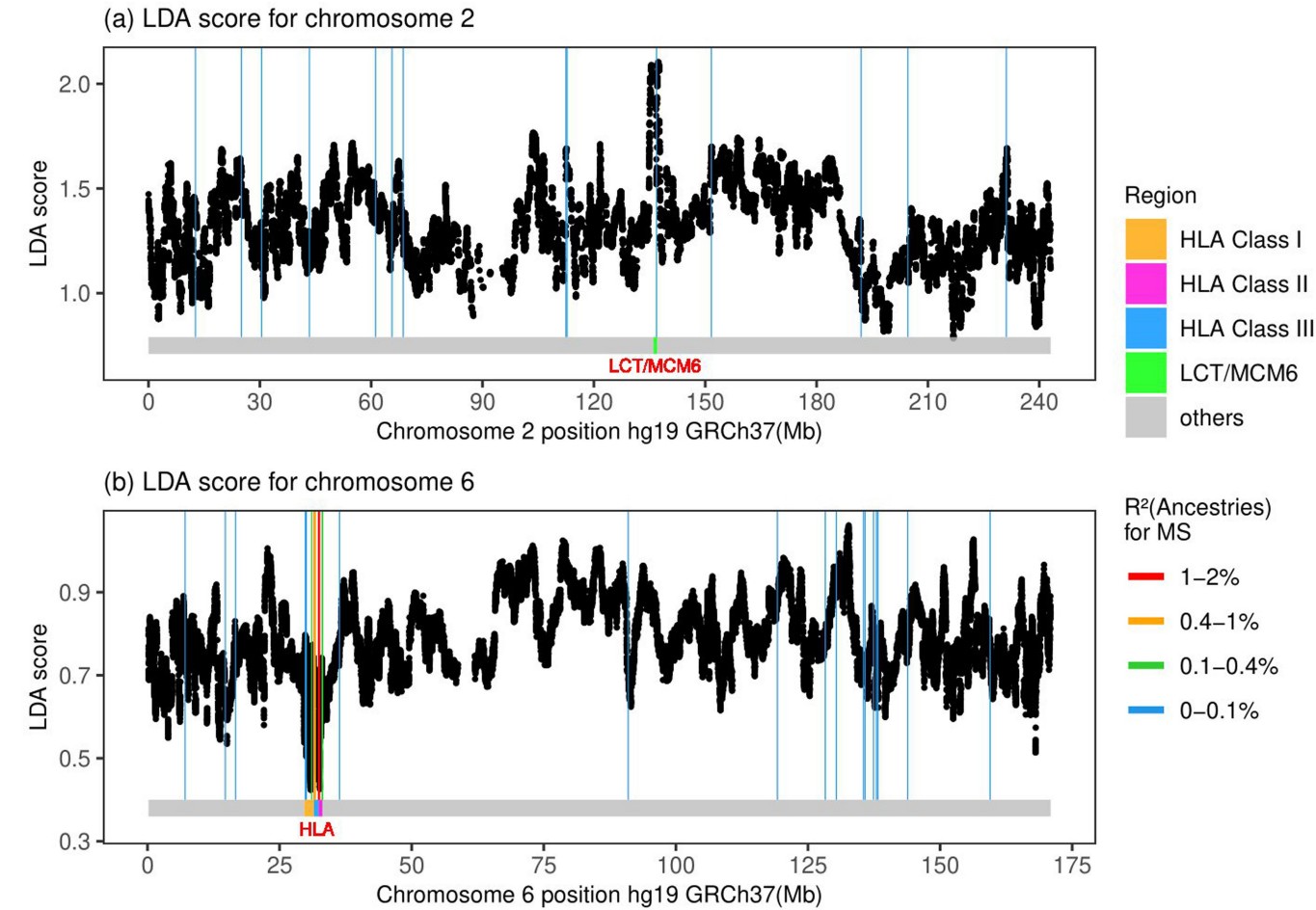

**Extended Data Fig. 3 | LDAS on chromosome 2 and 6.** LDA score is a) high in the LCT/MCM6 region while it is b) low in the HLA region.

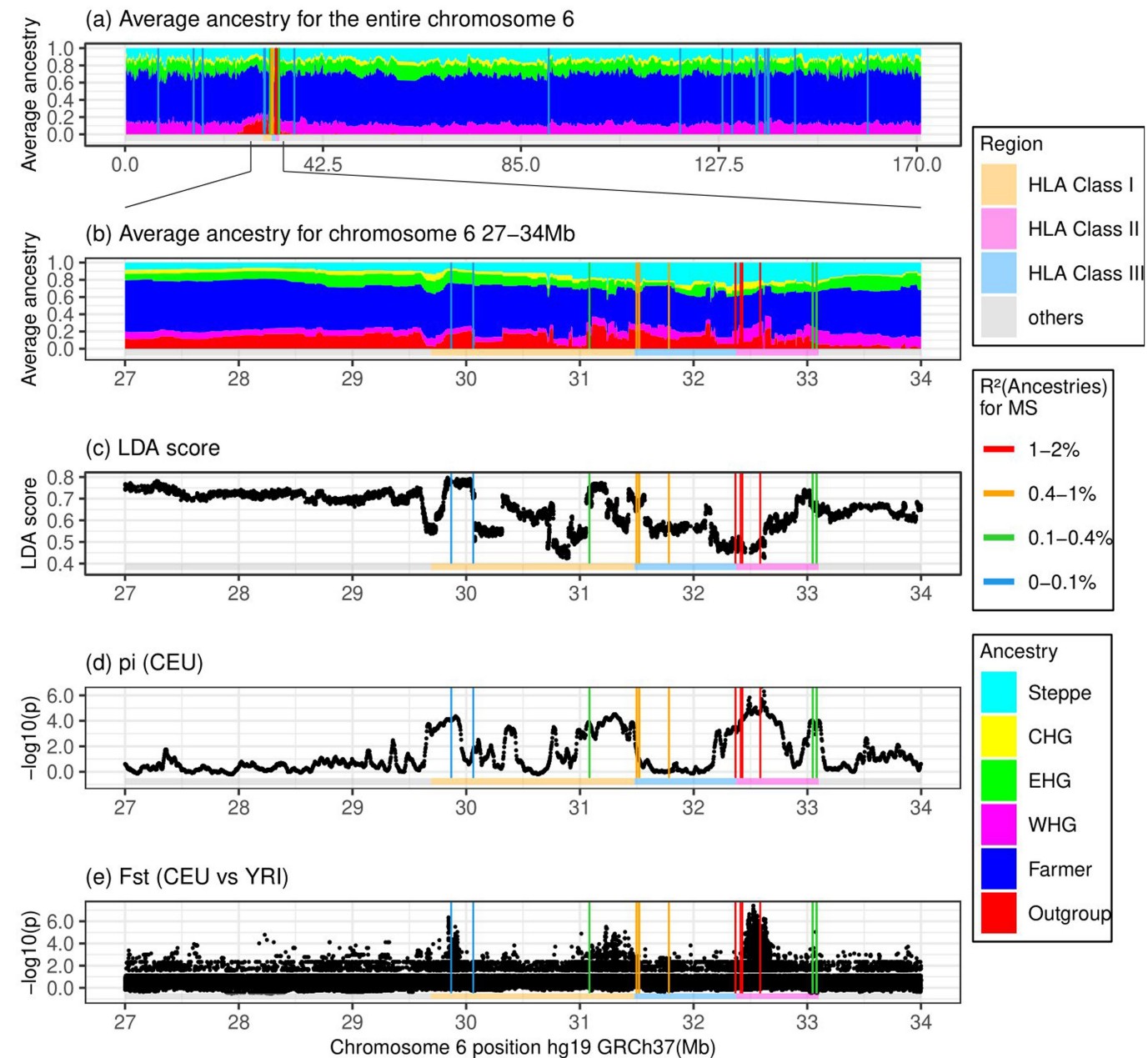

**Extended Data Fig. 4 | Signatures of selection at the HLA locus showing different regions of the HLA (horizontal coloured bar) and locations of MS-associated SNPs (vertical lines, coloured by the variance explained by 6 ancestries).** a): Whole Chromosome 6 "local ancestry" decomposition by genetic position. b). HLA "local ancestry" decomposition by genetic position. c): LDA score; low values are indicative of selection for multiple linked loci, while high values indicate positive selection. d): pi scores (nucleotide diversity) for CEU (Northern and Western European ancestry). MS-associated SNPs fall in highly diverse regions of the HLA. e): Fst scores (divergence between two populations) for CEU vs YRI(Yoruba); locally higher scores indicate regions that have undergone differential selection between the two populations.

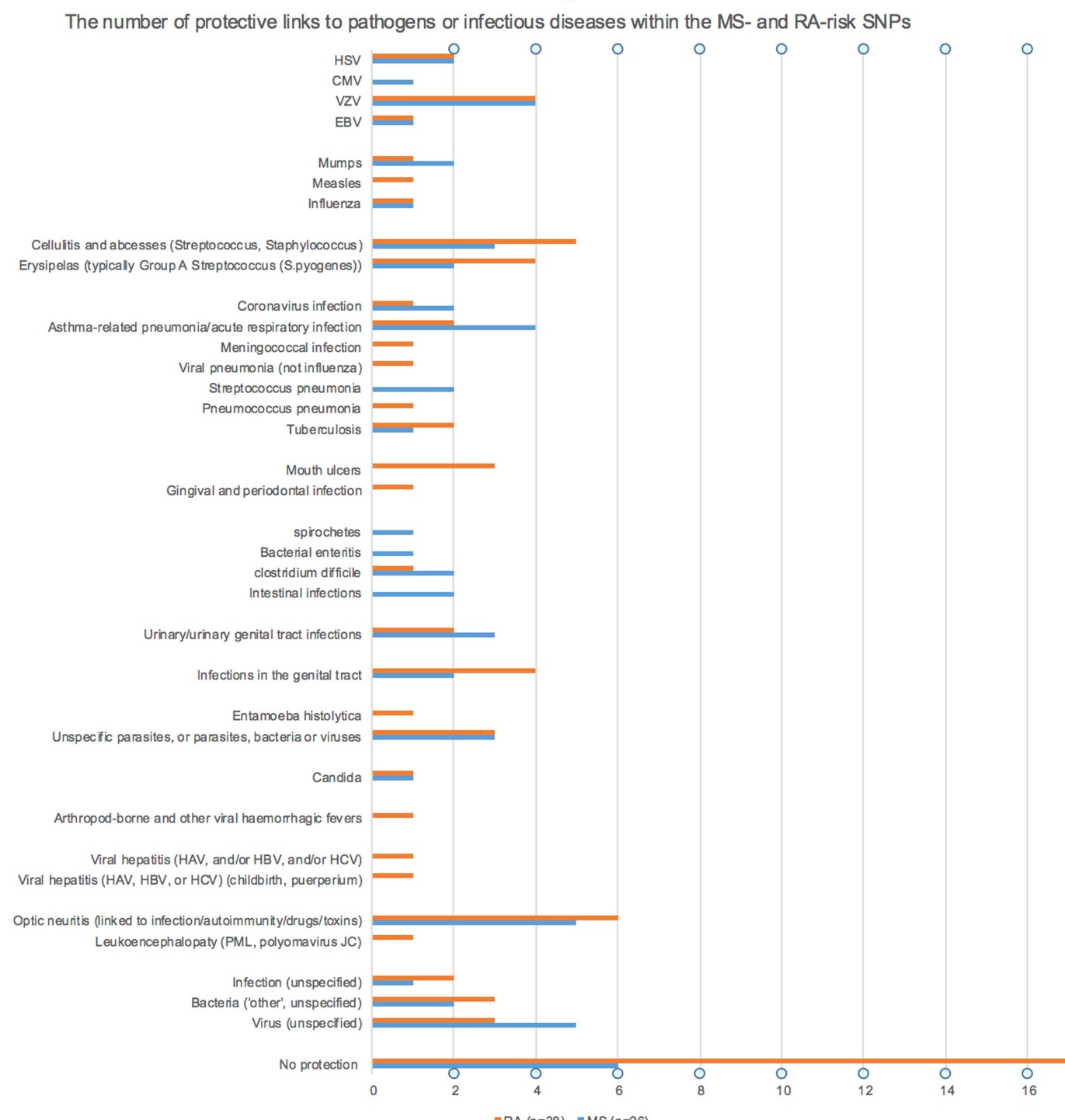

The number of protective links to pathogens or infectious diseases within the MS- and RA-risk SNPs

**Extended Data Fig. 5 | The number of protective associations with pathogens or infectious diseases for the MS- and RA-associated selected SNPs.** The number of protective associations to specific pathogens and/or diseases associated with the MS- and RA-SNPs that showed statistically significant evidence for selection using CLUES. One SNP can have a link to more than one pathogen and/or disease (see ST11 and ST12 for details on each SNP). Eight and twenty SNPs had no detectable links to any pathogen or infectious disease in the MS and RA SNP sets, respectively.

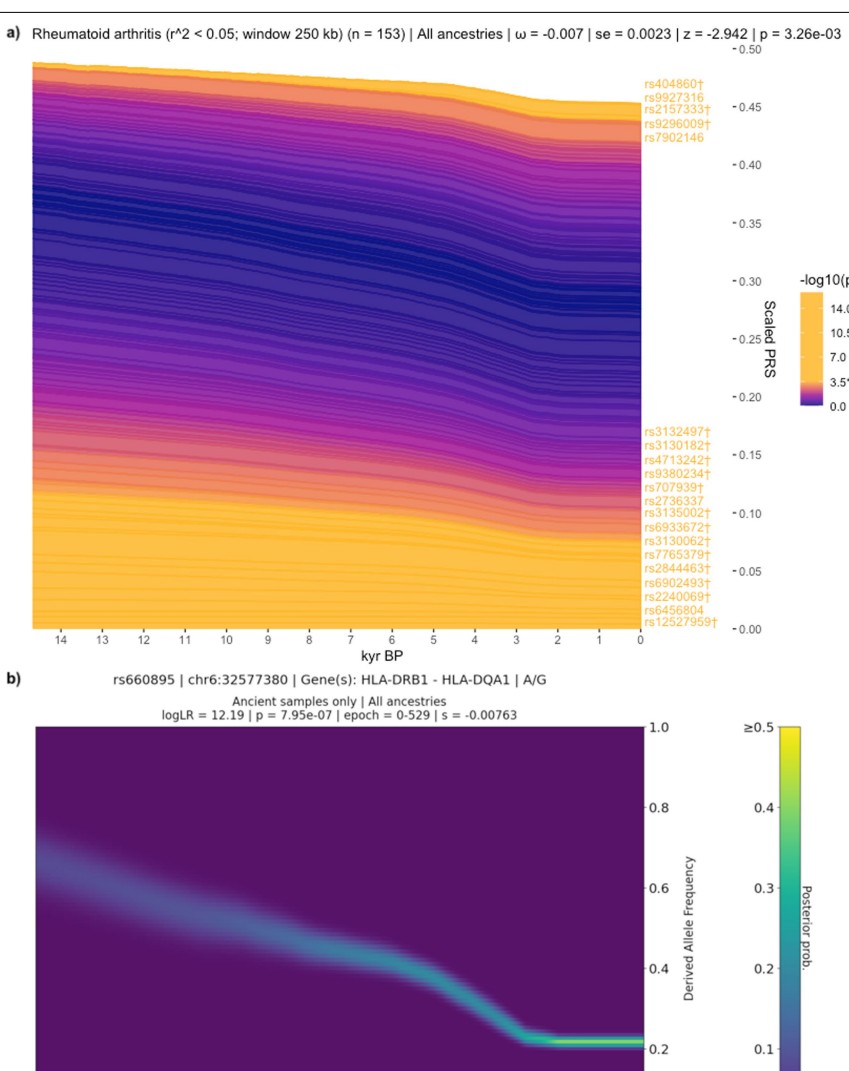

**a)** Rheumatoid arthritis (r^2 < 0.05; window 250 kb) (n = 153) | All ancestries | ω = -0.007 | se = 0.0023 | z = -2.942 | p = 3.26e-03

rs404860†
rs9927316
rs2157333†
rs9296009†
rs7902146

rs3132497†
rs3130182†
rs4713242†
rs9380234†
rs707939†
rs2736337
rs3135002†
rs6933672†
rs3130062†
rs7765379†
rs2844463†
rs6902493†
rs2240069†
rs6456804
rs12527959†

-log10(p)
14.0
10.5
7.0
3.5*
0.0

**b)** rs660895 | chr6:32577380 | Gene(s): HLA-DRB1 - HLA-DQA1 | A/G
Ancient samples only | All ancestries
logLR = 12.19 | p = 7.95e-07 | epoch = 0-529 | s = -0.00763

Posterior prob.
≥0.5

**Extended Data Fig. 6 | Evidence for selection on RA-associated SNPs.**
a) Stacked line plot of the pan-ancestry PALM analysis for RA, showing the contribution of SNPs to disease risk over time. SNPs are shown as stacked lines, the width of each line being proportional to the population frequency of the positive risk allele, weighted by its effect size. When a line widens over time the positive risk allele has increased in frequency, and vice versa. SNPs are sorted by the magnitude and direction of selection, with positively selected SNPs at the top, negatively selected SNPs at the bottom, and neutral SNPs in the middle. SNPs are coloured by their corresponding p-value in a single locus selection test. The asterisk marks the Bonferroni corrected significance threshold, and nominally significant SNPs are shown in yellow and labelled by their rsIDs. SNPs marked with the dagger symbol are located in the HLA locus. The Y-axis shows the scaled average polygenic risk score (PRS) in the population, ranging from 0 to 1, with 1 corresponding to the maximum possible average PRS (i.e. when all individuals in the population are homozygous for all positive risk alleles) and the X-axis shows time in units of thousands of years before present (kyr BP). b) Posterior likelihood trajectory for rs660895, tagging HLA-DRB1*04:01, inferred by CLUES. Statistical significance was assessed by applying a Bonferroni correction for the number of tests performed for each trait.

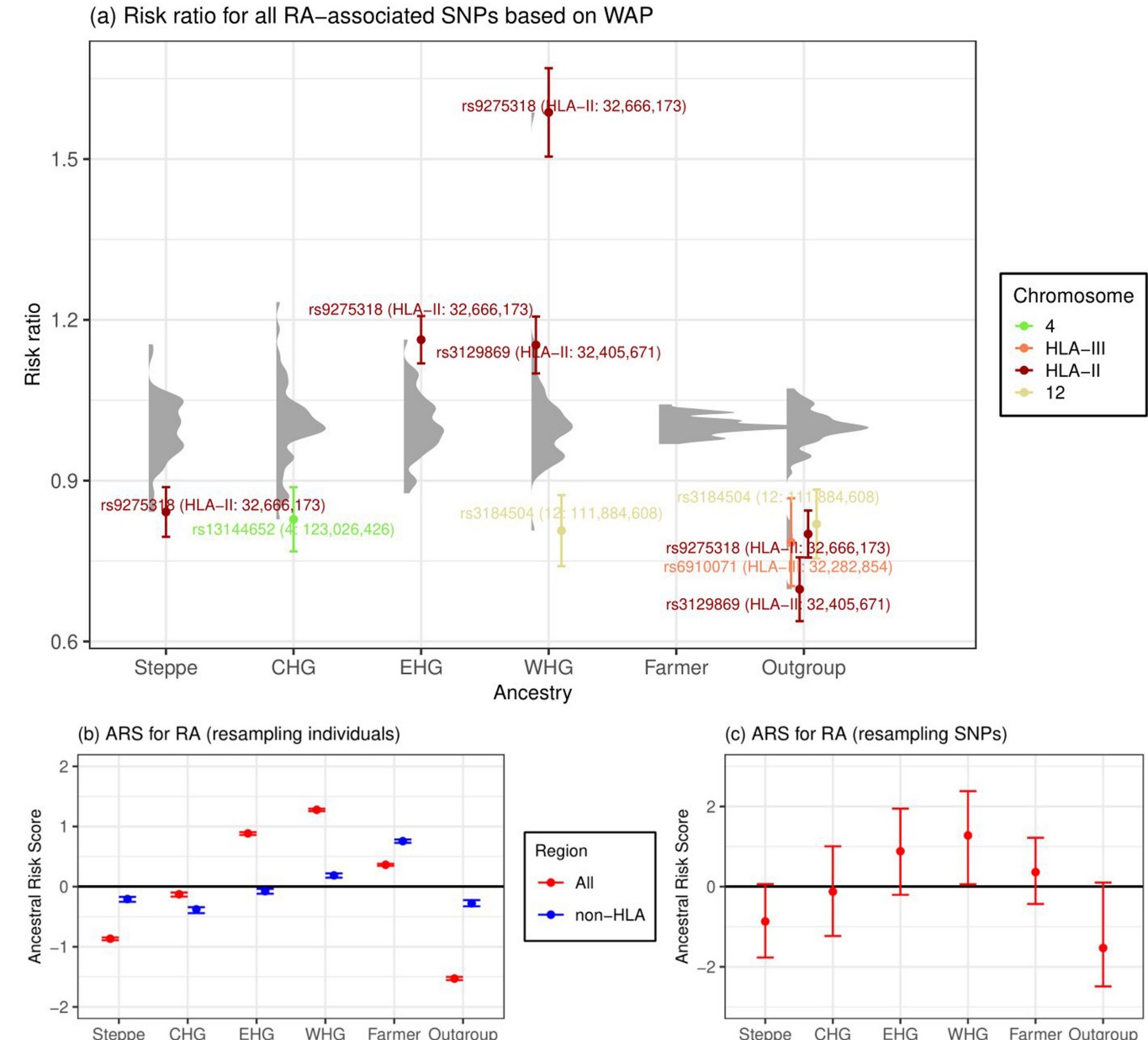

**Extended Data Fig. 7 | Associations between local ancestry at fine-mapped RA SNPs and RA in a modern population.** a) Risk ratio of SNPs for RA based on weighted average prevalence (WAP; see Methods), when decomposed by inferred ancestry. A mean and standard deviation are calculated for each ancestry based on bootstrap resampling, for each chromosome (n = 408,884 individuals). The distribution of risk ratios at each ancestry is shown as a raincloud plot. SNPs significant at the 1% level are shown individually, coloured by chromosome or HLA region, and those with risk ratio >1.1 or <0.9 are annotated with rsID, HLA region and position (build GRCh37/hg19).

b-c) Genome-wide Ancestral Risk Scores (ARS, see Methods) for RA. Mean and confidence intervals are estimated by either bootstrapping over individuals (b, which can be interpreted as testing power to reject a null hypothesis of no association between RA and ancestry; n = 1000 bootstrap resamples with replacement over 24,000 individuals) and bootstrapping over SNPs (c, which can be interpreted as testing whether ancestry is associated with RA genome-wide; n = 1000 bootstrap resamples with replacement over 55 SNPs). We show results for all associated SNPs (red) and non-HLA SNPs only (blue) when bootstrapping over individuals.

# Reporting Summary

## Statistics

For all statistical analyses, confirm that the following items are present in the figure legend, table legend, main text, or Methods section.

| n/a | Confirmed | |
|---|---|---|
| ☐ | ☒ | The exact sample size (*n*) for each experimental group/condition, given as a discrete number and unit of measurement |
| ☐ | ☒ | A statement on whether measurements were taken from distinct samples or whether the same sample was measured repeatedly |
| ☐ | ☒ | The statistical test(s) used AND whether they are one- or two-sided<br>*Only common tests should be described solely by name; describe more complex techniques in the Methods section.* |
| ☐ | ☒ | A description of all covariates tested |
| ☐ | ☒ | A description of any assumptions or corrections, such as tests of normality and adjustment for multiple comparisons |
| ☐ | ☒ | A full description of the statistical parameters including central tendency (e.g. means) or other basic estimates (e.g. regression coefficient) AND variation (e.g. standard deviation) or associated estimates of uncertainty (e.g. confidence intervals) |
| ☐ | ☒ | For null hypothesis testing, the test statistic (e.g. $F$, $t$, $r$) with confidence intervals, effect sizes, degrees of freedom and $P$ value noted<br>*Give P values as exact values whenever suitable.* |
| ☒ | ☐ | For Bayesian analysis, information on the choice of priors and Markov chain Monte Carlo settings |
| ☒ | ☐ | For hierarchical and complex designs, identification of the appropriate level for tests and full reporting of outcomes |
| ☐ | ☒ | Estimates of effect sizes (e.g. Cohen's *d*, Pearson's *r*), indicating how they were calculated |

*Our web collection on statistics for biologists contains articles on many of the points above.*

## Software and code

Policy information about availability of computer code

| Data collection | GLIMPSE<br>BCL Convert<br>AdapterRemoval (2.2.4)<br>BWA (0.7.17)<br>Picard MarkDuplicates (2.18.26)<br>mapDamage2.0<br>ContamMix<br>ANGSD (0.931) |
|---|---|

| Data analysis | READ<br>HaploGrep2<br>PLINK2<br>ADMIXTURE<br>qpAdm<br>Chromopainter<br>The modified version of CLUES used in this study is available from https://github.com/standard-aaron/clues. The pipeline and conda environment necessary to replicate the analysis of allele frequency trajectories and polygenic selection in Supplementary Note 6 are available on Github at https://github.com/ekirving/ms_paper. The code to create Ancestry Anomaly scores based on Chromosome painting is on Github at https://github.com/danjlawson/ms_paper. The code to compute LDA and LDA score is available on Github at https://github.com/YaolingYang/LDAandLDAscore. The code for HTRX is on Github at https://github.com/YaolingYang/HTRX. The code for ARS calculation is on Github at https://github.com/will-camb/ms_paper. |
|---|---|

For manuscripts utilizing custom algorithms or software that are central to the research but not yet described in published literature, software must be made available to editors and reviewers. We strongly encourage code deposition in a community repository (e.g. GitHub). See the Nature Portfolio guidelines for submitting code & software for further information.

## Data

Policy information about availability of data

All manuscripts must include a data availability statement. This statement should provide the following information, where applicable:
- Accession codes, unique identifiers, or web links for publicly available datasets
- A description of any restrictions on data availability
- For clinical datasets or third party data, please ensure that the statement adheres to our policy

All collapsed and paired-end sequence data for novel samples sequenced in this study will be made publicly available on the European Nucleotide Archive, together with trimmed sequence alignment map files, aligned using human build GRCh37. Previously published ancient genomic data used in this study are detailed in ST13, and are all already publicly available.

The UK Biobank is a public resource open to approved researchers. More information at https://www.ukbiobank.ac.uk/
The 1000 Genomes resource is publicly available. More information at https://www.internationalgenome.org/

## Research involving human participants, their data, or biological material

Policy information about studies with human participants or human data. See also policy information about sex, gender (identity/presentation), and sexual orientation and race, ethnicity and racism.

| Reporting on sex and gender | Sex was assigned based on sex chromosomes. Sex-specific results were not calculated. |
|---|---|
| Reporting on race, ethnicity, or other socially relevant groupings | Reporting was restricted to a self-identified 'white British' cohort, with PCA outliers removed (details in Bycroft et al., 2018). |
| Population characteristics | N/A |
| Recruitment | N/A |
| Ethics oversight | Use of the UK Biobank resource was approved in 2020. |

Note that full information on the approval of the study protocol must also be provided in the manuscript.

## Field-specific reporting

Please select the one below that is the best fit for your research. If you are not sure, read the appropriate sections before making your selection.

☐ Life sciences  ☐ Behavioural & social sciences  ☒ Ecological, evolutionary & environmental sciences

For a reference copy of the document with all sections, see nature.com/documents/nr-reporting-summary-flat.pdf

## Ecological, evolutionary & environmental sciences study design

All studies must disclose on these points even when the disclosure is negative.

| Study description | This study uses data from ancient and modern individuals to estimate ancestral contributions to risk for multiple sclerosis and rheumatoid arthritis, and test for signals of selection. |
|---|---|
| Research sample | Modern samples were from the UK Biobank and 1000 Genomes project. Ancient samples were previously published, and 86 new samples published here for the first time. |

| Sampling strategy | 86 samples is enough to capture much of the genetic variation present in Medieval Denmark. It also brings this population to a similar sampling size as other ancient groups. |
|---|---|
| Data collection | The ancient sampling procedure is described in the SI. Sequencing is described here:<br>SSequencing data was generated from a total of 86 Medieval samples (ST1), using semi-automated laboratory procedures. Laboratory work on aDNA was conducted in the dedicated ancient DNA clean-room facilities at the Lundbeck Foundation GeoGenetics Centre (Globe Institute, University of Copenhagen).<br><br>In brief, two parallel sub-samples of <150 mg were obtained from human skeletal material and demineralized as described earlier, using pre-digestion for 30 min (Damgaard et al., 2015). Two aDNA extractions were performed per subsample, using a 96 well format, combining 150 µl of demineralized material with 1.5 ml binding buffer (500 ml Qiagen PB, supplemented with 15 ml Sodium acetate 3M, and 1.25 ml 5M NaCl, phenol red, adjusted to pH=5) and 10 µl of paramagnetic beads (G-Bioscience, #786-915) for 15 minutes (Rohland et al., 2018). Pelleted beads were washed twice in 450 µl and 100 µl 80% ethanol + 20% 10mM Tris-HCl, respectively, and eluted in 10 mM Tris-HCl + 0.05% Tween-20. From each subsample one extract (35 µl) was incubated with 10 µl USER enzyme (NEB #M5505) for 3h at 37°. DNA shotgun sequencing libraries were prepared in 96-well format essentially as described elsewhere (Meyer and Kircher 2010), using a small (25ul) or large (50ul) total reaction volume for non-USER and USER-treated extracts, respectively, including 21.25 µl or 42.5 µl DNA template. Clean-up procedures after end-repair and adapter-ligation were performed with 10 µl of paramagnetic beads (G-bioscience) in 10 volumes of the binding buffer described above. The requirement for PCR amplification was evaluated by qPCR using 1µl of pre-amplified library. Indexing PCR, using 8-bp unique dual indexing (Illumina TruSeq UDI0001-0096) in 50 or 100 µl reaction volumes, with KAPA HiFi HotStart Uracil+ (KapaBiosystems #KR0413) according to manufacturer's recommendations, with typically 14 amplification cycles. Final purification of libraries was performed using a 1:1.6 ratio of library to HighPrep™ PCR beads (MagBio, #AC-60250). Length distribution and concentration of individual purified libraries was controlled using the Fragment Analyzer (High Sensitivity kit). Libraries were pooled equimolar before sequencing. Sequencing was performed on Illumina NovaSeq6000 at the GeoGenetics Sequencing Core, Copenhagen, using S4 200 cycles kits version 1.5. |
| Timing and spatial scale | Samples from three cemeteries in Denmark were sequenced. The urban medieval churchyard of Our Lady (Vor Frue) and associated building structures were excavated by Aalborg Historiske Museum/Nordjyske Museer between 2011 to 2013. The cemetery Ahlgade 15-17 is an urban cemetery in the center of Holbæk, north of the main street of Ahlgade and adjacent to the fjord and the harbour. It was excavated in 1985 to 1986 by Museum Vestsjælland, previously called Museet for Holbæk og Omegn. The cemetery Tjaerby was excavated by Kulturhistorisk Museum Randers in 1998 to 2010.<br><br>More details are available in the SI. |
| Data exclusions | Low coverage and related samples were excluded. |
| Reproducibility | Bootstrap resampling was used for the ARS and WAP analyses. HTRX was trained out-of-sample. |
| Randomization | No groupings used. Covariates used were age, sex, first 20 PCs for analyses involving the UK Biobank. |
| Blinding | Blinding not possible. |

Did the study involve field work? ☐ Yes ☒ No

# Reporting for specific materials, systems and methods

We require information from authors about some types of materials, experimental systems and methods used in many studies. Here, indicate whether each material, system or method listed is relevant to your study. If you are not sure if a list item applies to your research, read the appropriate section before selecting a response.

## Materials & experimental systems

| n/a | Involved in the study |
|---|---|
| ☒ | Antibodies |
| ☒ | Eukaryotic cell lines |
| ☐ | ☒ Palaeontology and archaeology |
| ☒ | Animals and other organisms |
| ☒ | Clinical data |
| ☒ | Dual use research of concern |
| ☒ | Plants |

## Methods

| n/a | Involved in the study |
|---|---|
| ☒ | ChIP-seq |
| ☒ | Flow cytometry |
| ☒ | MRI-based neuroimaging |

## Palaeontology and Archaeology

| Specimen provenance | The urban medieval churchyard of Our Lady (Vor Frue) and associated building structures were excavated by Aalborg Historiske |
|---|---|

Specimen provenance

Museum/Nordjyske Museer between 2011 to 2013. The churchyard belonged to the church and convent of Our Lady and is located in the eastern part of the medieval town of Aalborg. Approximately 900 graves were recovered of which 272 could be sampled for DNA analysis. The churchyard was excavated in connection with a large sewerage project, and only parts of the churchyard was exhumed.

The cemetery Ahlgade 15-17 is an urban cemetery in the center of Holbæk, north of the main street of Ahlgade and adjacent to the fjord and the harbour. It was excavated in 1985 to 1986 by Museum Vestsjælland, previously called Museet for Holbæk og Omegn.

The cemetery belonged to the former parish church of St. Nicolai and date from the late 12th century to 1573 when the church was abandoned. However, the cemetery is thought to have been taken out of use shortly after the reformation in 1536.

Tjaerby: Rural cemetery ca. 5 km east of Randers on the north side of Randers fjord. It was excavated by Kulturhistorisk Museum Randers in 1998 to 2010. The excavation area revealed a stone church, and a cemetery containing ca. 1200 graves from which 351 individuals were sampled for DNA analysis in this project. The cemetery dates to ca. 1050 to late 1536, but skeletal remains were only preserved from graves dating after 1200. Remnants of a farmhouse and a wooden church predating the cemetery (900-1100) were also recovered1,2. The surrounding area consisted of forest and meadows.

Specimen deposition

Specimens are with the museums described above.

Dating methods

Described in 'sample provenance'. Dating was either C-14 or by archaeological context. Dates are reported in SI.

☒ Tick this box to confirm that the raw and calibrated dates are available in the paper or in Supplementary Information.

Ethics oversight

No ethical approval required.

Note that full information on the approval of the study protocol must also be provided in the manuscript.

