## [Peer Review File · Nature]

Manuscript Title: Elevated genetic risk for multiple sclerosis originated in Steppe Pastoralist populations

Reviewer Comments & Author Rebuttals

Reviewer Reports on the Initial Version:

Referees' comments:

Referee #1 (Remarks to the Author):

This is excellent paper establishing the link between past selection in the HLA-region and increased risk of Multiple Sclerosis in present-day population. Overall, I found the work presented fascinating and that the key conclusions are well supported by the data. My comments/suggestions below are primarily aimed at improving the clarity of the manuscript, which, as it stands, it is extremely hard to follow.

The authors present a total of 5 new statistical approaches to demonstrate that Steppe ancestry is associated with risk of MS in contemporary populations but there is virtually no description of how each of these methods work in the actual manuscript. The reader needs to dig deep into the ~100 pages of supplements to get a sense of how each of these new methods works. Even then, one is left wondering how well they work given that each of these methods could be a paper in itself with associated simulations, etc to formally demonstrate that the approaches are well calibrated and lead to accurate estimates. That said, and given that all analyses point towards the same direction, I am confident that the conclusions are correct, but I am left wondering if it would not be better to focus on only a subset of the analyses (and explain those in better detail) since some of them appear to be redundant (more on that later).

Along the same lines, even the dataset itself is hardly described in the main text (they only say "Using the largest dataset from the Stone Age coupled with 86 new Medieval and post-Medieval genomes from the Denmark"). I realize that a full description of the dataset is provided on an accompanying study and in the Supplements but the actual paper should be clear on its own.

In the risk-ratio analyses (Figure 3) the authors describe how the Steppe ancestry has the highest risk ratio for MS in nearly all HLA SNPs while Farmer ancestry and "outgroup" appear to be protective. However, the "outgroup" ancestry for one SNP outside the HLA region ("rs771767") seems to show a risk ratio as large as that of HLA. Can the authors describe what this SNP is tagging?

I am not sure to understand the value of the Ancestral Risk Score (ARS) analyses over the risk ratio. If the alleles that contribute the most for MS risk are in the HLA regions, and we already know that those are enriched for Steppe ancestry aren't we just measuring the same thing in a different way? Indeed, from what I can tell, all the significant in ARS comes from the HLA region alone (i.e., the other non-HLA MS-risk alleles do not significantly contribute to the enrichment of Steppe ancestry).

The least supported claim of the paper is the link between selection on MS-risk alleles and protection against infectious diseases. The link was based on manual curation of the published literature and most likely based on genetic associations that are not statistically robust. Indeed, as the authors mentioned, most genetic associations with infectious diseases have not been based on GWAS and instead are derived from gene candidate studies. Gene candidate studies are (i) biased towards genes that are more likely to be involved in immune responses, and (ii) afflicted by high rates of false positives, making it hard to know how many of those associations are truly real. I would like to have seen some evidence that the overlaps that they found between MS-SNPs and genetic risk for infectious diseases were larger than expected by chance when focusing on risk alleles for other immune-related disorders.

After having read the paper at least 5 times, my conclusion is that the HLA-associated risk factor from MS is enriched for Steppe ancestry and it was likely positively selected in the past (maybe to protect against infectious diseases but I found that less compelling). That conclusion, however, is quite different from the claim on the title that "Genetic risk for MS originated in pastoralist steppe populations". The variance explained in MS for the HLA locus is below 3% and therefore it is not representative of "Genetic risk for MS", which extends way beyond HLA.

Referee #2 (Remarks to the Author):

In this work, Barrie and colleagues take a deep dive into investigating the ancient genetic ancestry of multiple sclerosis (MS) risk loci using the ancient genome dataset from the Stone Age they describe in an accompanying paper. They show that Steppe ancient ancestry is associated with an increased risk of MS while the farmer and outgroup African ancestries are associated with protection against the diseases. They also show that MS-associated risk variants have gone under positive selection in the ancestral Steppe population as well as descendant Europeans. They speculate that this positive selection may have been due to the protective effect of these variants against different pathogens. The authors also look into the selection pattern of rheumatoid arthritis (RA)-associated risk SNPs and observe an opposite pattern. RA-associated variants show evidence of negative selection in ancient genomes although they see that these negatively selected RA-risk variants are also associated with

protection against the same pathogens as the ones associated with positively selected MS-risk variants.

Overall, this study is interesting. It brings insight into the relatively high prevalence of MS in individuals of European ancestry and its higher prevalence in Northern vs Southern Europe. I do have a few concerns/questions about the strength of some of the findings as well as some of the conclusions. I list them below:

1- I had a hard time judging the significance of some of the findings, specifically the ones related to the HTRX and LDA methods. The authors mention both methods/metrics are developed for this paper. Neither of these methods is peer-reviewed. Reviewing these two methods in depth requires more details, examples, and comparison with existing methods as such I am not sure about the robustness of either method and the significance of the conclusion based on these two metrics.

- How "recombinant favouring selection" (line 304) is related to (or different from) heterozygous advantage and balancing selection both of which are at play in the HLA region.

2- How accurate is ancient LAI in HLA region? As authors know imputation in HLA is not accurate in the absence of high-quality population-specific reference panels (see Luo et al Nat gen, 2021) similarly LAI relies on the presence of high-quality and high-density reference panels. Given the sparsity of aDNA how accurate do authors think is their ancient LAI? Is there any way to quantify this or assign a confidence score to ancient LAI calls?

- How well does the sum of ancient LAI correlate with ancient global ancestry inference for the represented ancestries?

3- The ancient ancestry, based on what's reported here, is different between Northern and Southern Europeans. The authors also report the ancestry profile of RA and MS risk variants to be strikingly different and somewhat opposite of each other. How do authors reconcile these observations with the fact that both RA and MS are more frequent in Northern Europe?

- On a related note, I find the conclusion that infectious diseases are the driver of both the negative selection of RA-associated risk variants and positively selected MS-risk variants speculative. It is true that infectious diseases are a major selection pressure and that the risk of other immune-mediated diseases (including autoimmune diseases) is likely to be affected by past and current pathogens. However, most of the pathogens cited by the authors are/have been covering wide geographical regions encompassing Eurasia and thus I am not sure why they should have presented selective pressure in one ancestral group and not in others. Can you comment?

4- There is no mention of other similar work in the intro or discussions. Two examples that come to my mind are the recent paper about the Black death and Chron's disease risk from the Barreiro group (Nature 2022) and the one on TB risk and changes in MAF of a known TYK2 risk variant from the Quintana-Murci group (AJHG 2021). Please cite the relevant previous work.

- Related to the above point have you looked at the changes in MAF of the variants reported in the above two papers in your cohort through time (and if possible, in different ancient ancestral groups)? I would expect to see your data recapitulate the same patterns as reported before. Adding this as a sup can add to this work's robustness.

Other:

5- Line 220: Across the entire HLA region, haplotypes explain at least 17% more out-of-sample variation than GWAS (2.90%, compared to 2.48%). I am not sure where the number 17% comes from, $2.90 - 1.48 \neq 17$. The number is again cited in the discussion please elaborate.

6- Through the figures, CHG, WHG, EHG, farmer, and outgroup are always represented but results related to ANA, Steppe, and Yamnaya groups are shown in some figures only (see fig 2, 3, and 5). I am not an ancient genome expert, but I find the lack of consistency confusing in following the paper and interpreting of the results from different figures.

7- Why did the authors choose CLUES over other selection methods (EHH, composite score, etc) will the results change if another selection metric is used?

8- Line 377: by "directional selection" do you mean both positive and negative? If yes, it makes this paragraph a bit confusing given that it mentions negative selection at the start of the paragraph.

Referee #3 (Remarks to the Author):

In this manuscript by Barrie et al. the authors provide evidence that genetic risk variants for multiple sclerosis (MS) rose to higher frequency in a specific ancient population, were brought into Europe app. 5000 years ago and were subject to positive selection. Furthermore, by providing evidence of association of some of the MS risk variants to protection from different infectious diseases the authors provide a possible explanation for the observed positive selection of these MS risk alleles.

I believe that the data presented in this report will be interesting for researchers with a focus on MS genetics, as they provide evidence for a specific ancient population and time period in driving MS risk development. The results presented here could stimulate more research in the area of ancestry association to gain more insight into the mechanisms of MS risk.

A potentially very interesting result is the finding that some MS risk variants under positive selection during the investigated time period appear to be protective against different pathogens/diseases. While this is a possible explanation for the observed positive selection, I believe this could be further investigated as I describe in detail below (see comment 9).

Most of the results and (statistical) analyses were clear and conducted thoroughly so that they support the drawn conclusions. However, I have a few questions/comments listed below.

Major points:

1. Throughout the manuscript the authors write about genetic risk for MS/MS risk SNPs in general. However, while reading the manuscript it becomes clear that most of the reported effects could be observed/shown almost only for variants in the MHC region on chromosome 6. As an example, they state in the abstract that "...many of the genetic risk variants for MS rose to higher frequency among pastoralists...", which suggests that this can be observed for many of the by now 233 known MS risk variants across all chromosomes while they show it only for variants at the MHC region. In my opinion it should be made clear, that most of the results apply to MS risk variants within the MHC region while the observed effect could not be shown for the other >200 MS risk variants. I believe that the abstract as well as some parts of the Results and the Discussion section should be rephrased to make this clear (see also comment n. 3).
2. There are three SNPs in Figure 3a with a risk ratio > 1.2 or 0.8 for "Outgroup" ancestry. Have the authors looked at these SNPs in more detail?
3. The authors calculated Ancestral Risk Scores (ARS) as an aggregated risk for each ancestry and show that Steppe ancestry had a large risk while Neolithic Farmer and Outgroup ancestry had the

lowest ARS (Figure 3). It is then stated that this pattern holds even when excluding SNPs from the MHC region (Figure S4.1). However, Figure S4.1 shows that most of the reported effect is actually due to the MHC SNPs as there is no significant difference observable in Figure S4.1c) (confidence intervals cross 0 and there is no difference observable between Steppe and Outgroup ancestry). I am not convinced by these data that Steppe ancestry is associated with MS genome wide and I believe that the authors should provide for evidence for this or rephrase the text with regards to this point.

4. Figure 4 shows that ancestry explains more phenotype variance than GWAS or HTRX for all but the one LD block (4f) which is the LD block where the variance explained by GWAS and HTRX is highest. Similarly, in the analysis on the whole MHC region (Figure 4a) ancestry explains less variance than GWAS or HTRXs. This fact and its interpretation is not well described in the text. Is there any evidence that the fact that ancestry explains more variance in the other 3 LD block might not be explained by the ancestry effects in this 32.41-32.68Mb block? This seems relevant as the authors show that there is LDA between the different SNPs across the LD blocks. Did the authors perform any conditional analyses to show this (as has been done to show independent association to MS risk for the single SNPs investigated here)?

5. Can the authors provide more data to support the following statement at lines 235-236): “This interaction risk can be attributed to particular ancestries”.

6. Are all the SNPs that are significant in Figure 5a within the MHC region? If so, I believe that this should be clearly stated as it seems relevant that the observation is limited to the MHC region.

7. Can the authors describe more clearly what conclusion they draw from the results of the LDA score analysis (including Figure 6)?

8. In the performed literature research, the authors found evidence for association of a number of MS-associated SNPs with evidence for positive selection with a number of different phenotypes (pathogens, infectious diseases). 94 SNPs were investigated here (not LD-pruned), but it is not clear, whether these are all SNPs in the MHC region. This should be stated. What was the definition of a SNP being associated with another trait? Genome-wide significance (not according to ST13)? The authors state in the discussion (lines 433-434) that they have shown “that the majority of selected MS-associated SNPs are associated with protection against a wide range of pathogens”. It is not clear if all 94 investigated in the HLA region (according to ST13 this is true for all but 2) and to how many independent MS-risk signals they map. The statement above implies that most of MS risk signals are also protective of pathogens or infectious diseases which might be misleading.

9. The authors suggest that being protective for other (infectious diseases) could explain the positive selection for some of the MS risk alleles. This hypothesis could be supported if the authors could provide evidence for positive selection for other, non-MS-related variants with protective effects on the same traits. Have the authors performed such an analysis or could they do that?

10. The authors state in the Discussion section that ancestry considerations could be exploited for individual risk prediction. Individual genetic risk prediction for MS is currently not possible to a meaningful degree. I find it hard to believe – given the data shown in this report – that including ancestry information can improve prediction accuracy to any significant degree. Have the authors

attempted any risk prediction (for example using the UK biobank data) including ancestry consideration? Can the authors provide evidence that this would increase prediction accuracy?

Minor points:

1. Figure 3a seems overcrowded. I cannot distinguish “shaded” datapoints from not shaded ones, and the mean and confidence interval boxes are very difficult to see. In my opinion this figure should either be split or show only relevant data points. What is the meaning of the size of the dots in this figure? Can the authors explain why it is useful to have a mean and a confidence interval reported for each chromosome as independent disease risk signals might also be independent with regards to ancestry across a single chromosome?

2. In some of the figures the abbreviations used in the axis labels are not explained. For example “k_{ye}” in Figure 1c, “CE” in Figure 2B, “DAF” in Figure 5b, These should all be explained.

3. Supplementary Figure 4.1 is not well explained – does the top plot show the association of any ancestry with MS (per SNP) or only Steppe and Farmer ancestry?

Author Rebuttals to Initial Comments:

Comments from Reviewer 1:

Referee #1 (Remarks to the Author):

1) This is an excellent paper establishing the link between past selection in the HLA-region and increased risk of Multiple Sclerosis in present-day population. Overall, I found the work presented fascinating and that the key conclusions are well supported by the data. My comments/suggestions below are primarily aimed at improving the clarity of the manuscript, which, as it stands, it is extremely hard to follow.

Response: Thank you for pointing this out. We have made substantial changes to the main text in the results and methods section (described fully below). These are largely insertions at the beginning of each methods section to describe the purpose of the analysis. We have also provided a new Supplementary Figure 9.1 showing an overview of the methods used and what each method tells us. We hope that the combination of these changes makes the paper easier to follow.

Changes:

Methods section changes: We split the section "Population genetic analyses" into "Standard Population genetic analyses" and "Population painting".

In "Population genetic analyses", we added the sentence "The main population-genetics approach we base our inference on is Population-based painting (detailed below). However, to robustly understand population structure, we applied other standard techniques."

In "Population painting", we now describe the chromosome painting. We have added a sentence describing NNLS: "NNLS explains the genome-wide haplotype matches of an individual as a mixture of the genome-wide haplotype-matches of the reference populations. This setup allows both the reference panel and any additional samples (i.e. modern) to be described using these 6 ancestries (Figure 1)." We also added that chromosome painting generates local and global ancestry: "This generated both local ancestry probabilities and genome-wide ancestry fractions for each painted individual."

In the section “Local ancestry”, this is now split into “Local ancestry from Population painting” and “Pathway painting”, to emphasize the two sources of local ancestry used in this paper. We added the sentence “Chromosome Painting provides an estimate of the probability that an individual from each reference population is the closest match to the reference individual at every position in the genome. This provides our first estimate of local ancestry from Allentoft et al. (2022)⁹: the population of the first reference individual to coalesce with the target individual, as estimated by Chromopainter”; and “An alternative approach is to identify which of the four major ancestry pathways (Farmer, CHG, EHG, WHG) each position in the genome best matches to. This has the advantage of not forcing haplotypes to choose between “Steppe” ancestry and its ancestors, but the disadvantage of being more complex to interpret. To do this, we modelled ancestry path labels...”

In the section “Regions of unusual ancestry and gene enrichment”, which we now call “Anomaly Score: Regions of Unusual Ancestry”, we have added the sentence “This is the Normal-distribution approximation to the Poisson-Binomial score for excess ancestry, for which a detailed simulation study is presented in Nelson et al. (2017)”.

We merged the sections “Cluster Analysis” and “Weighted Average Prevalence” which are now grouped as “Weighted Average Prevalence (WAP)”. To make the purpose of this analysis clearer, we added the sentences “we developed a statistic that could account for the origin of risk to be identified with multiple ancestry groups, which do not have to be the same set for each SNP” and “We can then compute the Weighted Average Prevalence (WAP) which summarises these results into the ancestries”.

We also added a technical sentence to more fully describe the method “The standard

$$\text{deviation of } \bar{\pi}_{jk} \text{ is computed as } sd(\bar{\pi}_{jk}) = \sqrt{\sum_{m=1}^6 w_{jkm}^2 \sigma_m^2}, \text{ where } w_{jkm} = \frac{P_{jkm}}{\sum_{m=1}^6 P_{jkm}},$$

$\sigma_m = \frac{s(y_{jm})}{\sqrt{n_{jm}}}$ and $s(y_{jm})$ is the standard deviation of the outcome for the individuals in the m th cluster. We also test the hypothesis that $H_0: \bar{\pi}_{jk} = \bar{\pi}$ against $H_1: \bar{\pi}_{jk} \neq \bar{\pi}$, and compute the p-value as $p_{jk} = 2(1 - \Phi(\frac{|\bar{\pi} - \bar{\pi}_{jk}|}{sd(\bar{\pi}_{jk})}))$.

In the “PCA/UMAP of WAP/Average Dosage” section we added “To sort risk-associated SNPs into ancestry patterns according to that risk”.

In the “GWAS of Ancestry and Genotypes” section we added the sentence “The total variance of a trait explained by genotypes (SNP values), Ancestry, and haplotypes (described below) is a measure of how well each captures the causal factors driving that trait. We therefore computed the variance explained for each data type in a “head-to-head” comparison, either at specific SNPs or SNP sets. In this section we describe the model and covariates accounted for.”

In the “GWAS comparison for trait-associated SNPs” section, we added the sentence “In this section we describe how we moved from associations between observations on SNPs (either genotype values or ancestry) and a trait, to total variance explained.”

In the “Haplotype Trend Regression with eXtra flexibility (HTRX)” section, we added “Ancestry is a strong predictor of MS, but we wanted to understand whether it was tagging some causal factor that was not in our genetic data, or whether it was tagging either interactions or rare SNPs. To address this, we...”.

We renamed the section “Polygenic Selection Test” to “Quantifying selection via historical allele frequencies from Pathway Painting”. We added a sentence “The historical trajectory of SNP frequencies is a strong signal of selection when ancient DNA data are available. This is the main purpose of our Pathway Painting method, and can be used to infer selection at individual loci, and combined into a polygenic score by analysing sets of SNPs associated with a trait.”

In the section “Linkage Disequilibrium of Ancestry (LDA) and LDA Score (LDAS)”, we added the sentences “Just like the values of the genotype, ancestries can be correlated along the genome, and further, deviations from the expected length distribution for a particular ancestry is a signal of selection, dated by the affected ancestry. We propose an ancestry linkage disequilibrium (LDA) approach to measure the association of ancestries between SNPs, and an LDA Score (LDAS) to quantify deviations from the null hypothesis that ancestry is inherited at random across loci. LDA is defined in terms of local ancestry.”

In the Discussion, we split the RA results from the MS results to make our conclusions clearer. There is a new paragraph now: “Similarly, the new pathogenic challenges associated with agriculture, animal domestication, pastoralism, and higher population densities might have substantially increased the risk of triggering a systemic RA-associated inflammatory state in genetically predisposed individuals. This could lead to an increased the risk of a serious outcome following subsequent infections, years before any potential joint lesions, resulting in negative selection and thus, might present a parallel between RA-associated

inflammation in the Bronze Age and MS today, in which lifestyle changes have exposed previously favourable genetic variants as autoimmune disease risks. “.

2) The authors present a total of 5 new statistical approaches to demonstrate that Steppe ancestry is associated with risk of MS in contemporary populations but there is virtually no description of how each of these methods work in the actual manuscript. The reader needs to dig deep into the ~100 pages of supplements to get a sense of how each of these new methods works. Even then, one is left wondering how well they work given that each of these methods could be a paper in itself with associated simulations, etc to formally demonstrate that the approaches are well calibrated and lead to accurate estimates. That said, and given that all analyses point towards the same direction, I am confident that the conclusions are correct, but I am left wondering if it would not be better to focus on only a subset of the analyses (and explain those in better detail) since some of them appear to be redundant (more on that later).

Response: The five new statistical approaches that we believe are being referred to are the ARS, HTRX, painting Anomaly Score, LDA, and WAP. As we hope is now clearer, we think that each method adds something unique to the paper.

Regarding validation and detail, the ARS is actually introduced in our companion paper, Irving-Pease et al (2022), with demonstrations of the statistics on previously studied phenotypes like height and BMI.

For HTRX, we now cite a preprint that explains this method in more detail.

We would emphasise that the Anomaly Score and LDA are relatively simple transformations of the raw painting which is used in many places, and recapitulates known examples of ancestry-specific selection such as the LCT locus (supplementary figure 6.4).

Simulation studies for both LDA and HTRX methods are included in the supplementary materials.

To better explain what everything is for, we have provided a new Supplementary Figure 9.1, which outlines what data we've used and why we've used them, and provides a synopsis of the paper from the perspective of the methods-based evidence we use to shore up our claims.

We also discuss below (in response to your comment 5) what can be concluded from the ARS that cannot be concluded from other methods.

Changes: Each sub-section in the methods section now starts with a few motivating sentences explaining the purpose of the analysis (outlined above, comment 1), and there is a new Supplementary Figure 9.1. We have also now included a reference to a preprint describing the HTRX analysis (Yang & Lawson, 2022).

3) Along the same lines, even the dataset itself is hardly described in the main text (they only say “Using the largest dataset from the Stone Age coupled with 86 new Medieval and post-Medieval genomes from the Denmark”). I realize that a full description of the dataset is

provided on an accompanying study and in the Supplements but the actual paper should be clear on its own.

Response: We agree that it was unclear in the main text what the dataset consists of.

Changes: We have added "...using the largest ancient genome dataset from the Stone Age (full description in Allentoft et al., 2022)..." and the sentence "This dataset totals 1,750 imputed diploid shotgun-sequenced ancient genomes (ST15), of which 1,509 are from Eurasia" in the second paragraph of the results section. ST15 contains metadata for all ancient samples used in this study, including their original publications.

4) In the risk-ratio analyses (Figure 3) the authors describe how the Steppe ancestry has the highest risk ratio for MS in nearly all HLA SNPs while Farmer ancestry and "outgroup" appear to be protective. However, the "outgroup" ancestry for one SNP outside the HLA region ("rs771767") seems to show a risk ratio as large as that of HLA. Can the authors describe what this SNP is tagging?

Response: Thank you for pointing this out. Your comment was also made by reviewer 3. To address this, we computed the confidence interval and p-values of WAP (described in Supplementary Note 3). The SNPs identified as high risk ratio (as well as a SNP from chr 1 with lowest risk ratio) in the outgroup ancestry were found to be insignificant at the 1% level, so they are no longer shown in Figure 3.

Changes: Only statistically significant SNPs ($p < 0.01$) are now shown in Figure 3, meaning this SNP is no longer highlighted.

5) I am not sure to understand the value of the Ancestral Risk Score (ARS) analyses over the risk ratio. If the alleles that contribute the most for MS risk are in the HLA regions, and we already know that those are enriched for Steppe ancestry aren't we just measuring the same thing in a different way? Indeed, from what I can tell, all the significant in ARS comes from the HLA region alone (i.e., the other non-HLA MS-risk alleles do not significantly contribute to the enrichment of Steppe ancestry).

Response: We agree that this was not made clear in the original main text. The ARS provides evidence that overall risk is also associated with Steppe ancestry at SNPs not in the HLA region: when we calculate the ARS for these SNPs, Steppe ancestry still confers the highest risk, while Farmer is the lowest. Outgroup is protective in the HLA but confers risk at SNPs outside it. No other method except ARS tells us about the aggregate non-HLA risk for MS.

Changes: We now include the ARS results (using an individual-bootstrapping method) for non-HLA SNPs in Figure 3.

6) The least supported claim of the paper is the link between selection on MS-risk alleles and protection against infectious diseases. The link was based on manual curation of the published literature and most likely based on genetic associations that are not statistically robust. Indeed, as the authors mentioned, most genetic associations with infectious diseases have not been based on GWAS and instead are derived from gene candidate studies. Gene candidate studies are (i) biased towards genes that are more likely to be involved in immune responses, and (ii) afflicted by high rates of false positives, making it hard to know how many of those associations are truly real.

Response: We agree with the reviewer's comment that some of the associations from the manual literature curation may not be statistically robust, because of the concerns outlined with candidate gene studies and small sample numbers etc. Furthermore, the current vaccination programs against previously prevalent diseases such as measles, rubella and mumps, make associations difficult to observe. To address this, we have used three sources of data to investigate the associations with the selected MS-associated SNPs and a range of pathogens: UK Biobank GWAS, FinnGen GWAS, and the manual literature curation. For the latter, which seems to be of the most concern, we have included references in ST13 and ST14. We think that together, these all point towards a wide range of pathogens driving the increased risk of MS in the past.

We have also softened the language used in the main text to reflect the poor quality of the data.

Changes: We have added the sentence "We emphasise that although this evidence is strongly suggestive, many of these putative associations may not be statistically robust due to underpowered GWAS and the bias in candidate gene studies."

7) I would like to have seen some evidence that the overlaps that they found between MS-SNPs and genetic risk for infectious diseases were larger than expected by chance when focusing on risk alleles for other immune-related disorders.

Response: It is difficult to define what the null distribution of SNPs should be for comparison with MS: it is unlikely that there has been selection for autoimmune diseases per se and it is highly likely that most, if not all, autoimmune diseases in part are the by-product of adaptive changes to a multitude of pathogen challenges in the past, and we have not made a contrasting claim about MS versus other autoimmune diseases. We therefore do not think that repeating our analyses for another autoimmune disease would be informative about whether the signal we are seeing for MS is 'by chance'. However, to further interrogate the relationship between MS-SNPs and genetic risk for infectious diseases, we performed an additional analysis to look at the cross-over between selected MS-associated SNPs and SNPs which are associated with phenotype in the newly-released FinnGen study. We found that again many infectious disease phenotypes share SNPs with those driving the selection for MS, providing further evidence of selective pressures via pathogen exposure (described more fully above).

8) After having read the paper at least 5 times, my conclusion is that the HLA-associated risk factor from MS is enriched for Steppe ancestry and it was likely positively selected in the past (maybe to protect against infectious diseases but I found that less compelling). That conclusion, however, is quite different from the claim on the title that "Genetic risk for MS originated in pastoralist steppe populations". The variance explained in MS for the HLA locus is below 3% and therefore it is not representation of "Genetic risk for MS", which extends way beyond HLA.

Response: The genetic risk for MS is indeed bigger than the "identifiable genetic risk" component in our data at pre-identified MS risk loci from the IMSGC (i.e. there is 'missing heritability', as is standard with GWAS data). As is usual with genetic studies, the goal of analysing relatively minor contributions to total variability is ultimately to understand the causal pathways underlying the disease. However, in our analysis, the figure of 3% is for

HLA alone from Figure 4 (a). We explain here why this is an underestimate, why the true contribution of the HLA is higher, and also why risk for MS is Steppe-associated outside the HLA.

3% is an underestimate of the HLA contribution because McFadden's R^2 systematically underestimates: 'glm' is a more reasonable model instead of 'lm' which would show larger R^2 , and McFadden's R^2 (for glm) is a different estimate of R^2 from standard R^2 for 'lm', see Supplementary methods for detail. Simulations (Supplementary Figure 4.4 and Yang & Lawson, 2022 Figure 1) show that by changing the outcome from continuous to binary, McFadden's R^2 for 'glm' becomes much smaller than standard R^2 for 'lm'. Furthermore, we are also underpowered with UKB data due to the small number of cases.

The HLA locus represents about 20% of the total identified genetic variance (IMSGC, 2019), while the risk genes associated with MS account for approximately half of the observable heritability (Parnell & Booth, 2017). Furthermore, our ARS analysis suggests that even when excluding the HLA, the risk for MS is highest in Steppe ancestry (new Figure 3) in the loci that have been identified. However, while Steppe ancestry confers risk for MS outside of the HLA, most of the signal we observe is from HLA variants, as would be expected. We have therefore modified parts of the text and the title of the manuscript to reflect this.

Response: The title was, by necessity, a simplification - on reflection we think that the updated title "Elevated genetic risk for multiple sclerosis originated in Steppe Pastoralist populations" is more accurate. Our claims have been rewritten with this and other comments in mind.

Comments from Reviewer 2:

Referee #2 (Remarks to the Author):

1) In this work, Barrie and colleagues take a deep dive into investigating the ancient genetic ancestry of multiple sclerosis (MS) risk loci using the ancient genome dataset from the Stone Age they describe in an accompanying paper. They show that Steppe ancient ancestry is associated with an increased risk of MS while the farmer and outgroup African ancestries are associated with protection against the diseases. They also show that MS-associated risk variants have gone under positive selection in the ancestral Steppe population as well as descendant Europeans. They speculate that this positive selection may have been due to the protective effect of these variants against different pathogens. The authors also look into the selection pattern of rheumatoid arthritis (RA)-associated risk SNPs and observe an opposite pattern. RA-associated variants show evidence of negative selection in ancient genomes although they see that these negatively selected RA-risk variants are also associated with protection against the same pathogens as the ones associated with positively selected MS-risk variants.

Overall, this study is interesting. It brings insight into the relatively high prevalence of MS in individuals of European ancestry and its higher prevalence in Northern vs Southern Europe. I do have a few concerns/questions about the strength of some of the findings as well as some of the conclusions. I list them below:

1- I had a hard time judging the significance of some of the findings, specifically the ones related to the HTRX and LDA methods. The authors mention both methods/metrics are

developed for this paper. Neither of these methods is peer-reviewed. Reviewing these two methods in depth requires more details, examples, and comparison with existing methods as such I am not sure about the robustness of either method and the significance of the conclusion based on these two metrics.

Response: This comment relates to comment 2 by reviewer #1. We have provided a new Supplementary Figure 9.1, which outlines what data we've used and why we've used them, and provides a synopsis of the paper from the perspective of the methods-based evidence we use to shore up our claims.

We have now cited a preprint for HTRX method in the main text (Yang & Lawson, 2022). This provides further details of the new method, which is compared to existing methods (SNP-based regression and HTR).

LDAS is a relatively simple transformation of the raw painting which is used in many places, and recapitulates known examples of ancestry-specific selection such as the LCT locus (supplementary figure 6.4). Simulation studies for both methods are included in the supplementary materials.

Changes: We now cite the preprint for our new method (HTRX) in the main text, in the paragraph introducing the method.

2) - How "recombinant favouring selection" (line 304) is related to (or different from) heterozygous advantage and balancing selection both of which are at play in the HLA region.

Response: Recombinant favouring selection is different to heterozygous advantage or balancing selection as these describe single-locus processes. In contrast, our simulation study (Supplementary Figure 6.2-3) shows that the LDAS signature cannot be replicated by these selection types, instead requiring the evolution of two distinct (recombining) alleles in different populations.

Changes: To emphasise this in the text, we added in the results section: "Extending multi-SNP selection models, our explanation (Supplementary Figure 6.1) is that at least two separate loci rose selectively in separate populations that later admixed and remained selected in the HLA, justifying a new term, "recombinant favouring selection". This means that there was selection for diverse ancestry in the HLA region, driven by recombination."

3) How accurate is ancient LAI in HLA region? As authors know imputation in HLA is not accurate in the absence of high-quality population-specific reference panels (see Luo et al Nat gen, 2021) similarly LAI relies on the presence of high-quality and high-density reference panels. Given the sparsity of aDNA how accurate do authors think is their ancient LAI? Is there any way to quantify this or assign a confidence score to ancient LAI calls?

Response: This is a good point to raise as the HLA is a notoriously difficult region to study. Two observations give us confidence that the LAI in the HLA region is accurate. The first is that we observe reasonably long admixture tracts, meaning there is not a clear failure due to imputation or phasing problems. The second is that the chromopainter algorithm as implemented will regress to the genome-wide mean for each ancestry if the nearest neighbour (i.e. local ancestry inferred) is uncertain; the fact that we see deviations away from this mean (i.e. the extremes of ancestry dosages) is evidence that this signal is real:

any problems with phasing or imputation would push the signal closer to the mean in this region.

4) - How well does the sum of ancient LAI correlate with ancient global ancestry inference for the represented ancestries?

Response: This is a pertinent question. This is the same method used in Margaryan et al (2020) in which a variety of sensitivity analyses were performed in the extensive supplement. The sum of ancient local ancestry is the measure, called “chunklengths” or just “lengths” in the literature, used to perform ancestry inference. The quantity we called LAI here is not precisely an ancestry measure - it is the closest genealogical relationship, from which true ancestry can be inferred (as done by e.g. MOSAIC, Salter-Townsend & Myers 2019). Here we use NNLS to estimate ancestry as done by Margaryan 2020, and introduced in the very widely used GLOBETROTTER method (Hellenthal et al 2014). They correlate but measure different quantities, and we stress that accurate estimation of global ancestry is not critical for the analysis presented here.

5) 3- The ancient ancestry, based on what’s reported here, is different between Northern and Southern Europeans. The authors also report the ancestry profile of RA and MS risk variants to be strikingly different and somewhat opposite of each other. How do authors reconcile these observations with the fact that both RA and MS are more frequent in Northern Europe?

Response: RA risk is most associated with eastern and western hunter-gatherer ancestry, which is also higher in northern Europe than southern Europe, so we do not think there is a contradiction here. Average ancestry components for each country can be seen in Allentoft et al. (2022) Figure 5. The ‘opposite’ nature of our findings between MS and RA is that the former shows positive selection in one ancestry while the latter shows negative selection in another.

6) - On a related note, I find the conclusion that infectious diseases are the driver of both the negative selection of RA-associated risk variants and positively selected MS-risk variants speculative. It is true that infectious diseases are a major selection pressure and that the risk of other immune-mediated diseases (including autoimmune diseases) is likely to be affected by past and current pathogens. However, most of the pathogens cited by the authors are/have been covering wide geographical regions encompassing Eurasia and thus I am not sure why they should have presented selective pressure in one ancestral group and not in others. Can you comment?

Response: It is true that most of the listed pathogens cover wide geographical ranges today; however, not all of these pathogens were widespread during the Bronze Age. Indeed, the description of a mumps outbreak by Hippocrates, approximately 2432 BP, suggests that this was not an endemic disease at that point in time (Mammas & Spandidos, 2016), and the critical community size for measles to become endemic is ~250,000-500,000 individuals, which rules out endemism in European Bronze Age communities (Düx et al., 2020).

Furthermore, phylogenetic analyses estimate that many of these pathogens have their most recent common ancestor (MRCA) in this time period (please see Discussion). These

estimates support the notion that these pathogens were differentially infecting human populations and that there was a large increase in transmission of - and selection on - them which coincided with the population size increase and population dispersals and admixtures during the Bronze Age. If the pathogens were circulating to the same extent in all populations, and all populations had had a similar level of immunity towards them, such a large increase in transmission and selection pressure would likely not have occurred, and the MRCA to extant lineages of these pathogens would not have originated in the Bronze Age. Moreover, some infectious diseases, like mumps and measles, may have been novel zoonotic diseases spreading through human populations.

The main argument that we make in the paper which addresses this is that it was lifestyle differences between ancient populations which drove their differential exposure to pathogens, and subsequent population-dependent evolution of the HLA. For example, the Steppe pastoralists, who consumed large amounts of meat and dairy products, would have been in much closer contact with domesticated animals than their farming or hunter-gatherer counterparts.

Finally, we show that HLA-DRB1*15:01 originated in the population basal to European farmer and Steppe lineages. The fact that this variant was part of the standing genetic variation presumably meant that it was available to be rapidly selected when the selective pressures changed - which would not have been possible in the WHG/EHG populations where it was not present.

7) 4- There is no mention of other similar work in the intro or discussions. Two examples that come to my mind are the recent paper about the Black death and Chron's disease risk from the Barreiro group (Nature 2022) and the one on TB risk and changes in MAF of a known TYK2 risk variant from the Quintana-Murci group (AJHG 2021). Please cite the relevant previous work.

Response: We agree that these citations are relevant as recent examples of pathogen-driven immune gene evolution.

Changes: We have incorporated the citations in the discussion: we have added the sentence "Similar examples of pathogen-driven evolution have recently been published (Klunk et al., 2022, Kerner et al., 2021)".

8) - Related to the above point have you looked at the changes in MAF of the variants reported in the above two papers in your cohort through time (and if possible, in different ancient ancestral groups)? I would expect to see your data recapitulate the same patterns as reported before. Adding this as a sup can add to this work's robustness.

Response: We have not done this as we did not perceive it to be directly relevant to the MS/RA question (although in theory it is of course possible, it is not a small amount of work to undertake).

9) Other:

5- Line 220: Across the entire HLA region, haplotypes explain at least 17% more out-of-sample variation than GWAS (2.90%, compared to 2.48%). I am not sure where the

number 17% comes from, $2.90 - 1.48 = 1.42$. The number is again cited in the discussion please elaborate.

Response: This is a calculation of the ratio rather than the absolute difference, i.e. $(2.90\% - 1.48\%) / 1.48\% = 17\%$. Therefore it represents a 17% increase in performance.

10) 6- Through the figures, CHG, WHG, EHG, farmer, and outgroup are always represented but results related to ANA, Steppe, and Yamnaya groups are shown in some figures only (see fig 2, 3, and 5). I am not an ancient genome expert, but I find the lack of consistency confusing in following the paper and interpreting of the results from different figures.

Response: Thank you for pointing this out. We have updated Figure 2 to show 'Steppe' rather than 'Yamnaya' - this was an error using an old naming system. The reason that Figure 5 differs in naming is because it is referring to painting pathways rather than populations. This figure is based on the 'pathway painting' local ancestry dataset rather than the 'population painting'.

Changes: We have updated Figure 2.

We have directed readers to Methods 'Pathway painting' in the caption for Figure 5. We have also made significant changes to the Methods section to differentiate the two forms of local ancestry (now called population painting and pathway painting):

We split the section "Population genetic analyses" into "Standard Population genetic analyses" and "Population painting". In "Population painting", we now describe the chromosome painting. The next section "Local ancestry" is now split into "Local ancestry from Population painting" and "Pathway painting", to emphasize the two sources of local ancestry used in this paper. We added the sentence "Chromosome Painting provides an estimate of the probability that an individual from each reference population is the closest match to the reference individual at every position in the genome. This provides our first estimate of local ancestry from Allentoft et al. (2022): the population of the first reference individual to coalesce with the target individual, as estimated by Chromopainter"; and "An alternative approach is to identify which of the four major ancestry pathways (Farmer, CHG, EHG, WHG) each position in the genome best matches to. This has the advantage of not forcing haplotypes to choose between "Steppe" ancestry and its ancestors, but the disadvantage of being more complex to interpret. To do this, we modelled ancestry path labels..."

We hope that these changes make the differences between the two sources of local ancestry data clear. We also hope that the new Supplementary Figure 9.1 makes the difference between 'population painting' and 'pathway painting' clearer.

11) 7- Why did the authors choose CLUES over other selection methods (EHH, composite score, etc) will the results change if another selection metric is used?

Response: We chose to use CLUES as the method for inferring evidence of selection at individual trait associated loci for several reasons. The original CLUES algorithm (Stern et al. 2019 *PLOS Genetics*) has recently been adapted to make use of time-series data, obtained from aDNA. In our companion paper, we find that aDNA dramatically increases statistical power to detect variants under selection; via a systematic comparison of CLUES selection results using either modern data alone or a time-series of aDNA genotype probabilities

(Irving-Pease et al. 2022 *bioRxiv*). We also find that the complex admixture history of human population in West Eurasia has masked evidence of selection at many trait associated loci across the genome, which motivates our use of local ancestry stratification in our CLUES time-series analyses. In addition, CLUES allows us to infer the underlying allele frequency trajectory of each variant, which provides important information about the timing of selection, in addition to the intensity. Lastly, the posterior likelihood results from each individual CLUES model can be aggregated into a formal statistical test of polygenic selection, using the method PALM (Stern et al. 2021 *AJHG*).

12) 8- Line 377: by “directional selection” do you mean both positive and negative? If yes, it makes this paragraph a bit confusing given that it mentions negative selection at the start of the paragraph.

Response: We agree that the choice of wording here was confusing.

Changes: We have deleted the word “directional”.

Comments from Reviewer 3

Referee #3 (Remarks to the Author):

1) In this manuscript by Barrie et al. the authors provide evidence that genetic risk variants for multiple sclerosis (MS) rose to higher frequency in a specific ancient population, were brought into Europe app. 5000 years ago and were subject to positive selection. Furthermore, by providing evidence of association of some of the MS risk variants to protection from different infectious diseases the authors provide a possible explanation for the observed positive selection of these MS risk alleles.

I believe that the data presented in this report will be interesting for researchers with a focus on MS genetics, as they provide evidence for a specific ancient population and time period in driving MS risk development. The results presented here could stimulate more research in the area of ancestry association to gain more insight into the mechanisms of MS risk.

A potentially very interesting result is the finding that some MS risk variants under positive selection during the investigated time period appear to be protective against different pathogens/diseases. While this is a possible explanation for the observed positive selection, I believe this could be further investigated as I describe in detail below (see comment 9).

Most of the results and (statistical) analyses were clear and conducted thoroughly so that they support the drawn conclusions. However, I have a few questions/comments listed below.

Major points:

1. Throughout the manuscript the authors write about genetic risk for MS/MS risk SNPs in general. However, while reading the manuscript it becomes clear that most of the reported effects could be observed/shown almost only for variants in the MHC region on chromosome 6. As an example, they state in the abstract that “...many of the genetic risk variants for MS rose to higher frequency among pastoralists...”, which suggests that this can be observed for many of the by now 233 known MS risk variants across all chromosomes while they show it

only for variants at the MHC region. In my opinion it should be made clear, that most of the results apply to MS risk variants within the MHC region while the observed effect could not be shown for the other >200 MS risk variants. I believe that the abstract as well as some parts of the Results and the Discussion section should be rephrased to make this clear (see also comment n. 3).

Response: We agree that most of the signal is driven by changes in the MHC, and we have modified the text in the abstract and results section to make this clear. However, we note that although most of the risk for MS is driven by SNPs on the HLA, the signal remains (i.e. Steppe ancestry contributes the most risk for MS) even when excluding the HLA (we address this more fully below). This is now shown using ARS in Figure 3.

The explanation for this observation is that the selection signal is not driven by MS itself as a phenotype (i.e. the MS phenotype was never adaptive); instead, protection against a broad range of pathogens drove selection of the HLA, which now confers risk for MS.

Changes: We now show the ARS results (using individual-bootstrapping method) for non-HLA SNPs in Figure 3. Additionally, we have changed the title of the manuscript to “Elevated genetic risk for multiple sclerosis originated in Steppe Pastoralist populations” in order to reflect this.

2) 2. There are three SNPs in Figure 3a with a risk ratio > 1.2 or 0.8 for “Outgroup” ancestry. Have the authors looked at these SNPs in more detail?

Response: Thank you for pointing this out. This comment was also made by reviewer 1. To address this, we computed the confidence interval and p-values of WAP (described in Supplementary Note 3). The SNPs identified as high risk ratio (as well as a SNP from chr 1 with lowest risk ratio) in the outgroup ancestry were found to be insignificant at the 1% level, so they are no longer shown in Figure 3.

The protective SNPs are still significant however. rs9268839 is tagging rs10093 (ST4), which in turn is associated with the DQA1 locus (ST3). rs3135388 is a tag for HLA-DRB1*15:01 (ST4). Both of these are also significant as high OR for Steppe ancestry: this means that in each case ancestry is strongly associated with the SNP effect.

Changes: Only significant SNPs are now shown in Figure 3, meaning the high effect SNP in Outgroup ancestry is no longer highlighted.

3) 3. The authors calculated Ancestral Risk Scores (ARS) as an aggregated risk for each ancestry and show that Steppe ancestry had a large risk while Neolithic Farmer and Outgroup ancestry had the lowest ARS (Figure 3). It is then stated that this pattern holds even when excluding SNPs from the MHC region (Figure S4.1). However, Figure S4.1 shows that most of the reported effect is actually due to the MHC SNPs as there is no significant difference observable in Figure S4.1c) (confidence intervals cross 0 and there is no difference observable between Steppe and Outgroup ancestry). I am not convinced by these data that Steppe ancestry is associated with MS genome wide and I believe that the authors should provide for evidence for this or rephrase the text with regards to this point.

Response: We agree that this was unclear as originally written. However, the ARS does provide evidence that overall risk is associated with Steppe ancestry at SNPs not in the HLA region: when we calculate the ARS for these SNPs and bootstrap using individuals, Steppe ancestry still confers the highest risk, while Farmer is the lowest. We now include this in

Figure 3. Figure S4.1, which the reviewer rightly points out does not demonstrate this, showed results when bootstrapping over SNPs rather than individuals for *all* SNPs.

Changes: We have updated Figure 3 to show the ARS results using individual-bootstrapping methods for non-HLA SNPs.

4) 4. Figure 4 shows that ancestry explains more phenotype variance than GWAS or HTRX for all but the one LD block (4f) which is the LD block where the variance explained by GWAS and HTRX is highest. Similarly, in the analysis on the whole MHC region (Figure 4a) ancestry explains less variance than GWAS or HTRXs. This fact and its interpretation is not well described in the text. Is there any evidence that the fact that ancestry explains more variance in the other 3 LD block might not be explained by the ancestry effects in this 32.41-32.68Mb block? This seems relevant as the authors show that there is LDA between the different SNPs across the LD blocks. Did the authors perform any conditional analyses to show this (as has been done to show independent association to MS risk for the single SNPs investigated here)?

Response: We agree that this is a possible, and potentially even likely explanation. We do not intend to claim that ancestry is causal, only that it is an effective tool in understanding the total effect from haplotypes that evolved in a particular population. Because we cannot separate the effects of interactions, tagging of rare causal SNPs, and tagging between different regions, we did not feel any conditional analysis is likely to be informative on these data. To investigate further requires more MS cases, to give power for specific interactions to be detected.

Changes: We now say “the increased performance of local ancestry over regular GWAS in some regions can be explained by tagging of SNPs outside the region” to make this clear.

5) 5. Can the authors provide more data to support the following statement at lines 235-236): “This interaction risk can be attributed to particular ancestries”.

Response: The interaction risk is only quantified in the HTRX analysis (Figure 4). The difference in R^2 between HTRX and GWAS is most likely to be due to interaction (though we cannot rule out tagging rare snps). The risk SNPs in these regions are associated with ancestry (Figure 3) and are in ancestry LD (Figure 4h). Whilst this interaction risk is unlikely to be explained by an independent factor, we agree it is not strongly evidenced and therefore changed the text.

Changes: We replaced the statement with: “Multiple SNPs at the 32.41-32.68Mb region are Steppe-associated, have high MS odds ratios, and are in LDA (Figure 4), which may explain the increased HTRX variance explained.”

6) 6. Are all the SNPs that are significant in Figure 5a within the MHC region? If so, I believe that this should be clearly stated as it seems relevant that the observation is limited to the MHC region.

Changes: The majority of the selected SNPs shown in Figure 5 are located in the HLA region (chr6:28477797-33448354). To make this clearer, we have updated the figure to include an annotation indicating which SNPs fall within this region. We have also updated all the relevant Supplementary Figures in section 6.

7) 7. Can the authors describe more clearly what conclusion they draw from the results of the LDA score analysis (including Figure 6)?

Response: Other referees requested similar clarity and we have tried to address this by better explaining the point of each analysis as they are introduced. Specifically, LDAS explores the nature of selection, which we explain with Supplementary Figure S9.1 and in the main text with the following content.

Changes: We have added the sentence “our explanation (Supplementary Figure 6.1) is that at least two separate loci rose selectively in separate populations that later admixed and remained selected in the HLA, justifying a new term, “recombinant favouring selection”. This means that there was selection for diverse ancestry in the HLA region, driven by recombination” in the results section to explain our main conclusions from the LDA analysis.

8) 8. In the performed literature research, the authors found evidence for association of a number of MS-associated SNPs with evidence for positive selection with a number of different phenotypes (pathogens, infectious diseases). 94 SNPs were investigated here (not LD-pruned), but it is not clear, whether these are all SNPs in the MHC region. This should be stated.

Response: To harmonise the MS and RA analyses, we have now used only pruned SNPs for the literature search (previously we used pruned for RA and unpruned for MS). This change does not change any results substantially, although it slightly decreases the number of MS-SNPs available for analysis (full list in ST13). However, even this reduced SNP set still shows a range of pathogen associations similar to those described previously. We now also describe how many of these selected SNPs are in the HLA region.

Changes: We have modified the text to show this: “showed statistically significant evidence for selection using CLUES (n = 39, of which 33 (85%) are in the HLA region)”.

9) What was the definition of a SNP being associated with another trait? Genome-wide significance (not according to ST13)?

Response: We used two definitions: the first was a UKB trait or crossover analysis which required both to be genome-wide significant ($p < 5e-8$). The second was in the manual curation of the literature, where not all associations were from GWAS (e.g. some from candidate gene studies etc) so a variety of thresholds were used, based on study design. Since the first submission, we have also replicated the UKB crossover analysis on the newly published FinnGen data, again using genome-wide significance as a threshold. This is described more fully at the top of this letter.

Changes: We have added the caveat “We emphasise that although this evidence is strongly suggestive, many of these putative associations may not be statistically robust due to underpowered GWASs and the bias in candidate gene studies” in the results section to highlight the variety in these studies.

10) The authors state in the discussion (lines 433-434) that they have shown “that the majority of selected MS-associated SNPs are associated with protection against a wide range of pathogens”. It is not clear if all 94 investigated in the HLA region (according to ST13 this is true for all but 2) and to how many independent MS-risk signals they map. The

statement above implies that most of MS risk signals are also protective of pathogens or infectious diseases which might be misleading.

Response: As described above, to harmonise the MS and RA analyses, we have now used pruned SNPs for the literature search (previously we used pruned for RA and unpruned for MS). This means that each SNP association is independent (without changing any other results substantially).

11) 9. The authors suggest that being protective for other (infectious diseases) could explain the positive selection for some of the MS risk alleles. This hypothesis could be supported if the authors could provide evidence for positive selection for other, non-MS-related variants with protective effects on the same traits. Have the authors performed such an analysis or could they do that?

Response: We agree that a systematic analysis of evidence for selection favouring non-MS related variants with protective effects for infectious disease would be informative; however, such an analysis is outside of the scope of this paper. A confounding issue with attempting to identify specific infectious disease drivers is that there is a general lack of well-powered GWAS for most infectious diseases (due to low prevalence in present-day Europe). However, to address the general question of which other traits may have contributed to the observed signals of selection for MS risk, we performed two new analyses. Firstly, we did an overlap analysis with both the UK Biobank and the FinnGen study (which has higher incidence rates of infectious disease than UKBB). Consistent with our interpretation that selection was on a mosaic of traits across history, we observed that all of the MS-associated selected SNPs were also associated with one or more other traits in UKBB (Supplementary Figure 7.3) and FinnGen (Supplementary Figure 7.6), and that 56% were associated with an infectious disease phenotype or related symptom (Supplementary Figure 7.7). We then sought to determine if the observed signal of polygenic selection favouring MS risk could be better explained by selection acting on a genetically correlated trait, using a formal statistical test designed to disentangle polygenic selection on genetically correlated traits. Our results show that no single UK Biobank or FinnGen trait significantly attenuated the signal of polygenic selection favouring MS risk (Supplementary Note 6).

12) 10. The authors state in the Discussion section that ancestry considerations could be exploited for individual risk prediction. Individual genetic risk prediction for MS is currently not possible to a meaningful degree. I find it hard to believe – given the data shown in this report – that including ancestry information can improve prediction accuracy to any significant degree. Have the authors attempted any risk prediction (for example using the UK biobank data) including ancestry consideration? Can the authors provide evidence that this would increase prediction accuracy?

Response: We claim “Our HTRX association results imply that ancestry-specific haplotypes, rather than only tagging SNPs, can be extracted from data for risk prediction.” i.e. we are claiming that HTRX-style haplotypes, not ancestry, is potentially useful for risk prediction (on the basis that it explains more variation). We agree that this is not likely to lead to actionable genetic risk, as we only see a 17% increase in predictive power (and 100% would not be enough for that). It is however a considerable improvement for other uses. This is computed from out-of-sample prediction using McFadden’s R^2 in the genotype/local ancestry/HTRX

analysis (Figure 4). However, it is not possible to attempt this with other cohorts (e.g. other Biobanks) because we only have local ancestry labels in the UK Biobank, and generating these in other databases would require a very significant computational and financial cost. Unfortunately, due to the low cases present in the UK Biobank it is not possible to detect associations outside the HLA region.

Changes: We have clarified the claim by removing the word “individual” from risk prediction.

13) Minor points:

1. Figure 3a seems overcrowded. I cannot distinguish “shaded” datapoints from not shaded ones, and the mean and confidence interval boxes are very difficult to see. In my opinion this figure should either be split or show only relevant data points. What is the meaning of the size of the dots in this figure? Can the authors explain why it is useful to have a mean and a confidence interval reported for each chromosome as independent disease risk signals might also be independent with regards to ancestry across a single chromosome?

Response: We have made changes to make this much clearer. We also agree with the reviewer's comment that independent SNP signals are more important, while showing the mean and standard deviation for each ancestry doesn't make much sense. That's also a reason why the mean and sd for each chromosome has been removed in the updated Figure 3.

Changes: The distribution of all SNPs' risk ratios at each ancestry are shown as a raincloud plot, while only SNPs significant at the 1% level are shown individually, coloured by chromosome or HLA region.

14) 2. In some of the figures the abbreviations used in the axis labels are not explained. For example “kye” in Figure 1c, “CE” in Figure 2B, “DAF” in Figure 5b, These should all be explained.

Response: Thank you for pointing this out.

Changes: All figures now have their abbreviations explained, and we have harmonised the units and shading between figures (e.g. same colours not used for each ancestry in figures 1 and 2, same units for time used in figures 2 and 5).

15) 3. Supplementary Figure 4.1 is not well explained – does the top plot show the association of any ancestry with MS (per SNP) or only Steppe and Farmer ancestry?

Response: We agree this was unclear.

Changes: We have updated the figure caption to make clear that the top plot is for all 6 ancestries, and have pointed towards the methods.

Reviewer Reports on the First Revision:

Referees' comments:

Referee #1 (Remarks to the Author):

The authors have positively addressed my concerns. My last suggestion would be that the author cite (and maybe integrate in their discussion) the recent work from Kerner et al (Cell Genomics) showing that genetic adaptation to pathogens resulted in increased risk of inflammatory disorders in post-Neolithic Europe.

Referee #2 (Remarks to the Author):

I would like to thank the authors for the responses they provided. Please find below my comments.

1- I am not convinced by the answer regarding the accuracy of LAI in the HLA region. The argument provided is circular (i.e. we don't know if it's accurate but if it was inaccurate, we knew). If LAI in a region is not accurate due to a lack of good references, you may still get long haplotypes. In fact, most phasing methods did not "fail" in long LD regions prior to the existence of high-density non-European reference populations - they were simply phasing based on the closest European haplotypes available which could be inaccurate. Similarly, if you apply RFMix to the HLA region in Hispanics even in the absence of Native American haplotypes you will get long haplotypes with local ancestry assigned (I assume the same is true for ChromPainter), the question is whether these LAI calls and haplotype lengths are correct. There are plenty of papers on how to assess the accuracy of LAI calls (simulations, downsampling of reference haplotypes, etc) in the literature.

2- Related to the point above in response to comment 3, the authors mention "accurate estimation of global ancestry is not critical for the analysis presented here". I agree with this statement, however, in the absence of other measures of accuracy for LAI (see above), the correlation of the sum of LA tracks with global ancestry proportions can serve as an indirect measure of the accuracy of LAI. Previous studies show a high correlation between the output of methods such as MOSAIC and others such as RFMix (Shubert R et al, 2022 and Browning S et al, 2023 for example).

I am still compelled to believe that the main finding (i.e HLA-associated risk factor from MS is enriched for Steppe ancestry) is correct. However, in light of the above points and the (lack of) answers to the previous comments, I cannot judge whether this is the case for sure. Having some quantitative measure of LAI accuracy would be ideal, in its absence, the minimum the authors can do is to discuss this clearly as a limitation in the discussion in the main text.

3- Regarding the 17% increase in out-of-sample variation, the way this number is calculated, as proportion, is not conventional for comparing the amount of variance explained by two models/data/etc in statistics, usually people compare the absolute variance explained values. As such the number 17% is misleading, especially in the discussion where the absolute variance explained values are not quoted.

The current text in the discussion reads: Furthermore, while epistasis between MS-associated variants in the HLA region has been demonstrated before^{28,29,30,31}, we have shown that accounting for this explains 17% more variance than independent SNPs effects alone.

An accurate phrasing would read: Furthermore, while epistasis between MS-associated variants in the HLA region has been demonstrated before^{28,29,30,31}, we have shown that accounting for this explains more variance than independent SNPs effects alone (2.90%, compared to 2.48%).

The message is the same in both cases, but I find the former misleading and overhyped. I recommend removing 17% from the text and representing only absolute values to convey a more accurate message.

4- I strongly recommend adding a section to the discussion to discuss the limitations of this work and how they may affect the interpretation of the results. Based on several points raised by the three reviewers adding such a section can bring the necessary nuance for readers who are not experts in the ancient genome and population genetics.

Referee #3 (Remarks to the Author):

The authors have revised the manuscript which is now easier to follow and they have addressed my concerns satisfactorily in their revision. My only remaining suggestion is to mention Supplementary Figure S9.1 earlier in the manuscript as it provides a helpful overview of the methods used.

Author Rebuttals to First Revision:

Referee #1 (Remarks to the Author):

1) The authors have positively addressed my concerns. My last suggestion would be that the author cite (and maybe integrate in their discussion) the recent work from Kerner et al (Cell Genomics) showing that genetic adaptation to pathogens resulted in increased risk of inflammatory disorders in post-Neolithic Europe.

Response: We are pleased that we have satisfactorily addressed the reviewer's concerns. We added the Kerner et al (2021) and Klunk et al (2022) references during the last round of reviews in response to comments from reviewer #2. We agree that Kerner et al (2023) should also be cited. In response to recent concerns about the Klunk et al. plague paper, raised by Ian Mathieson and colleagues (Barton et al 2023: <https://www.biorxiv.org/content/10.1101/2023.03.14.532615v1>), we have now removed this reference. Their concerns include several technical errors which the authors believe invalidate the results of the paper.

Changes: The sentence ““Similar examples of pathogen-driven evolution have recently been published (Klunk et al., 2022, Kerner et al., 2021)” has been changed to “Similar examples of pathogen-driven evolution have recently been published (Kerner et al., 2021, Kerner et al., 2023)”.

Referee #2 (Remarks to the Author):

1) I am not convinced by the answer regarding the accuracy of LAI in the HLA region. The argument provided is circular (i.e. we don't know if it's accurate but if it was inaccurate, we knew). If LAI in a region is not accurate due to a lack of good references, you may still get long haplotypes. In fact, most phasing methods did not “fail” in long LD regions prior to the existence of high-density non-European reference populations - they were simply phasing based on the closest European haplotypes available which could be inaccurate. Similarly, if you apply RFMix to the HLA region in Hispanics even in the absence of Native American haplotypes you will get long haplotypes with local ancestry assigned (I assume the same is true for chromopainter), the question is whether these LAI calls and haplotype lengths are correct. There are plenty of papers on how to assess the accuracy of LAI calls (simulations, downsampling of reference haplotypes, etc) in the literature.

Response: We appreciate the reviewer's concern over HLA accuracy. We have performed additional analysis as requested, and we also cite further publications which support our line of argument.

We would like to first address phasing and then LAI separately. The reviewer is correct to point out that phasing methods may not fail in the absence of the correct reference haplotypes, and that LAI can result in long haplotypes in the absence of correct haplotypes. However, if the phasing step has failed, the inferred ancestry tracts will be shorter due to incorrect switches in copying. We do not observe this, providing our first line of evidence that phasing is not problematic.

Our ancient samples were phased and imputed using GLIMPSE (Rubinacci et al 2022); our companion paper describes the full results for the validation of this approach on ancient genomes using downsampling and simulations (Mota et al 2022: <https://www.biorxiv.org/content/10.1101/2022.07.19.500636v1.full>). The most pertinent results are that this approach produces high accuracy of phasing and imputation for ancient genomes, comparable to modern genomes, with low switch error rates. It also shows that downstream analyses such as PCA, clustering and (most relevant here) runs of homozygosity (ROH) are similar between the down-sampled imputed and high coverage genomes, except for African genomes. To ensure that we used sites with high phasing and imputation accuracy, we filtered for a joint phasing/imputation score and minor allele frequency (INFO score ≥ 0.5 and MAF ≥ 0.05) produced by GLIMPSE (Rubinacci et al 2022).

Given that we have confidence in the phasing of the ancient genomes, the next question is about the accuracy of the local ancestry inference. The reviewer is right to point out that long haplotypes alone do not guarantee the LAI is correct. We have three reasons (detailed below) to believe that our LAI is accurate in the HLA region: firstly, as suggested by the reviewer, we have downsampled our reference panel and repainted a subset of UK Biobank haplotypes, and show that the painting concordance is sufficiently similar (given the considerable degradation expected) between the HLA and the rest of the genome. Secondly, we cite publications showing that chromopainter is at least as accurate as competing methods. Thirdly, we use two different methods for LAI and both show the same Steppe association of MS-associated variants, giving us confidence in our conclusions.

For the first point, as the reviewer requested, we downsampled the ancient reference panel, removing 20% of individuals from each reference population, and then repainted a subsample (24,000 individuals, chromosome 6) of the UK Biobank. We did not repaint the entire UKB due to the very considerable computational costs associated with doing so. We compared the local ancestry probabilities from the original painting with those from the repainting, within the HLA region and across the entirety of chromosome 6. We show our results as the average of the squared Pearson correlation (“r_squared”) between the two paintings across all individuals, as in Browning et al 2023 ([https://www.cell.com/ajhg/pdfExtended/S0002-9297\(22\)00544-4](https://www.cell.com/ajhg/pdfExtended/S0002-9297(22)00544-4)). The results are summarised in the table below:

Ancestry	Region	r_squared
CHG	Chromosome	0.647803
CHG	HLA	0.653909
EHG	Chromosome	0.793122
EHG	HLA	0.693388
Farmer	Chromosome	0.797812
Farmer	HLA	0.689209
Outgroup	Chromosome	0.750927
Outgroup	HLA	0.861486
WHG	Chromosome	0.796889
WHG	HLA	0.737290
Steppe	Chromosome	0.720947
Steppe	HLA	0.720090

These results show that the HLA is not anomalous in its replicability by these measures - i.e., it performs broadly the same as the other regions of the genome.

For the second point, having shown that the phasing and imputation of the ancient samples are accurate, and that the HLA painting is similar in sensitivity to the other parts of the genome, there is also good support for Chromopainter's accuracy over other state-of-the-art methods. In Wu et al (2021), the authors compare Chromopainter to a range of other LAI methods, concluding that its advantages are "Moderate memory consumption and certain accuracy". In Molinaro et al (2021), the authors suggest that CP-like approaches (here: WINC) are much better than RFmix, and that ELAI is the only other approach which has comparable accuracy.

Finally, we have so far discussed the chromopainting of the UK Biobank, but we use a second method of LAI in this paper, namely the path-based neural network classifier approach which is used in our CLUES and PALM analyses. We use 1000 Genomes modern samples here instead of the UK Biobank, meaning this painting is an entirely independent painting replication. This method is fully introduced and benchmarked in a recent preprint (Pearson et al 2023: <https://www.biorxiv.org/content/10.1101/2023.03.06.529121v1.abstract>). In this paper, the authors show that this approach has high accuracy across a range of ancestries (the same as those used here), is robust to a variety of demographic scenarios, and outperforms GNOMix, a leading LAI method. Importantly, this LAI pursues a methodologically very different approach to chromopainter, and yet both methods support the association of Steppe ancestry with the MS-associated variants in the HLA region.

We hope that we have convinced the reviewer that we are justified in having confidence in our LAI, and that the HLA region is not significantly different from other parts of the genome to elicit concern. We have outlined how the various methods used have been benchmarked in other papers (phasing/imputation, LAI using chromopainter, LAI using a neural network classifier), and we have performed additional analyses as requested to ensure that the

painting within the HLA is not more sensitive to reference panel downsampling than other regions. We therefore think that the LAI within the HLA should not be a cause for concern, especially given that two independent LAI methods draw the same conclusions which are central to this paper. Having said this, we appreciate that this is an important discussion due to its pertinence on the results of the paper, and we thank the reviewer for raising this.

2) Related to the point above in response to comment 3, the authors mention “accurate estimation of global ancestry is not critical for the analysis presented here”. I agree with this statement, however, in the absence of other measures of accuracy for LAI (see above), the correlation of the sum of LA tracks with global ancestry proportions can serve as an indirect measure of the accuracy of LAI. Previous studies show a high correlation between the output of methods such as MOSAIC and others such as RFMix (Shubert R et al, 2022 and Browning S et al, 2023 for example).

I am still compelled to believe that the main finding (i.e HLA-associate risk factor from MS is enriched for Steppe ancestry) is correct. However, in light of the above points and the (lack of) answers to the previous comments, I cannot judge whether this is the case for sure. Having some quantitative measure of LAI accuracy would be ideal, in its absence, the minimum the authors can do is to discuss this clearly as a limitation in the discussion in the main text.

Response: We hope that we have adequately addressed the reviewer’s concerns about the accuracy of LAI through citations and the additional analysis described above. The reviewer is right to point out that the sum of LA tracts should correlate with global ancestry, and this is the method used by NNLS (described in more detail in the methods section), which uses this sum (in the form of “chunklengths”) to calculate global ancestry proportions for each painted individual. We actually use this data in our companion paper “The Selection Landscape And Genetic Legacy Of Ancient Eurasians” (Irving-Pease et al., 2022), which reports regional variation in global ancestry in individuals in the UK Biobank. We also use this data for painted ancient samples in this study (for example the admixture proportions shown as vertical bars in Figure 1). So, although these global ancestry results are not emphasised in this paper, we do make use of the sum of LA tracts through NNLS. NNLS is a commonly used technique first introduced in Hellenthal et al (2014), and all global ancestry results which we report here and in the companion paper are in line with expectations from the literature.

3) Regarding the 17% increase in out-of-sample variation, the way this number is calculated, as proportion, is not conventional for comparing the amount of variance explained by two models/data/etc in statistics, usually people compare the absolute variance explained values. As such the number 17% is misleading, especially in the discussion where the absolute variance explained values are not quoted.

The current text in the discussion reads: Furthermore, while epistasis between MS-associated variants in the HLA region has been demonstrated before^{28,29,30,31}, we have shown that accounting for this explains 17% more variance than independent SNPs effects alone.

An accurate phrasing would read: Furthermore, while epistasis between MS-associated variants in the HLA region has been demonstrated before^{28,29,30,31}, we have shown that accounting for this explains more variance than independent SNPs effects alone (2.90%, compared to 2.48%).

The message is the same in both cases, but I find the former misleading and overhyped. I recommend removing 17% from the text and representing only absolute values to convey a more accurate message.

Response: We have made the requested changes, removing the 17% figure from both the Results and Discussion section. We now only quote the difference in the Results section.

Changes: In the Results section, “Across the entire HLA region, haplotypes explain at least 17% more out-of-sample variation than GWAS (2.90%, compared to 2.48%)” has been replaced with “Across the entire HLA region, haplotypes explain more out-of-sample variation than GWAS (at least 2.90%, compared to 2.48%)”. And in the Discussion, “Furthermore, while epistasis between MS-associated variants in the HLA region has been demonstrated before^{28,29,30,31}, we have shown that accounting for this explains 17% more variance than independent SNPs effects alone” has been replaced with “Furthermore, while epistasis between MS-associated variants in the HLA region has been demonstrated before^{28,29,30,31}, we have shown that accounting for this explains more variance than independent SNPs effects alone”.

4) I strongly recommend adding a section to the discussion to discuss the limitations of this work and how they may affect the interpretation of the results. Based on several points raised by the three reviewers adding such a section can bring the necessary nuance for readers who are not experts in the ancient genome and population genetics.

Response: We believe we have provided sufficient caveats throughout the paper to convey the limitations of our results. These include cautious language, acknowledgements of where data is unavailable or potentially underpowered, and the use of high statistical thresholds.

For example, when reporting results about the association of positively-selected MS-associated variants with disease-associated variants, we say “*We emphasise that although this evidence is strongly suggestive, many of these putative associations may not be statistically robust due to underpowered GWASs and the bias in candidate gene studies*”, and “*We are, however, underpowered to detect specific associations beyond this hypothesis due to poor knowledge of the distribution and diversity of past diseases, poor preservation of endogenous pathogens in the archaeological record, and a lack of well-powered GWASs for many infectious diseases, partly due to widespread vaccination programs*”. We also use cautious language such as we “*suggest that the Steppe ancestry gradient in modern populations - specifically at the HLA region - across the continent causes this phenomenon, in combination with environmental factors*” and “*Together, this evidence suggests that population dispersals, changing lifestyles and increased population density resulted in high and sustained transmission of both new and old pathogens, driving selection in immune response genes which are now associated with autoimmune diseases*”. We also use rigorous and cautious statistical measures throughout, for example we report lower boundaries for the HTRX measures rather than medians or upper boundaries; and we report

only results passing stringent p-value thresholds for genome-wide significance after correcting for multiple testing.

Therefore, because we have discussed the limitations of the various methods throughout the paper and sufficiently caveated our conclusions, we do not believe it is necessary to include a separate paragraph in the Discussion section repeating this information.

Referee #3 (Remarks to the Author):

1) The authors have revised the manuscript which is now easier to follow and they have addressed my concerns satisfactorily in their revision. My only remaining suggestion is to mention Supplementary Figure S9.1 earlier in the manuscript as it provides a helpful overview of the methods used.

Response: We are pleased that we have addressed the reviewer's concerns. We also agree that this figure would be better included earlier in the manuscript, and we have moved it to become Supplementary Figure 1.1.

Changes: We have moved the sentence "An overview of the evidence provided by all methods used can be found in Supplementary Figure 1.1." to be the last sentence of the Introduction.

Reviewer Reports on the Second Revision:

Referees' comments:

Referee #2:

Remarks to the Author:

The authors have answered my concerns and revised the manuscript accordingly. I do not have any further comments or suggestions